# Neanderthal-derived variants increase *SOX9* enhancer activity in craniofacial progenitors that shape jaw development

Kirsty Uttley*, Hannah J. Jüllig*, Carlo De Angelis, Julia M. T. Auer, Ewa Ozga, Hemant Bengani and Hannah K. Long‡

## ABSTRACT

Human facial appearance is a highly variable morphological trait, with both rare and common genetic variants shaping craniofacial morphology between individuals and in disease. Deletions encompassing an enhancer cluster 1.45 megabases upstream of *SOX9* (EC1.45) cause Pierre Robin sequence, a craniofacial disorder characterised by underdevelopment of the lower jaw and associated with cleft palate. We hypothesised therefore that single nucleotide variants within EC1.45 may cause more subtle alterations in facial form. Leveraging recent human evolution, and the distinct jaw morphology of Neanderthals, we investigated the impact of three Neanderthal-derived variants on EC1.45 function. Using zebrafish dual enhancer-reporters, we observed higher activity of Neanderthal EC1.45 in neural crest-derived progenitor cells during a specific developmental window. These EC1.45-active cells reside proximal to and are transcriptionally related to precartilaginous condensations that contribute to craniofacial skeletal development. Mimicking the observed increase in enhancer activity, we overexpressed human SOX9 in EC1.45-active cells and detected an expanded volume of developing cartilaginous precursors. Our work implicates Neanderthal-derived variants in increasing regulatory activity for a disease-associated enhancer, with the potential to impact craniofacial morphology across recent hominin evolution.

KEY WORDS: Transcriptional enhancer, Neanderthal variants, Cranial neural crest cells, Craniofacial development, Gene regulation, Morphological divergence

## INTRODUCTION

Non-coding genetic mutations or variants are increasingly implicated in rare or common human disease through perturbation of gene regulatory elements such as enhancers, which can impact embryonic gene expression patterns and developmental processes (Claringbould and Zaugg, 2021; French and Edwards, 2020; Long et al., 2016; Spielmann et al., 2018; Zhang and Lupski, 2015). Large non-coding

MRC Human Genetics Unit, Institute of Genetics and Cancer, University of Edinburgh, Crewe Road South, Edinburgh EH4 2XU, UK.
*These authors contributed equally to this work

‡Author for correspondence (hannah.long@ed.ac.uk)

K.U., 0000-0003-1929-6705; H.J.J., 0009-0006-1424-5860; H.K.L., 0000-0002-5694-0398

Handling editor: James Briscoe

deletions and translocation breakpoints over 1.2 megabases (Mb) upstream of *SOX9* are implicated in Pierre Robin sequence (PRS) (Tan and Farlie, 2013; Paletta et al., 1994), a malformation characterised by underdevelopment of the lower jaw, backwards displacement of the tongue, airway obstruction and frequent incidence of cleft palate (Amarillo et al., 2013; Benko et al., 2009; Gordon et al., 2009, 2014). While there are many putative regulatory elements at the *SOX9* locus, we previously characterised two enhancer clusters upstream of the PRS translocation breakpoint cluster that are each ablated by at least one patient deletion (Long et al., 2020). These enhancer clusters were active in mouse embryonic craniofacial structures and in *in vitro*-derived cranial neural crest cells (CNCCs), transient multipotent cells that give rise to the majority of craniofacial structures (Bronner and LeDouarin, 2012), and were decommissioned during differentiation to chondrocytes (Long et al., 2020). Genetic ablation of the mouse orthologue of enhancer cluster EC1.45, located 1.45 Mb upstream of *SOX9* (Fig. 1A), impacted lower jaw development and postnatal fitness. Informed by PRS patient mutations that cause severe jaw morphological malformations, this work highlighted a key role for EC1.45 in shaping jaw morphology and function (Long et al., 2020). Based on these observations, we hypothesised that single nucleotide variants (SNVs) within EC1.45 may alter enhancer activity, impacting to a more subtle degree *SOX9* developmental expression and jaw morphology.

Lower jaw shape is highly variable among vertebrate species, associated with a diversity in diet, feeding methods, vocalisation, communication and gait (Coombs et al., 2024; Morales-García et al., 2021; Woronowicz and Schneider, 2019). The fossil record shows dramatic changes in skeletal form across an estimated 4 million years of hominin evolution, including exceptionally rapid rates of mandibular shape evolution compared to other primates (Bergmann et al., 2021; Lacruz et al., 2019; Raia et al., 2018). While coding sequences tend to be highly conserved between modern humans and other hominin species (Suntsova and Buzdin, 2020; Weiss et al., 2021), non-coding sequence changes are much more abundant (Yan and McCoy, 2020). Indeed, non-coding variants can alter enhancer function and developmental gene expression patterns, driving morphological change across evolutionary timescales (Frankel et al., 2011; Long et al., 2016; Prescott et al., 2015; Rubinstein and De Souza, 2013). We previously identified three SNVs within EC1.45 that appear derived in an archaic group of humans, *Homo neanderthalensis* (Neanderthals). These variants are associated with a Neanderthal-specific hypomethylated region, suggestive of a gain of function for the Neanderthal enhancer (Gokhman et al., 2014, 2020; Long et al., 2020).

To explore the regulatory impact of these variants, we leveraged zebrafish as a model system due to broadly conserved craniofacial gene regulatory networks, well-characterised and orthologous

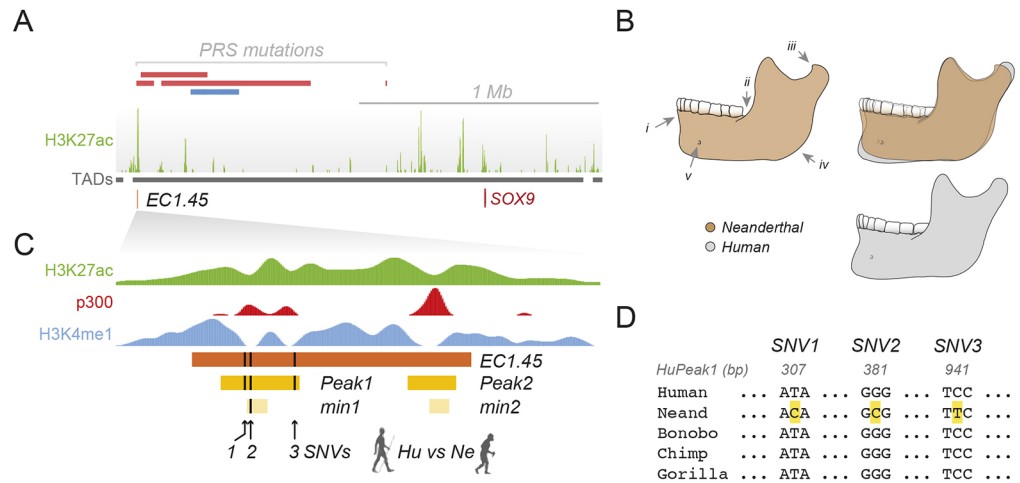

**Fig. 1. Three Neanderthal-derived single nucleotide variants overlap a *SOX9* disease-associated enhancer cluster.** (A) The cranial neural cell-specific enhancer cluster EC1.45 falls within a deletion hotspot and upstream of a cluster of translocation breakpoints identified in individuals with Pierre Robin sequence (PRS). H3K27ac ChIP-seq from CNCCs (Prescott et al., 2015) (green), topological associated domains (TADs) (grey) (Dixon et al., 2012), PRS patient deletions (red) and a cluster of translocation breakpoints (blue) (Amarillo et al., 2013; Benko et al., 2009; Gordon et al., 2009, 2014). (B) Neanderthal-derived jaw morphological features (apomorphies), including (i) posterior position of mental foramen, (ii) retromolar gap, (iii) posterior positioning of anterior marginal tubercle, (iv) laterally expanded condyle and (v) truncated gonion [inspired by Bergmann et al. (2021) and Rosas (2001)]. (C) Schematic of the EC1.45 region, highlighting Peak1-2 and min1-2 (Long et al., 2020). Three SNVs fall within EC1.45, all within Peak1, one of which overlaps min1. ChIP-seq for H3K27ac (green), p300 (red) and H3K4me1 (blue) highlight the enhancer-associated chromatin features of the locus in CNCCs. Data from Prescott et al. (2015). (D) A compressed DNA sequence alignment highlighting the sequence context for the three Neanderthal SNVs within the EC1.45 Peak1 region compared to human. Sequence alignment is also shown for bonobo, chimpanzee and gorilla – additional sequence variants for these species are not shown for simplicity.

craniofacial developmental processes to human, and external and transparent embryonic development (Fox and Waskiewicz, 2024; Medeiros and Gage Crump, 2012; Mork and Crump, 2015; Raterman et al., 2020). Importantly, the formation of Meckel's cartilage and subsequent adjacent mandibular bone formation is a highly conserved developmental process in both zebrafish and human (Eames et al., 2013; Gillis, 2019; Logjes et al., 2018; Reeck and Oxford, 2022; Svandova et al., 2020). Furthermore, *sox9a* mutant zebrafish embryos exhibit craniofacial and cartilage developmental defects analogous to human campomelic dysplasia, which is caused by SOX9 haploinsufficiency, highlighting the conservation of SOX9 function across vertebrate facial formation (Wagner et al., 1994; Yan et al., 2002, 2005).

In this study, we first characterised that EC1.45 is active in a population of neural crest progenitor cells that reside proximal to and appear transcriptionally related to precartilaginous condensations (PCCs) that contribute to the craniofacial skeleton, including to Meckel's cartilage (Kague et al., 2012). Using a dual enhancer-reporter system, we revealed that Neanderthal EC1.45 exhibits increased enhancer activity compared to human. Mimicking this increased activity, we demonstrated that overexpression of human SOX9 in EC1.45-active cells caused an increase in craniofacial precartilaginous template volume, indicating a possible link to jaw morphological changes observed between humans and Neanderthals. Our data implicate ancient hominin sequence variants in driving differential enhancer activity in a specific population of craniofacial progenitors and provide functional insights into the impact of increased *SOX9* expression in these cells. In summary, we have characterised Neanderthal-derived enhancer variants within a human disease locus that increase regulatory activity and, within the context of a complex regulatory domain, may have participated in a stage- and tissue-specific increase in *SOX9* expression during development, contributing to alterations in jaw morphology.

## RESULTS

### Neanderthal-specific variants in a human disease-associated enhancer cluster

Given the pathogenic consequences associated with ablation of the EC1.45 enhancer cluster (Fig. 1A), we hypothesised that DNA sequence changes in this regulatory element may cause a more subtle morphological change. To explore this possibility, we considered recent human evolution, and the distinct jaw morphological features of Neanderthals compared to anatomically modern humans, including a retromolar space, a truncated gonion and a posterior position of the mental foramen (Fig. 1B) (Bergmann et al., 2021; Nicholson and Harvati, 2006; Rosas, 2001). From three high-quality Neanderthal genome sequences, we identified three SNVs overlapping EC1.45 (Fig. 1C,D) (Prüfer et al., 2014; Green et al., 2010; Mafessoni et al., 2020). Two of the Neanderthal variants are present in all three high-quality genomes, while the third (SNV3) is present only in one, suggesting that it may be a sequencing artefact or have been polymorphic in the Neanderthal population.

The EC1.45 element comprises two regions defined by p300 peaks from chromatin immunoprecipitation sequencing (ChIP-seq) called Peak1 and Peak2. We had previously narrowed these down to two minimal active sequences (min1 and min2) that recapitulate the regulatory activity of both EC1.45 and Peak1-2 in an *in vitro* luciferase reporter assay (Long et al., 2020) (Fig. 1C). One of the variants, SNV2, overlaps with the min1 region and creates a new CpG in the Neanderthal genome (Fig. 1D), perhaps related to the Neanderthal-specific DNA hypomethylation at this locus that was previously detected in ancient bone samples (Gokhman et al., 2014, 2020). Sequence variation is also observed for other non-human primate species in Peak1, but this is non-overlapping with the Neanderthal variants described here, and the Neanderthal SNVs are not present in gnomAD v4.0, apart from one instance of the

potentially polymorphic SNV3 variant (1/152,152 rs563260668) (Karczewski et al., 2020). Notably, the three variants do not match the predicted ancestral state and hence they appear to be derived in Neanderthals (Fig. 1D) and thus may be associated with the appearance of Neanderthal-specific jaw features.

## EC1.45 enhancer activity is detected in cells directly adjacent to the developing craniofacial skeleton

To explore enhancer activity dynamics of human EC1.45 across development, we generated a zebrafish Tol2-mediated transgenic reporter line with the Peak1-2 region cloned upstream of the minimal gata2 promoter and *eGFP* (Fig. 2A), which we named *Tg(HuEC1.45-P1P2:eGFP)* [abbreviated to *Tg(HuP1P2:GFP)*]. The expression pattern of the human EC1.45 Peak1-2 enhancer cluster was assessed by confocal imaging of *Tg(HuP1P2:GFP)* embryos crossed to the reporter line *Tg(sox10:mRFP)* (Kucenas et al., 2008; Kirby et al., 2006) (mRFP, membrane-bound red fluorescent protein). During zebrafish development, *sox10* is expressed in CNCCs and continues to be expressed in the cranial PCCs that will form skeletal elements of the viscerocranium, which includes the lower jaw, and the ethmoid plate of the neurocranium, which forms the larval upper jaw (Kucenas et al., 2008; Mork and Crump, 2015). At 1 day post fertilisation (dpf), eGFP was detected broadly in the frontonasal region (Fig. S1A). By 2 dpf, eGFP expression persisted in the frontonasal region and appeared in a paired location adjacent to the oral cavity (Fig. 2B and Movies 1 and 2). Specifically, enhancer activity was detected alongside Meckel's precartilaginous condensations and extended anteriorly along the oral cavity (Eames et al., 2013) (see Fig. 2C and Fig. S1B for schematics of developing zebrafish cartilage structures, orthology to human Meckel's cartilage development and the relative location of EC1.45-P1P2 reporter activity). At 3 and 4 dpf, enhancer activity continued to be detectable adjacent to Meckel's cartilage, at a lateral and slightly dorsal position, and appeared to extend into the Meckel's precartilaginous template at the jaw joint region (Fig. 2B,C, Fig. S1B and Movie 3). These activity patterns were consistent across four founder lines (Fig. S2), and time-lapse imaging from 2-4 dpf confirmed that the paired Meckel's-adjacent signal was the same population at 2, 3 and 4 dpf (Movie 4). For the enhancer-positive cells in the frontonasal region, a posterior subset of these cells appeared to contribute to the forming palate between 2 and 3 dpf, with eGFP-expression observed in the ethmoid plate at these stages (Movie 5, arrow). This is consistent with previous work that used fate mapping to show that frontonasal CNCCs adjacent to the nasal epithelium populate the ethmoid plate (Swartz et al., 2011). Similar domains of enhancer activity were observed for *Tg(HuP1P2: GFP)* crossed to a *col2a1a* reporter line, *Tg(col2a1a:RFP)*, which also marks precartilaginous condensations and chondrocytes during craniofacial development (Dale and Topczewski, 2011; Paudel et al., 2022) (Fig. S1B,C).

The expression domains for EC1.45-P1P2 observed during zebrafish development were reminiscent of enhancer reporter activity we previously described for the mouse embryo (Long et al., 2020). At mouse embryonic day 9.5 (E9.5), EC1.45 was active in the developing frontonasal prominence, while at E11.5, enhancer activity was observed in the lateral nasal process, medial nasal process, maxillary process, mandibular process, periocular mesenchyme and limb bud. Strikingly, we also detected eGFP expression in the developing fin at 3 and 4 dpf for the *Tg(HuP1P2: GFP)* line (Fig. S1D), in keeping with the limb expression patterns observed during mouse development (Long et al., 2020). To confirm that EC1.45 enhancer activity overlaps with endogenous

*sox9* gene expression, we performed hybridisation chain reaction RNA fluorescent *in situ* hybridisation (HCR RNA-FISH) for the *Tg(HuP1P2:GFP)* transgenic line, using probes against *eGFP* and *sox9a* mRNA at 2 dpf. We focused on *sox9a*, based on previous work demonstrating a greater role for this gene in pharyngeal cartilage development compared to *sox9b* (Yan et al., 2005). Enhancer activity, marked by *eGFP* mRNA expression, was observed in both the frontonasal region and a paired region adjacent to Meckel's cartilage (Fig. 2D), correlating with the expression domains previously observed from eGFP protein fluorescence. Quantification of enhancer activity at the jaw-adjacent region showed a high degree of overlap with *sox9a* expression (around 96% *of eGFP*-positive cells also expressed *sox9a*), while enhancer activity was not detected in a nearby domain of *sox9a* expression that likely represents condensing cartilage (Fig. 2D,E). This is in concordance with our previous observations *in vitro* that EC1.45 activity is rapidly decommissioned during chondrogenesis (Long et al., 2020). Together, a zebrafish reporter of human EC1.45 regulatory activity matches key expression domains observed from mammalian development, and overlaps with endogenous expression of *sox9a,* revealing a conserved regulatory logic for EC1.45 enhancer activity from fish to human. A zebrafish enhancer reporter line therefore provides enhanced temporal and spatial insights into the developmental activity of the EC1.45 disease-associated regulatory element, particularly for a population of cells in proximity to the developing lower jaw.

## Neanderthal variants increase EC1.45 enhancer activity compared to human during early craniofacial development

To explore how the three Neanderthal-derived SNVs impact the activity of the EC1.45 enhancer cluster, we leveraged the Q-STARZ assay (Quantitative Spatial and Temporal Assessment of Regulatory element activity in Zebrafish), which enables the activity of two enhancers to be tested using a single enhancer reporter construct during zebrafish embryonic development (Fig. 3A) (Bhatia et al., 2021; Uttley et al., 2023). Firstly, the Neanderthal EC1.45 Peak1-2 sequence was placed upstream of a minimal promoter and *eGFP*, and separated by an insulator sequence from the human version of Peak1-2 that drives expression of mCherry [*Tg(NeP1P2:eGFP; HuP1P2:mCh)*, abbreviated to *Tg(Ne:GFP;Hu:Ch)*] (Fig. 3A). Stable transgenic lines were generated by Tol2-mediated transposition, and embryos were collected for imaging by confocal microscopy across the first 3 days of development. The spatial activity of Neanderthal Peak1-2 (eGFP signal) appeared highly similar to that of the human enhancer (mCherry signal) across development (Fig. 3B and Fig. S3A,B). However, we observed that the absolute expression level of eGFP driven by Neanderthal Peak1-2 appeared to be detectably higher than for mCherry driven by human Peak1-2, especially at 2 dpf (Fig. 3B). We recapitulated these results, with observed higher activity of the Neanderthal enhancer at 2 dpf, in a 'swap' line with the human enhancer upstream of eGFP, and the Neanderthal enhancer upstream of mCherry [*Tg(HuP1P2:eGFP;NeP1P2:mCh)*, abbreviated to *Tg(Hu:GFP;Ne:Ch)*] (Fig. 3A,B). Finally, we created a transgenic line in which the expression of both eGFP and mCherry were controlled by the human enhancer sequence [*Tg(HuP1P2:eGFP; HuP1P2:mCh)*, abbreviated to *Tg(Hu:GFP;Hu:Ch)*]. As expected, we did not observe a difference in eGFP and mCherry expression levels at these developmental stages (Fig. 3A,B and Fig. S3A,B).

To quantify the observed differences, we measured mean fluorescence intensity for eGFP and mCherry in the frontonasal regions (including some cells contributing to the developing palate)

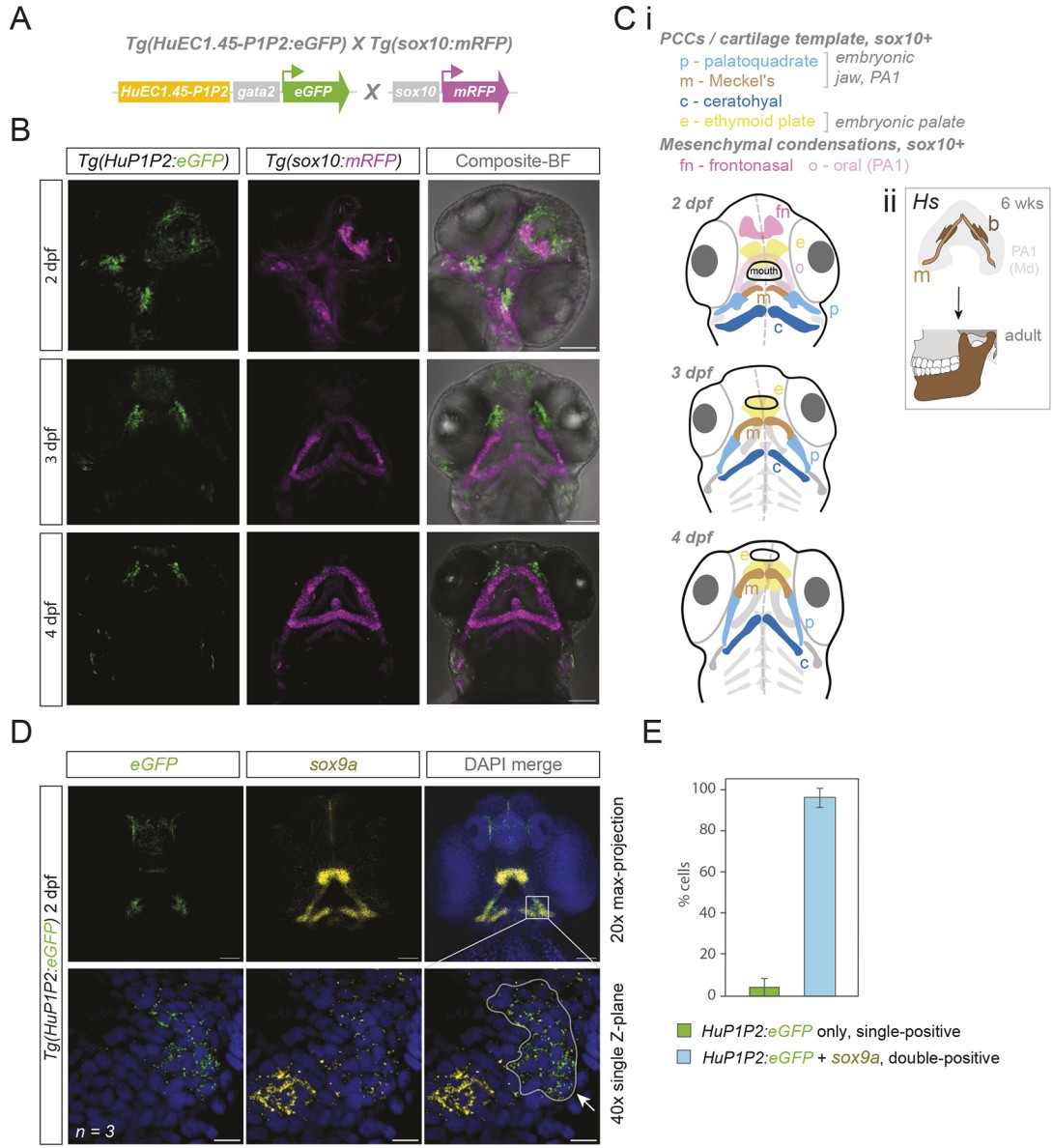

**Fig. 2. EC1.45 Peak1-2 is active during embryonic facial development in the frontonasal region and adjacent to the developing jaw.** (A) Schematic of the cross between the human EC1.45 Peak1-2 enhancer reporter *Tg(HuEC1.45-P1P2:eGFP)* and the *Tg(sox10:mRFP)* reporter. The Peak1-2 region is upstream of a minimal gata2 promoter driving eGFP expression and flanked by insulators and Tol2 sites. (B) Confocal microscopy images (maximum intensity projections) of the cranial region of embryos at 2, 3 and 4 dpf for *Tg(HuEC1.45-P1P2:eGFP)* crossed with *Tg(sox10:mRFP)*. Scale bars: 100 μm. (C) (i) Schematics of the zebrafish embryonic cranial region (ventral view) indicating the location of *sox10:mRFP* reporter activity in PCCs, which mature to cartilage templates from 2 to 4 dpf, and additional neural crest-derived mesenchymal populations (fn and o). See also Eames et al. (2013). (ii) Human (Hs) embryonic lower jaw development at 6 weeks (upper; see also Logjes et al., 2018) compared to adult jaw (lower, brown). (D) HCR RNA-FISH for eGFP and *sox9a* in zebrafish embryos at 2 dpf from the *Tg(HuEC1.45-P1P2:eGFP)* transgenic line. Top: 20× max projection images for the cranial region. Bottom: single *z*-plane images used for quantification, demonstrating overlap of EC1.45 activity with *sox9a* expression (*n*=3 embryos). Scale bars: 100 μm for 20× images and 50 μm for 40× zoom. Arrow and outline highlight the region of *eGFP* and *sox9a* double-positive cells. (E) Quantification of the proportion of eGFP-positive cells from D that express only eGFP (4%) or are also positive for *sox9a* (96%), from 40× images (*n*=3 embryos, data are mean±s.d.). b, bone; m, Meckel's cartilage; Md, mandibular prominence; p, palatoquadrate; e, ethmoid plate; c, ceratohyal; fn, frontonasal; o, oral; PA1, pharyngeal arch 1; BF, brightfield.

and mandible-adjacent structures from 1 to 3 dpf (see Fig. S3C for representative images of surfaces generated to quantify expression). Of note, when imaging the Q-STARZ reporter lines, image acquisition settings were optimised to avoid saturation for the brighter Neanderthal signal, while maintaining detection of the weaker human signal. Confirming our observations, we detected significant differences in the mean fluorescence intensity for eGFP and mCherry at 2 dpf in the *Tg(Ne:GFP;Hu:Ch)* and *Tg(Hu:GFP; Ne:Ch)* lines, but not for the *Tg(Hu:GFP;Hu:Ch)* line (Fig. 3C).

Specifically, the Neanderthal enhancer drove significantly higher expression levels at this developmental stage for both the frontonasal and future jaw regions (Fig. 3B,C). While we observe some evidence for a similar bias at 1 and 3 dpf, this was not statistically significant for the two swap lines (Fig. S3D,E).

To explore further the differences in enhancer activity observed at 2 dpf, we performed timelapse imaging from 1 to 2 dpf which revealed no apparent differences in the onset or spatial localisation of enhancer activity in the jaw region (Movies 6 and 7).

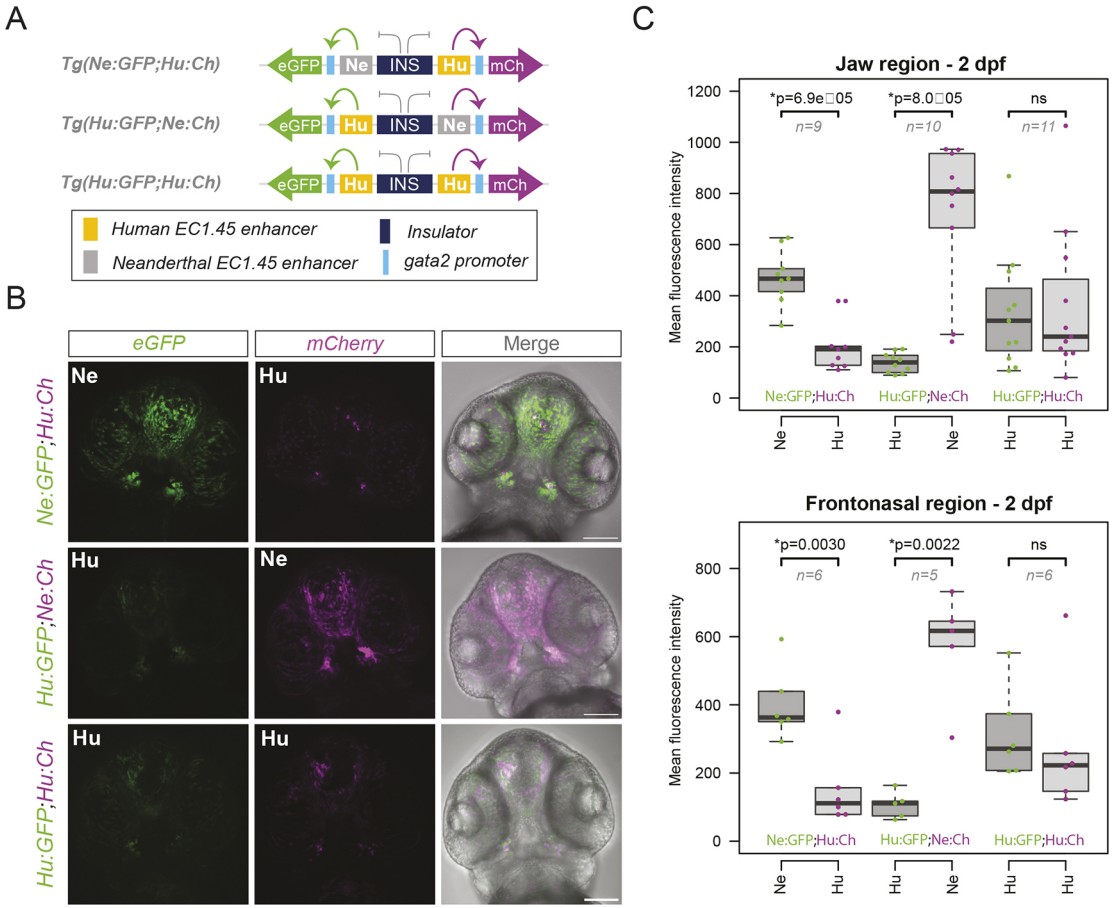

**Fig. 3. Neanderthal SNVs drive increased developmental enhancer activity compared to human.** (A) Schematic of three Q-STARZ transgenic reporter lines created to compare human and Neanderthal EC1.45 activity. For *Tg(Ne:GFP;Hu:Ch)*, the Neanderthal Peak1-2 sequence is upstream of eGFP and the human Peak1-2 sequence is upstream of mCherry. For *Tg(Hu:GFP;Ne:Ch)*, the enhancers are exchanged. For *Tg(Hu:GFP;Hu:Ch)*, the human Peak1-2 is upstream of both eGFP and mCherry. (B) Representative confocal images (maximum intensity projections) for embryos at 2 dpf, ventral view of cranial region. Scale bars: 100 μm. (C) Quantification from B of the jaw region (upper) and frontonasal region (lower) plotted as a box-and-whisker plot, representing median and interquartile range. *P* values from a Wilcoxon signed-rank test are shown. ns, not significant. See Fig. S3C for representative segmented regions used for quantification.

HCR RNA-FISH for *eGFP* and *mCherry* in the dual-reporter transgenic lines further demonstrated that most enhancer-active cells are positive for both human and Neanderthal enhancer activity (Fig. S4). Of note, single-positive cells were detected for all three lines, most commonly at the periphery of the enhancer activity domain, perhaps reflecting stochasicity of enhancer activity in regions where key transcription factor (TF) expression reaches a sub-threshold level for driving robust enhancer activity (Uttley et al., 2023). A skew of detected single-positive eGFP- or mCherry-expressing cells for the *Tg(Ne:GFP;Hu:Ch)* or *Tg(Hu:GFP;Ne:Ch)* lines, respectively, further supports that the Neanderthal enhancer has stronger activity at this time point. Together, three Neanderthal-derived SNVs within EC1.45 increased enhancer activity in a temporally controlled manner during a specific window of embryonic craniofacial development.

### EC1.45 is active in cranial neural crest and precartilaginous mesenchymal progenitor cells

To explore the relative activity and cell type specificity of the human versus Neanderthal EC1.45 further, we performed single cell-RNA sequencing (scRNA-seq) for the two human-Neanderthal enhancer reporter lines at 2 dpf. Fluorescence-activated cell sorting (FACS) was used to isolate all cells that were eGFP positive, mCherry positive or double positive from dissected embryonic cranial

regions at 2 dpf (Fig. S5A,B). This resulted in 40,000 cells for the *Tg(Hu:GFP;Ne:Ch)* line and 10,000 cells for the *Tg(Ne:GFP;Hu:Ch)* line, which were processed as two separate samples (Fig. S5B).

Following quality control and filtering, we carried out K-nearest neighbour analysis and clustering on 8467 cells from the *Tg(Hu:GFP;Ne:Ch)* line, yielding nine distinct clusters. Cell type annotation for these clusters was performed through identification of cluster-specific marker genes, aided by the Daniocell atlas (Sur et al., 2023; Farrell et al., 2018) (Fig. 4A, Fig. S5C and Table S1). This annotation revealed three CNCC-like clusters marked by canonical neural crest marker genes, such as *twist1a* and *snai1a/2*, including a frontonasal population (expressing *alx1* and *alx4a/b*; Mitchell et al., 2021), and two clusters resembling pharyngeal arch 1 (PA1) populations (expressing *dlx* genes and *barx1*; Talbot et al., 2010; Simões-Costa and Bronner, 2015), one of which appeared more proliferative (Fig. 4A,D and Fig. S5C,D). A number of additional clusters were annotated as various neuronal cell types (Fig. 4A and Fig. S5C).

To identify enhancer-active cells within this dataset, we plotted the expression of either eGFP or mCherry, finding a notable enrichment in the CNCC clusters, particularly those of PA1 (Fig. 4B and Fig. S5E). Cells were then categorised as having enhancer activity for the human (eGFP positive), Neanderthal (mCherry positive) or both reporters (double positive) by the presence of three

or more eGFP and/or mCherry reads. Once again, the highest proportion of cells with three or more mCherry or eGFP reads were the two PA1 CNCC clusters (43.4 and 45.9%), suggesting that these cells exhibit the greatest enhancer activity at this stage (Fig. 4B,C and Table S2). In keeping with endogenous EC1.45 regulating human *SOX9* during development, enrichment of eGFP and mCherry expression in the CNCC clusters was concurrent with *sox9a* expression (Fig. 4B,C and Fig. S5E). Indeed, while we would not expect all *sox9a*-expressing cells to be EC1.45 positive, we do find that the majority of EC1.45 active cells express *sox9a* (i.e. 71% of all cells expressing *eGFP/mCherry* also express *sox9a*, Fig. 4C,D). This is in keeping with our earlier observations from HCR RNA-FISH, where the EC1.45-active domain lies within a wider domain of *sox9a* expression (Fig. 2D,E). The lower

level of *sox9b* expression in these clusters is consistent with a lesser role for *sox9b* in craniofacial development in the zebrafish compared to *sox9a* (Fig. S5E,F) (Yan et al., 2005).

We next explored further whether differences in enhancer activity may be due to distinct cell type expression patterns. However, we did not observe any differences in the cell type identity of human versus Neanderthal active cells from our scRNA-seq data, which appear to broadly group together in the CNCC clusters and express similar marker genes (Fig. 4B,C and Fig. S5G). Similar results were observed from the smaller sample of cells obtained from the *Tg(Ne: GFP;Hu:Ch)* line, where 407 cells were grouped into three clusters of PA1 CNCCs, FN CNCCs and neuronal cells (Fig. S6A,B), where the eGFP and mCherry expression was again most enriched in the PA1 CNCC cluster (Fig. S6C). These observations were in

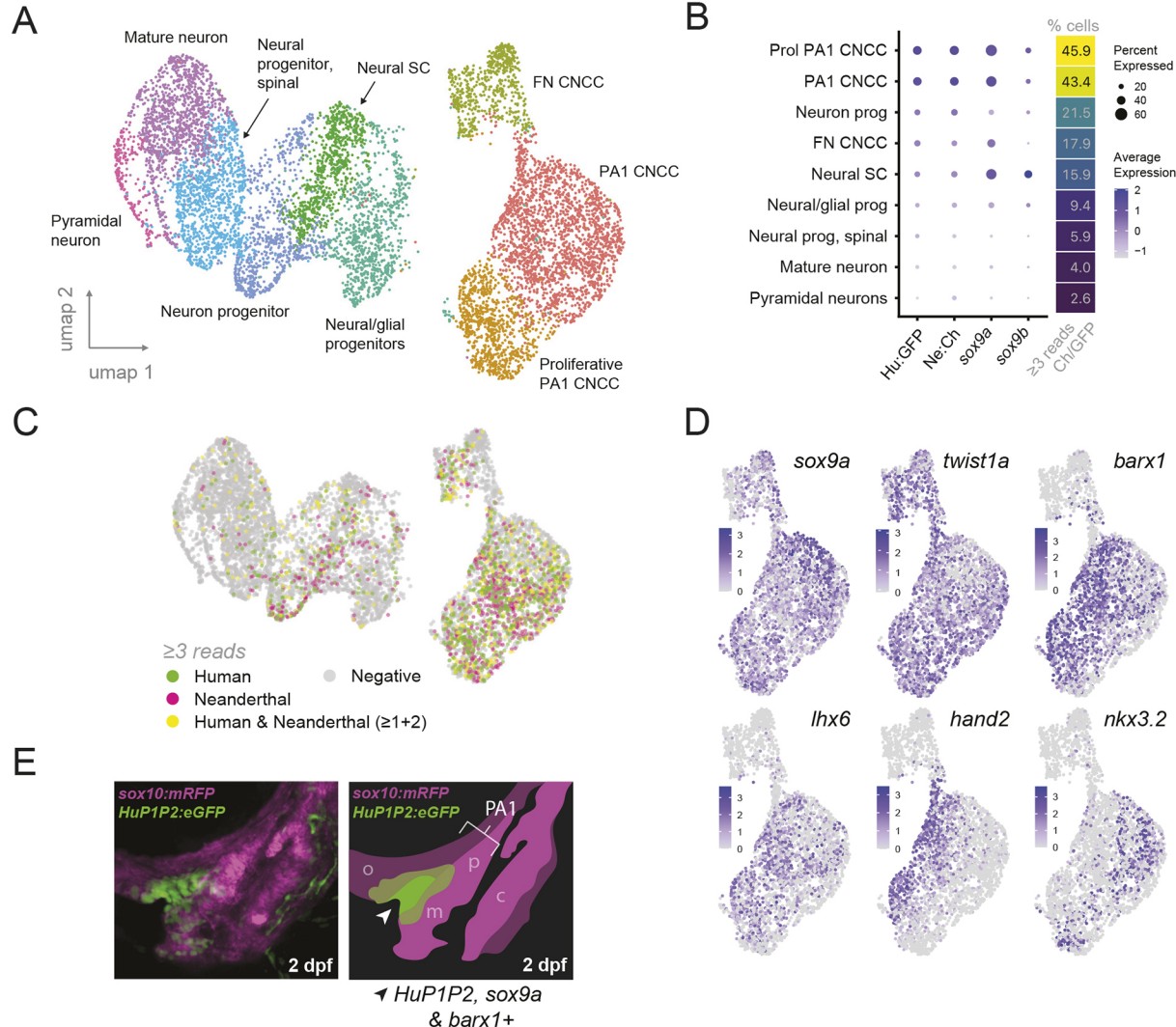

**Fig. 4. EC1.45 active cells transcriptionally resemble cranial neural crest cell-derived facial mesenchymal condensations.** (A) Cells from scRNA-seq for *Tg(Hu:GFP;Ne:Ch)* embryos at 2 dpf, visualised on a Uniform Manifold Approximation and Projection (UMAP) plot created by Louvain clustering using Seurat. Clusters were annotated aided by marker gene expression and the Daniocell atlas. (B) Dot plot showing average expression level and percentage of cells expressing *eGFP*, *mCherry*, *sox9a* or *sox9b* (clusters from A). Percentage of cells that express greater than three reads of eGFP and/or mCherry are shown for each cluster. (C) Expression of eGFP (≥3 reads, green) or mCherry (≥3 reads, magenta), or both (≥3 reads in total for eGFP and mCherry, yellow), in single cells visualised on UMAP plots. (D) Expression of neural crest marker genes (*sox9a* and *twist1a*), facial condensation marker genes (*sox9a*, *barx1* and *lhx6*) and genes that pattern PA1 (*hand2* and *nkx3.2*) in single cells visualised on UMAP plots (subset of clusters from A). (E) A lateral image of developing craniofacial structures at 2 dpf for a cross between *Tg(HuEC1.45-P1P2:eGFP)* and *Tg(sox10:mRFP)*. Schematic (right) and arrowhead highlight the domain of activity for HuEC1.45-P1P2. PA1, pharyngeal arch 1; SC, stem cell; FN, frontonasal; CNCC, cranial neural crest cell; m, Meckel's precartilaginous condensation (PCC); p, palatoquadrate PCC; c, ceratohyal PCC; o, oral mesenchymal condensations.

accordance with our earlier HCR RNA-FISH analysis of mCherry and eGFP transcripts, which appeared to broadly overlap spatially at 2 dpf, especially in the core of the enhancer activity domain (Fig. S4).

To further examine the neural crest cell clusters with highest enhancer activity, we investigated genes known to pattern PA1 at this stage of development. We observed expression of *barx1*, *lhx6*, *hand2* and *nkx3.2* in subsets of the neural crest cells from the two PA1 clusters, and broad expression of *sox9a* and *twist1a* (Fig. 4D). Mirroring the known spatial expression of these markers, we found that *hand2* (expressed in the ventral domain of PA1) and *nkx3.2* (expressed in the joint-forming intermediate domain of PA1) were expressed in a mutually exclusive and restricted pattern, while *barx1* and *lhx6* (expressed in both dorsal and ventral regions of PA1) exhibited more broad expression across the cluster, including an overlap with *hand2* (Fig. 4D) (Nichols et al., 2013; Paudel et al., 2022). Comparing this to the pattern of *eGFP* and *mCherry* expression, EC1.45 Peak1-2 enhancer-active cells are broadly distributed across the PA1 clusters, similar to the *barx1* and *lhx6* expression patterns (Fig. 4C,D). This is intriguing given published work showing a key role for *barx1* and *lhx6* in jaw mesenchymal condensations that mature to form PCCs and give rise to facial cartilages (Nichols et al., 2013; Paudel et al., 2022). Significantly, *barx1*-expressing cells in the jaw region have been shown to activate *sox9a* during maturation into PCCs (Paudel et al., 2022). Furthermore, *barx1* mutant embryos exhibit reduced *sox9a* and *sox10* expression in the developing jaw, and defects in chondrogenesis (Nichols et al., 2013). The association of EC1.45 enhancer activity with mesenchymal condensation markers is compelling, as it supports our hypothesis from enhancer reporter imaging that EC1.45 Peak1-2 is active in the vicinity of Meckel's PCCs from 2 dpf (Fig. 2B,C, Fig. 4E and Fig. S1B). Together, scRNA-seq from two dual enhancer reporter lines supports the conclusion that Neanderthal SNVs do not appear to impact the cell type-specific activity of the EC1.45 regulatory element during early zebrafish craniofacial development. Instead, Neanderthal regulatory variants appear to drive stronger enhancer activity during facial development, including in cells that appear primed to contribute to cartilage formation in the developing jaw region at 2 dpf.

To further explore which tissues EC1.45-positive cells give rise to, and whether we observe direct contribution to jaw cartilage formation, we created a Cre-driver construct controlled by the human EC1.45-P1P2 enhancer sequence (referred to as *HuEC1.45-P1P2:Cre*). We decided to leverage a lineage-tracing strategy as our previous work had suggested that the EC1.45 enhancer is decommissioned during chondrogenesis and therefore the enhancer reporter signal was not expected to persist in the developing cartilage (Long et al., 2020). The *HuEC1.45-P1P2: Cre* construct was injected with Tol2 mRNA into one-cell embryos resulting from a cross between a lineage tracing reporter line *Tg(ubi: loxP-AmCyan-loxP-ZsYellow)* [abbreviated to *Tg(ubi:CSY)*] and the *Tg(sox10:mRFP)* line (Fig. S7A). By imaging ZsYellow expression, this approach facilitated the tracking of the progeny of enhancer-active cells that had expressed the Cre driver in comparison to the *sox10:mRFP* reporter, which marks neural crest and PCCs. Injected embryos were screened at 1 dpf for craniofacial signal, and further screened at 2 dpf for signal in the paired Meckel's adjacent region that is characteristic of EC1.45 enhancer activity (Fig. S7Bi). We identified embryos with ZsYellow signal in the expected tissues at these stages (Fig. S7B), along with some spurious recombination, as is expected for F0 embryos. For one embryo that showed ZsYellow-positive cells adjacent to the

Meckel's PCC at 2 dpf, we carried out time-lapse imaging for 12 h from 2 dpf (Fig. S7Bi and Movie 8) and observed emergence of a ZsYellow-positive *sox10*-positive cell overlapping with Meckel's cartilage. ZsYellow-positive *sox10*-positive cells were also detected in the developing palatoquadrate PCC (Fig. S7Bi). At later developmental stages, we also identified embryos that exhibited ZsYellow signal in the expected enhancer pattern, in addition to ZsYellow expression in cells within the adjacent Meckel's cartilage at 3 and 4 dpf, including in cells with characteristic elongated chondrocyte morphology (Fig. S7Bii,iii, arrowheads).

Based on these observations, we returned to our enhancer reporter line *Tg(HuP1P2:GFP)* to investigate whether this line could be used for short-term lineage tracing (Paudel et al., 2022). From our earlier *in vitro* studies, we expect EC1.45 enhancer to be turned off during chondrogenesis (Long et al., 2020); however, the eGFP reporter may persist for longer than the enhancer is active and reporter transcription has ceased. Indeed, at 2 dpf we observed eGFP-positive cells overlapping with *sox10* expression in the Meckel's PCC for the *Tg(HuP1P2:GFP)* line (Fig. S8A, white arrowheads; Fig. 4E and Movie 1), and we repeated this observation with a cross to the *Tg(col2a1a:RFP)* line (Fig. S8B, white arrowhead). To confirm we could also observe this activity in an independent enhancer reporter line, we crossed the *Tg(Ne:GFP;Hu: Ch)* line to the *sox10* reporter, and again observed eGFP-positive cells within the Meckel's PCC marked by mRFP expression, this time driven by the Neanderthal enhancer (Fig. S8C, white arrowheads). Together, these results support the model that EC1.45-active cells can contribute to PCCs and give rise to mature chondrocytes of the developing lower jaw.

## SOX9 overexpression in EC1.45-positive lineage impacts precartilaginous mesenchymal condensations

Our observation that Neanderthal-specific DNA sequence alterations increase EC1.45 enhancer activity implies an associated increase in endogenous *SOX9* expression during embryonic facial development, potentially affecting jaw shape. We therefore reasoned we could mimic this change through transient overexpression of human SOX9 in EC1.45-active cells during zebrafish development, to reveal how increased enhancer activity may impact jaw morphology. We therefore created a Tol2 construct in which human EC1.45-P1P2 controls expression of human *SOX9*, followed by the ribosome skipping sequence T2A and eGFP, *HuEC1.45-P1P2:hSOX9-T2A-eGFP* (Fig. 5A). We injected the overexpression construct together with Tol2 mRNA into one-cell *Tg(sox10:mRFP)* embryos to visualise the impact on craniofacial skeletal development, using the *HuEC1.45-P1P2: eGFP* Tol2 construct as a control (Fig. 5A).

Initially, injection of the SOX9 overexpression plasmid caused a spectrum of developmental phenotypes at 1 dpf. We therefore reduced the amount of SOX9 plasmid injected by combining with the eGFP-only construct at a 1:1 or 1:3 ratio. At these reduced SOX9 levels, the majority of injected embryos appeared grossly normal with milder and less frequent developmental defects (Fig. S9A). Only embryos with normal morphology were selected for imaging, analysis and quantification. For both *hSOX9-T2A-eGFP* and *eGFP* overexpression, we observed mosaic eGFP expression consistently in the face at 1 dpf, indicating success of the injections (Fig. S9B). Embryos from four replicate experiments with detectable eGFP expression and screened for lack of overt developmental abnormalities were selected for confocal imaging at 2 dpf (Fig. 5B). To assess the impact of SOX9 overexpression, we quantified the volume of the mRFP-positive mesenchymal

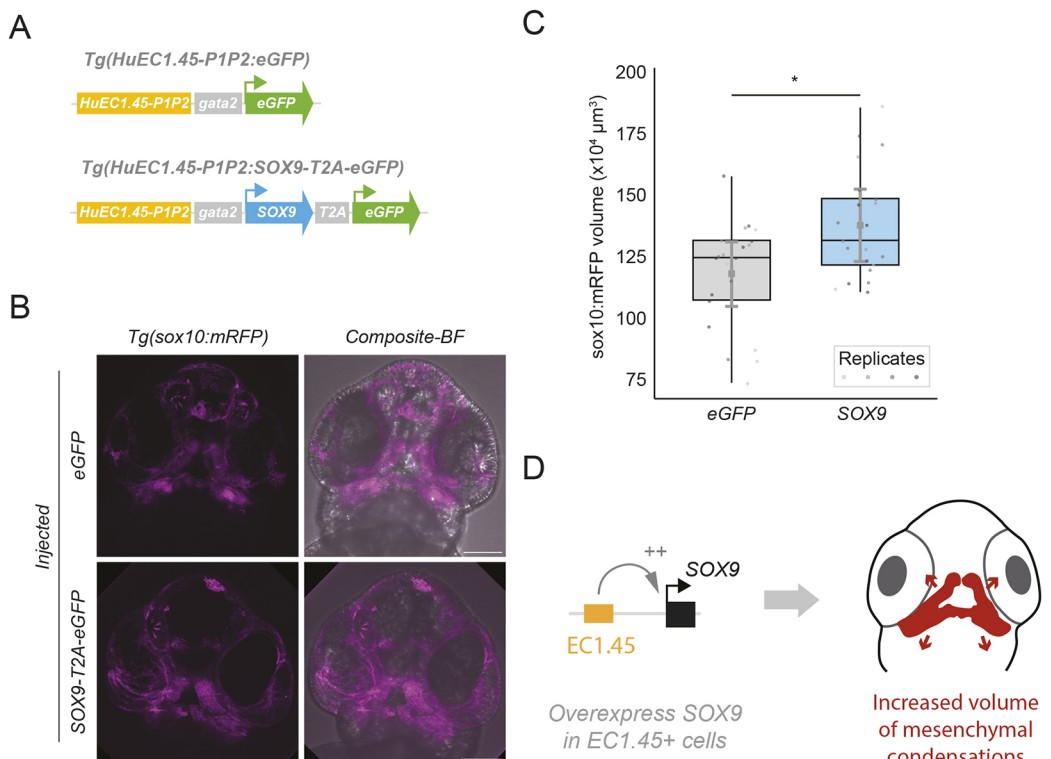

**Fig. 5. Overexpression of human SOX9 during zebrafish embryonic development in EC1.45 active cells impacts craniofacial development.**
(A) Schematic of the *HuEC1.45-P1P2:hSOX9-T2A-eGFP* construct. The Peak1-2 region is upstream of a minimal gata2 promoter driving human SOX9 expression, followed by T2A-eGFP. The *HuEC1.45-P1P2:eGFP* construct from Fig. 2A was used as a control. (B) Representative maximum intensity projections of confocal microscopy images for a ventral view of the craniofacial region of a 2 dpf embryo. Surfaces were derived from mRFP signal encompassing the mesenchymal cells of the Meckel's and palatoquadrate precartilaginous condensations, and the regions extending along the oral ectoderm (see Fig. S9C for representative segmented surfaces). Segmented volume for eGFP-injected image=126×10$^4$ μm$^3$; SOX9-T2A-eGFP injected image=133×10$^4$ μm$^3$. (C) Quantification of *sox10:mRFP*-marked segmented volumes from B. Black box-and-whisker plots show the distribution of volumes between eGFP- or SOX9-injected embryos, representing median and interquartile range, black whiskers indicate minimum and maximum values of data points. For eGFP group mean=120×10$^4$ μm$^3$, median=126×10$^4$ μm$^3$; SOX9 group mean=139×10$^4$ μm$^3$, median=133×10$^4$ μm$^3$. Grey error bars indicate the 95% credible intervals for the Bayesian mixed effects model posterior estimates; grey squares indicate the posterior mean estimates. The estimated increase in *sox10:mRFP*-positive volume for the SOX9-injected group was 19.6×10$^4$ μm$^3$ (95% probability indicating a credible interval between 3.3×10$^4$ μm$^3$ to 36.5×10$^4$ μm$^3$). Wilcoxon test *P*=0.031. (D) Schematic depicting that overexpression of human SOX9 in EC1.45 active cells leads to an increased volume of mesenchymal condensations at 2 dpf. BF, brightfield.

condensations at the jaw region, including the Meckel's PCC, from the *Tg(sox10:mRFP)* injected transgenic embryos (Fig. S9C). Strikingly, this revealed a small but significant increase in the volume of the mRFP-positive region in the SOX9 overexpression embryos compared to eGFP overexpression alone (Fig. 5C, Wilcoxon signed-rank test, *P*<0.05). We also performed a Bayesian mixed-effects model to control for multiple replicates and variation between injections. This model supported the finding that overexpression of SOX9 driven by the human EC1.45 Peak1-2 enhancer cluster significantly increases the mRFP-positive volume marking neural crest and PCCs (Fig. 5C, posterior probability >95%). These results therefore suggest that overexpression of human SOX9 in cells where EC1.45 is active impacts the development of PCCs that will go on to form aspects of the craniofacial skeleton (Fig. 5D).

In summary, we have uncovered an increase in activity for the Neanderthal EC1.45 orthologous enhancer cluster during craniofacial development and characterised EC1.45 regulatory activity in neural crest-derived mesenchymal cells, including in the jaw-forming region. These cells appeared transcriptionally related to a pre-cartilaginous state, which is associated with *barx1* expression, and by lineage tracing can contribute to PCC formation. We further demonstrated that overexpression of SOX9 in

EC1.45-active cells can impact the volume of condensations associated with jaw formation. We therefore propose that alteration of the EC1.45 enhancer sequence during Neanderthal evolution may have promoted an increase in enhancer activity during a window of craniofacial development that could contribute to altered abundance or morphology of cartilaginous precursors.

Finally, we hypothesised that increased EC1.45 activity driven by the three Neanderthal-derived SNVs may be due to altered TF binding and therefore performed differential motif calling for SNV1 and SNV2, which were detected in all three Neanderthal genomes. This analysis revealed several candidate TFs, a subset of which were expressed in the zebrafish EC1.45-active cell populations at 2 dpf and also in human embryonic facial cell types (Tables S3 and S4 and Fig. S10) (Khouri-Farah et al., 2025 preprint). These TFs, which include TEAD1/3, JUN, XBP1, CREB3L2 and SOX9 itself, represent excellent candidates for future study, in addition to exploration of the new CpG dinucleotide generated by SNV2. Importantly, there are many additional Neanderthal SNVs across the *SOX9* regulatory domain, and hence the impact of the EC1.45 regulatory changes should be considered within the wider context of the locus. Therefore, while the Neanderthal EC1.45 SNVs will not alone explain mandibular morphological variation between human and Neanderthal, these regulatory changes may have contributed to

increased *SOX9* expression during a specific developmental window, leading to alteration of the craniofacial skeletal template and subtle morphological changes in jaw shape (Fig. 6).

## DISCUSSION
Here, we have investigated the impact of three Neanderthal-derived regulatory variants on enhancer function at the *SOX9* locus in the context of craniofacial development and recent hominin evolution. Spatial expression of both the human and Neanderthal EC1.45 enhancer was initially restricted to a pool of neural crest-like progenitor cells during early craniofacial development located in the frontonasal region from 1 dpf, with a subset of cells contributing to the embryonic palate. Later, enhancer activity was also detected in a jaw-adjacent region from 2-4 dpf, between the oral ectoderm and developing Meckel's cartilage, reaching anteriorly along the oral cavity, and extending into the jaw joint region at later stages. From lineage-tracing experiments, enhancer-active cells were observed to contribute to Meckel's cartilage, an embryonic jaw structure (Fig. S7). We therefore propose that EC1.45 enhancer-active mesenchymal progenitor cells contribute to forming PCCs and thus help to shape skeletal structures of the craniofacial complex (Akiyama et al., 2005).

Strikingly, although differing by only three SNVs, Neanderthal EC1.45 exhibited increased activity during early craniofacial development, especially at 2 dpf. From time-lapse imaging, scRNA-seq and spatial assessment of enhancer activity domains, the human and Neanderthal enhancers were active in broadly overlapping cell types and spatial domains, suggesting for the most-part that activity differences were driven by increased Neanderthal EC1.45 activity in individual cells. A number of PA1 mesenchymal and PCC markers were robustly expressed in the EC1.45 enhancer-active cell populations at 2 dpf, including *pax9*, *lhx8a* and *lhx6*, which are associated with facial condensations, and *barx1*, which indicated a contribution of cells to PCCs (Nichols et al., 2013; Paudel et al., 2022). It has been shown that changes to the shape and size of PCCs can translate into an altered shape of the resultant adult craniofacial structures (Paudel et al., 2022). Given that Neanderthal variants increase the activity of EC1.45, we hypothesise that in the endogenous regulatory context this enhanced regulatory activity may have driven an increase in *SOX9* expression compared to human as PCCs are forming (Fig. 6).

When considering the implication of increased *SOX9* expression on facial formation, it is important to consider its numerous important regulatory roles (Jo et al., 2014). During facial development, SOX9 is first expressed at the neural plate border region, and in multiple species is required or sufficient for neural crest induction (Cheung and Briscoe, 2003; Schock and LaBonne, 2020; Spokony et al., 2002). It has been proposed that SoxE factors may play a role in maintaining the developmental potential of neural crest cells and promote cell survival (Buitrago-Delgado et al., 2018; Cheung et al., 2005; Sakai et al., 2006). Additionally, SOX9 is required for chondrogenesis and is considered the master regulator of this fate (Akiyama et al., 2002, 2005; Lefebvre and Dvir-Ginzberg, 2017). Indeed, inactivation of Sox9 in neural crest cells leads to a loss of chondrogenic potential, resulting in a switch to an osteoblast fate (Mori-Akiyama et al., 2003), causing jaw dysmorphology. In the context of zebrafish development, *sox9a* knockout causes craniofacial deformities (Yan et al., 2002, 2005), and a decrease in *sox9b* expression decreased chondrocyte cell numbers, resulting in jaw malformation (Xiong et al., 2008; Burns et al., 2015). Interestingly, pro-chondrogenic SOX9-target genes have been shown to be particularly sensitive to SOX9 dosage (Naqvi et al., 2023), so even very subtle changes to SOX9 expression could impact the process of chondrogenesis. We therefore hypothesise that a subtle increase of SOX9 expression in CNCCs and mesenchymal progenitor cells could impact neural crest cell differentiation by impacting cell survival or promoting specification to chondrocytes, thus influencing PCC patterning and cartilage formation, altering the ultimate shape and size of the lower jaw (Naqvi et al., 2023; Pitirri et al., 2022). In support of this, our experiments modulating levels of SOX9 expression in the zebrafish embryo revealed that overexpression of SOX9 in EC1.45-active lineages resulted in an increased size of the PCCs at 2 dpf. Future work will be needed to confirm the impact of EC1.45 Neanderthal variants at the endogenous *SOX9* locus, to quantify changes to gene expression, and to further explore the consequences of increased SOX9 expression on development and, ultimately, jaw morphology (Fig. 6).

Many facial features distinguishing Neanderthals from anatomically modern humans were acquired after the two lineages split from their last common ancestor (Rosas, 2001; Stelzer et al., 2019). These subtle differences in lower jaw morphology between modern human and Neanderthal are hypothesised to have arisen due to alterations to mastication, different mechanical stress responses and general adaptation to environmental conditions (Bastir et al., 2007; Sella-Tunis et al., 2018; Stelzer et al., 2019). Interestingly,

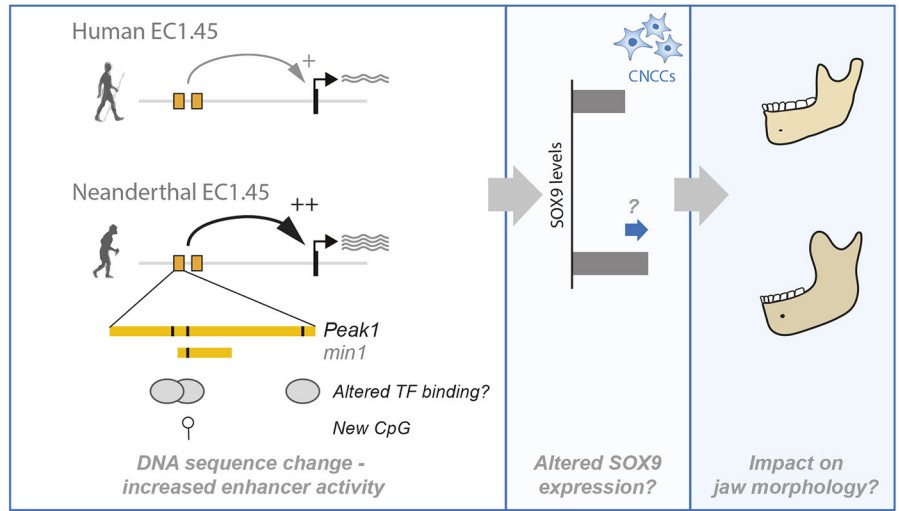

**Fig. 6. Model of increased Neanderthal enhancer activity and the hypothesised impact on jaw development.** A schematic illustrating the hypothesis that increased activity of Neanderthal EC1.45, driven by altered transcription factor binding or by an impact to CpG density and DNA methylation status, may have caused an increased level of SOX9 expression in CNCCs and their mesenchymal derivatives. This alteration may have impacted jaw formation and contributed to Neanderthal-specific alterations in jaw morphology. Neanderthal-derived variants generate a new CpG and may impact transcription factor binding.

although coding sequences are generally highly conserved, genes involved in skeletal morphology have been subject to higher evolutionary change in the lineage leading to the Neanderthals than in the ancestral line common to archaic and modern human (Castellano et al., 2014). Sequence changes in the non-coding genome can drive phenotypic divergence (Frankel et al., 2011; Long et al., 2016; Rubinstein and De Souza, 2013), and have also been implicated in phenotypic differences between modern human and Neanderthal, for example in relation to skull morphology (Funato et al., 2024), and to vocal and facial anatomy (Gokhman et al., 2014, 2020). Here, we provide evidence that three Neanderthal-derived SNVs within the EC1.45 enhancer cluster may have contributed to altered craniofacial form across recent hominin evolution.

For future work, it will be extremely valuable to understand both the mechanisms by which the EC1.45 Neanderthal SNVs result in increased activity, and what the downstream consequences of increased *SOX9* expression are during development. One or more of the Neanderthal SNVs likely alters transcription factor binding. However, functional follow-up is hindered by the challenge of predicting or experimentally identifying the impacted transcription factor (Steinhaus et al., 2022; Corradin and Scacheri, 2014; Claringbould and Zaugg, 2021). By leveraging available empirically defined motifs for transcription factor binding, we identified several candidate TFs where binding is predicted to be impacted by the Neanderthal SNVs (Fig. S10) (Vorontsov et al., 2015). Prioritising for expression in the neural crest cell clusters, TEAD1/3 and SOX9 have a predicted gain of binding to the Neanderthal sequence for SNV1, while XBP1 and CREB3L2 are predicted to gain binding to SNV2. Of interest, Yap/Tead signalling has been shown to play an important role in neural crest migration and development (Wang et al., 2016; Bhattacharya et al., 2020), with mutations in YAP1 associated with orofacial clefting (Williamson et al., 2014). Furthermore, SNV2 generates a new CpG dinucleotide, which is of added interest, as EC1.45 overlaps a differentially methylated region (DMR) in ancient Neanderthal bone samples (Gokhman et al., 2014, 2020; Long et al., 2020). Further work will be required to uncover the mechanism by which the Neanderthal SNVs alter developmental enhancer activity.

Together, we have demonstrated that three Neanderthal-derived SNVs in the EC1.45 enhancer cluster enhance regulatory activity and may have played a role in alteration of Neanderthal specific jaw traits through modulation of *SOX9* expression. Given the association of EC1.45 enhancer loss with human craniofacial dysmorphology, we propose that other sequence variants within this enhancer may also be associated with normal-range variation in human jaw shape (Terhune et al., 2018) or may contribute to risk of jaw malformation upon interaction with other environmental or genetic changes. Indeed, a large number of variants can be identified across the EC1.45 enhancer cluster in the human population from the gnomAD database (Karczewski et al., 2020) and also across other primate species. Together, this work highlights how regulatory function can be impacted by even very small changes to enhancer sequence and proposes that alteration in early jaw progenitors could impact final skeletal form.

### Limitations
Clearly, alteration of EC1.45 activity cannot account for all anatomical differences observed in the Neanderthal jaw, which was likely shaped by multiple genetic changes across multiple loci. Even at the *SOX9* locus there are multiple putative enhancer elements active during CNCC development and chondrogenesis (Yao et al., 2015; Bagheri-Fam et al., 2006; Long et al., 2020; Ichiyama-Kobayashi

et al., 2024) – and there can be complicated interplays between multiple enhancers acting upon a single gene (Hörnblad et al., 2021; Osterwalder et al., 2018; Bothma et al., 2015; Brosh et al., 2023; Blayney et al., 2023). That said, we understand from GWAS that craniofacial shape is a highly polygenic trait with multiple variants contributing small effects to drive morphological variation within the human population (Naqvi et al., 2022). Thus, the Neanderthal-derived SNVs within EC1.45 may contribute to the larger picture of regulatory alterations leading to mandibular morphological divergence in the Neanderthal lineage. Importantly, alteration to enhancer activity for EC1.45 must be understood in the context of the entire regulatory domain, for which the impact of other Neanderthal variants remains to be explored. Nevertheless, future exploration of regulatory grammar within the EC1.45 enhancer cluster will help to uncover the impact of genetic changes on gene regulatory activity, developmental processes and resultant craniofacial morphology, relevant to both normal morphological variation and human disease.

## MATERIALS AND METHODS
### Zebrafish husbandry
Adult zebrafish lines were maintained as per standard protocols (Sprague et al., 2007) and carried out under a UK Home Office licence under the Animals (Scientific Procedures) Act 1986. Zebrafish embryos were raised at 28.5°C. Embryos collected for imaging or FACS were treated with 0.003% 1-phenyl-2-thio-urea (PTU) from 24 h post fertilisation (hpf) to prevent pigment formation.

### Generation of transgenic constructs
Dual enhancer-reporter constructs, based on the Q-STARZ system, were created by Gateway cloning (Invitrogen), as previously described (Bhatia et al., 2021; Uttley et al., 2023). Plasmids QSTARZ_destination and QSTARZ_insulator will be deposited with Addgene. Plasmid pCS2FA CO Tol2 TPase is available at Addgene (plasmid #133032). The human or Neanderthal versions of EC1.45 Peak1-2 were amplified from existing plasmids by PCR using Phusion high fidelity polymerase with Gateway recombination sites included in the primer sequences (coordinates of human enhancer sequences and Neanderthal SNVs are provided in Table S5). Enhancer sequences were then cloned into two pDONR entry vectors, pDONR P4-P1R and pDONR P2R-P3 (Invitrogen), using BP clonase. The insulator sequence was previously cloned into pDONR221 plasmid, containing 2.5 copies of the chicken HS4 sequence (Bhatia et al., 2021). The destination vector was previously synthesised by GeneArt containing a Gateway R4-R3 cassette flanked by PhiC31 and Tol2 recombination sites and minimal promoter-reporter gene units (gata2-eGFP and gata2-mCherry) (Bhatia et al., 2021). The final enhancer reporter plasmids were generated through a multi-way Gateway reaction mediated by LR clonase to combine the two enhancer sequences, the insulator and the destination vector. Three constructs were created: NeP1P2:eGFP;HuP1P2:mCh, HuP1P2:eGFP; NeP1P2:mCh and HuP1P2:eGFP;HuP1P2:mCh (abbreviated to Ne:GFP;Hu: Ch, Hu:GFP;Ne:Ch and Hu:GFP;Hu:Ch).

A single enhancer reporter HuEC1.45-P1P2:eGFP (abbreviated to HuP1P2:GFP) was generated similarly, as previously described (Ravi et al., 2013). The pDONR P4-P1R vector containing human Peak1-2 was combined with a pDONR221 construct containing a gata2-promoter-eGFP-polyA cassette together with a destination vector consisting of a Gateway R4-R2 cassette flanked by Tol2 recombination sites. Plasmid pDest_Tol2-gata2-hP1P2-eGFP has been deposited in Addgene (plasmid #246039).

### Generation of stable transgenic zebrafish lines
Dual and single enhancer-reporter lines were generated as previously described (Bhatia et al., 2021; Uttley et al., 2023). Embryos were collected from wild-type AB zebrafish and injected at the one-cell stage with a 1:1 mixture of Tol2 mRNA and enhancer-reporter construct (final concentration of 25 ng/μl each). Injected F0 embryos were screened at 24 hpf for mosaic eGFP/mCherry expression and raised to adulthood. Adult F0s were crossed with wild-type AB zebrafish and the F1 progeny screened for eGFP/ mCherry expression. Confirmed F0 founders were isolated with wild-type

AB adults and subsequently bred for embryo collection and maintenance of the transgenic line.

### Lineage tracing in transgenic zebrafish lines

A construct to drive Cre expression under the control of the EC1.45-Peak1-2 enhancer cluster and gata2-promoter (*HuEC1.45-P1P2:Cre*) was generated by replacing eGFP from the HuP1P2:eGFP reporter plasmid with Cre amplified from a pEXPGC2(tfap2b:Cre) expression vector (Brombin et al., 2022). *HuEC1.45-P1P2:Cre* was injected at a 1:1 ratio with Tol2 mRNA (final concentration of plasmid 25 ng/μl and mRNA 35 ng/μl) into one-cell stage embryos obtained from crossing the lineage tracing *Tg(ubi:CSY)* line with the *Tg(sox10:mRFP)* reporter line. Embryos were screened at 1 dpf using a Leica M165FCA fluorescence stereomicroscope for an expression switch from AmCyan to ZsYellow in craniofacial regions, resembling the activity patterns for the Tg(*HuP1P2:eGFP*) line.

### Transient overexpression of SOX9 during zebrafish embryogenesis

A construct to drive overexpression of human SOX9 under the control of the EC1.45-Peak1-2 enhancer cluster and gata2-promoter was generated, with T2A-eGFP included to mark injected cells. SOX9 was amplified from SOX9_pLX_TRC317 – TFORF2531 (Addgene plasmid #142954) including a T2A sequence in the primer overhang and sub-cloned into the HuP1P2:eGFP reporter plasmid.

Roughly 2 nl of the resultant plasmid, HuEC1.45-P1P2:hSOX9-T2A-eGFP, was injected into one-cell zebrafish embryos at a 1:1 ratio with Tol2 mRNA (final concentration of plasmid 25 ng/μl, mRNA 35 ng/μl) or HuP1P2:eGFP plus Tol2 mRNA as a control. To titrate down levels of SOX9 overexpression, the two plasmids HuEC1.45-P1P2:hSOX9-T2A-eGFP and HuP1P2:eGFP were mixed at a ratio of 1:1 or 1:3, before being combined with Tol2 mRNA, maintaining the total amount of injected DNA. Embryos were first scored at 1 dpf for phenotypic abnormalities, including: 'mild' phenotypes, such as delayed growth or mild heart oedema; 'moderate' phenotypes, indicating spinal deformities; and 'severe', describing embryos without observable head development. Only embryos classified as 'normal' without any of these phenotypes were included for imaging. Normal embryos were then screened at 1 dpf for eGFP signal using a Leica M165FCA fluorescence stereomicroscope. Detection of signal in the craniofacial structures was a requirement for embryos to be included for confocal imaging and quantification.

### Whole-mount hybridization chain reaction (HCR) RNA-FISH on zebrafish embryos

Whole-mount HCR was performed as previously described (Ibarra-García-Padilla et al., 2021) using the HCR Gold RNA-FISH kit (Molecular Instruments). Briefly, transgenic embryos at 2 dpf were fixed in 4% paraformaldehyde (PFA) overnight at 4°C. Samples were washed in PBST (PBS+0.1% Tween-20), with subsequent dehydration and permeabilization through two consecutive 10 min incubations in 100% methanol, followed by overnight storage in 100% methanol at −20°C. Samples were rehydrated and permeabilized through sequential 5 min incubations in graded methanol solutions (75%, 50% and 25%) prepared in 1× PBS, followed by washing in 1×PBST. Samples were further rehydrated and permeabilized with Proteinase K (10 μg/ml; 23 min) and finally post-fixed in 4% PFA (20 min).

For mRNA detection, samples were washed in 1×PBST, pre-hybridized in HiFi Probe Hybridization Buffer (Molecular Instruments) at 37°C for 30 min, followed by hybridization with HCR HiFi Probes (5 pmol per target) in HiFi Probe Hybridization Buffer at 37°C for 16 h. Probes were combined as appropriate, depending on the specific transgenic line used; for probe sequences, see Table S6. Following hybridization, unbound probes were removed by washing with pre-warmed HCR HiFi Probe Wash Buffer (Molecular Instruments) at 37°C, and 5×saline-sodium citrate buffer containing Tween-20 (SSCT) at room temperature. For HCR signal amplification, 15 pmol fluorophore-conjugated snap-cooled H1 and H2 hairpins were prepared according to the corresponding probe sets (eGFP-488, mCherry-546, *sox9a*-647) and added to the samples in HCR Gold Amplifier Buffer (Molecular Instruments). DAPI was included in the amplification buffer to enable nuclear staining (1 mg/ml). Samples were incubated overnight at room temperature protected from light, then washed

with 5×SSCT and 1×PBST, and finally cleared in a serial glycerol solution (25%, 50% and 70%) prepared in 1×PBST.

### Imaging
#### Live imaging

Prior to confocal imaging, embryos were treated with tricaine (MS-222) anaesthetic and screened for fluorescent signal using a Leica M165FCA or M165FC fluorescence stereomicroscope. Transgenic embryos were then mounted in 1% (w/v) low-melting point (LMP) agarose in E3 in a glass bottom dish.

All images collected for mean fluorescence intensity quantification (Fig. 3B and Fig. S3A,B) were acquired on a Nikon A1R confocal microscope using NIS elements AR software (Nikon Instruments Europe) and a 20× dry objective. A z-stack step size of 1.5 μm was used, and laser power and exposure settings for each channel were the same for all images, optimised to prevent saturation of signal from the brightest samples while still detecting signal from weaker channels.

Images of SOX9 overexpression embryos (Fig. 5B), and some enhancer-reporter embryos not used for quantification (including in Movie 1) were acquired using an Andor Dragonfly (spinning disk) confocal microscope (Andor Technologies) using Andor Fusion acquisition software. These images were captured in 40 μm pinhole mode on the iXon 888 EMCCD camera.

Images were prepared for figures using FIJI (Schindelin et al., 2012) and Imaris (Oxford Instruments). 10/20× dry objectives or 40× water-immersion (WI) objectives were used for acquisition.

#### Time-lapse imaging

Embryos were mounted as described above for live imaging. Part of the LMP agarose surrounding the embryo head and body was then carefully cut away using a microsurgical knife, leaving only the tail of the embryo embedded in agarose, therefore allowing normal embryonic development throughout imaging. Embryos were covered in E3 containing MS-222 and PTU for the duration of the time-lapse imaging and maintained at 28.5°C using an Okolab bold line stage top incubator chamber. Imaging was performed as described on the Andor Dragonfly (spinning disc) confocal, acquiring a z-stack image every 60 min.

#### HCR imaging

Following HCR, fixed and fluorescently labelled embryos were mounted and imaged in 1% (w/v) LMP agarose in a glass bottom dish and imaged using the Andor Dragonfly confocal microscope (Andor Technologies) with a 20× dry objective and a 40× WI objective. For the 20× objective, a z-stack size of 1.0 mm was used; for the 40× WI objective, a z-stack size of 0.5 mm was used. Laser power and exposure settings were kept constant between samples.

### Image analysis
#### Quantification of eGFP and mCherry fluorescence intensity

Enhancer activity was analysed by quantifying the mean fluorescence intensity of eGFP and mCherry in transgenic embryo images captured on the Nikon A1R. A combined eGFP/mCherry channel was created using Imaris coloc, setting the threshold for colocalization at zero. The generated colocalization channel was used to create Imaris surfaces to segment regions of signal, based on the absolute intensity thresholding of the colocalization channel with a defined surface detail of 4 μm. Segmented regions were filtered by size and classified into frontonasal or jaw regions of the face. For each surface, the mean fluorescence intensity of the original eGFP and mCherry signal was then quantified. A Wilcoxon signed-rank test was used to calculate significance $P$ values in R using ggPubr and plot using ggPlot2 (Wickham, 2009).

#### Quantification of mRFP volumes for *Tg(sox10:mRFP)* embryos

The effect of SOX9 or eGFP overexpression was analysed by quantifying the volume of mRFP signal in the jaw region of injected embryos (including Meckel's cartilage, palatoquadrate and ceratohyal regions). Using the Imaris software, a surface encompassing the entire jaw area was manually created. An Imaris machine learning surface creation model was trained on the mRFP signal using anonymized images by a blinded individual. All images were subsequently processed using this model, thus segmenting the region

of mRFP signal in the jaw in an automated manner and allowing unbiased quantification of the resulting volumes. A Bayesian mixed-effects model was used to assess differences in mRFP volume between the eGFP- and SOX9-injected groups, while accounting for variability across replicates performed on different days. The model formula 'Volume~Plasmid+(1+Plasmid | Replicate)' was fit using the brm function from brms (Bürkner, 2017). The random effects structure accounts for baseline differences in volume (intercept) and variation in the effect of plasmid (slope) between replicates. A 95% credible interval from the resulting posterior distribution was used to evaluate the magnitude and direction of any difference between groups. A Wilcoxon signed-rank test was also used to calculate significance *P* values using ggPubr and the data plot using ggPlot2 (Wickham, 2009).

### HCR image analysis
Images (acquired using 40× WI objective) were cropped using FIJI to include regions exhibiting highest fluorescent intensity across all channels, corresponding to the enhancer-active paired-jaw structures of 2 dpf embryos. To map RNA signals to specific cells, InstanSeg, an embedding-based instance segmentation pipeline integrated as a QuPath extension, was applied to accurately identify and segment nuclei and associated cell boundaries in the microscopy images (Bankhead et al., 2017; Goldsborough et al., 2024 preprint). Cells were segmented from two *z*-planes located 10 mm apart, to capture representative planes containing distinct cell populations.

The localization and co-expression of (1) *eGFP-mCherry* and (2) *eGFP-sox9a* RNA signals, detected via HCR, were assessed using QuPath-based cell classification (Bankhead et al., 2017), applying predefined intensity thresholds to individual fluorescence channels. For images from dual-enhancer embryos, the same intensity threshold of 111 was applied to both the eGFP and mCherry channels for consistent classification of positive cells. For images from *Tg(HuEC1.45:eGFP)* a threshold of 113 was set for eGFP and 117 for *sox9a* (reflecting the higher signal intensity). Cell segmentation was checked manually, and partial cells on the edge of the field of view were removed, as well as a small number of incorrectly identified segmented objects (e.g. very small objects consisting of partial cells or very large objects representing multiple cells).

### Dissociation and sorting of 2 dpf embryos
Embryos were dechorionated and placed into 0.5× Danieau's solution (Sprague et al., 2007) with tricaine (MS-222) anaesthetic. Embryo cranial regions were dissected using a scalpel blade and collected into 0.5× Danieau's solution on ice. The samples were washed twice using 0.5× Danieau's solution and once with FACSmax (Amsbio) after centrifugation at 300 *g* for 1 min at 4°C. Finally, samples were resuspended in 500 μl FACSmax and passed through a 35 μm cell strainer to obtain single cells. Cells were sorted based on fluorescence using the CytoFLEX SRT (Beckman Coulter).

### Single cell RNA-sequencing
Up to 40,000 sorted cells per sample were processed for scRNA-seq and Illumina library preparation using the 10× Genomics Chromium single-cell 3′ gene expression technology (v3.1) (Zheng et al., 2017) by the IGC FACS facility. Sequencing of the resulting libraries was carried out on the NextSeq 2000 platform (Illumina) using the NextSeq 1000/2000 P3 Reagents (100 cycles) v3 Kit, aiming for ~50,000 reads per cell. Cell Ranger (10x Genomics, v7.1.0) was used to create FASTQ files and perform alignment, filtering, barcode and UMI counting. A custom reference genome was created from GRCz11 using the Lawson Lab transcriptome annotation (downloaded from https://www.umassmed.edu/lawson-lab/reagents/zebrafish-transcriptome/; Lawson et al., 2020) and manually annotated eGFP and mCherry sequences.

### Analysis of single cell RNA-seq
#### Cell calling and quality control
Cell calling was performed using the emptyDrops function from DropletUtils, excluding mitochondrial and ribosomal genes to improve the filtering of barcodes corresponding to ambient RNA or cell fragments (Lun et al., 2019). QC was then performed using the scater package to remove cells with detected genes <400, library size <1000 and

mitochondrial reads >5% (McCarthy et al., 2017). Further QC was carried out using SoupX to identify and remove cells with high levels of ambient RNA (Young and Behjati, 2020). Cell counts after each stage of filtering are shown in Table S7.

### Clustering and cell type identification
Clustering was performed using Seurat (v5; Hao et al., 2024). Log normalisation, scaling and highly variable gene (HVG) detection was carried out using SCTransform, with regression of mitochondrial expression and cell-cycle stage. Seurat CellCycleScoring was used to assign cell cycle stages. Principle component analysis was performed using the HVG list, which were then used for K-nearest neighbour analysis and Louvain clustering. To identify cell types, Seurat FindAllMarkers was used to identify cluster gene markers and the Daniocell resource (https://daniocell.nichd.nih.gov/index.html; Sur et al., 2023) was used as a reference for cell type mapping using SingleR (Aran et al., 2019). At this stage, a small number of clusters corresponding to contaminating cell types were removed (blood, skin and muscle tissues) to improve clustering resolution for the cell types of interest (see Table S7). Final clustering was carried out using the following parameters – sample *Tg(Ne:GFP;Hu:Ch)* – 10 dimensions, resolution 0.5; sample *Tg(Hu:GFP;Ne:Ch)* – 15 dimensions, resolution 0.2. Plots were generated using Seurat FeaturePlot, DimPlot, DotPlot and VlnPlot functions, using default settings, including scaled expression values for DotPlot.

### Quantification of eGFP and mCherry reads
Cells with at least three eGFP/mCherry reads were assigned as enhancer-active (human, Neanderthal or double positive).

### Identification of transcription factor-binding motifs predicted to be impacted by Neanderthal-derived variants
A DNA sequence surrounding SNV1 and SNV2 was inputted into the Perfectos Ape tool to identify predicted transcription factor binding sites (TFBSs) impacted by the Neanderthal SNVs (Vorontsov et al., 2015). The HOCOMOCO v11, HT-SELEX, JASPAR and SwissRegulon TFBS motif collections were queried, and parameters included pvalue_cutoff=0.0005 and fold_change_cutoff=4.0.

### Acknowledgements
We thank Joanna Wysocka for advice and support at the conception of this project. We are grateful to Cameron Wyatt and the Zebrafish facility at the Institute of Genetics and Cancer (IGC), to Ann Wheeler and the Advanced Imaging Resource at the IGC, to the Flow Cytometry facility at the IGC for their technical support and advice with experiments, and to the Wellcome Trust Clinical Research Facility for sequencing support. We are grateful to Richard White for sharing the *Tg(ubi:LoxP-AmCyan-LoxP-ZsYellow)* zebrafish line. We thank Shipra Bhatia and Wendy Bickmore for advice with the Q-STARZ assay; Laura Murphy and James Iremonger for help with image analysis; Andrew Badrock, Erika Kague and Jana Travnickova for advice with zebrafish embryo phenotyping; Jaaved Mohammed for advice regarding Neanderthal genome sequences; Charli Corcoran for assistance with screening zebrafish founders; Elizabeth Freyer for assistance with FACS analysis; Susan Campbell for Chromium 10X sample preparation; and Hywel Dunn-Davies for support with statistics. We are also grateful to Wendy Bickmore for feedback on the manuscript.

### Competing interests
The authors declare no competing or financial interests.

### Author contributions
Conceptualization: K.U., H.J.J., H.K.L.; Formal analysis: K.U., H.J.J., H.K.L; Funding acquisition: H.K.L.; Investigation: K.U., H.J.J., C.D.A., J.M.T.A., E.O., H.B., H.K.L.; Methodology: K.U., H.J.J., H.K.L.; Supervision: K.U., H.K.L.; Validation: K.U., H.J.J.; Visualization: K.U., H.J.J., E.O., H.K.L.; Writing – original draft: H.K.L.; Writing – review & editing: K.U., H.J.J., H.K.L.

### Funding
K.U. was funded by a Medical Research Council Transition fellowship (WT13290025), H.J.J. was funded by ERASMUS+, E.O. and H.J.J. are funded by a Medical Research Council PhD Studentship (MC_ST_00035) and H.K.L. is supported by a Medical Research Council University Unit grant (MC_UU_00035/12). Open Access funding provided by The University of Edinburgh. Deposited in PMC for immediate release.

## Data and resource availability

The scRNA-sequencing data have been deposited with GEO under accession number GSE298217. Plasmid pDest_Tol2-gata2-hP1P2-eGFP has been deposited in Addgene ( plasmid #246039).

## The people behind the papers

This article has an associated 'The people behind the papers' interview with some of the authors.

## Peer review history

The peer review history is available online at https://journals.biologists.com/dev/lookup/doi/10.1242/dev.204779.reviewer-comments.pdf

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
