## [Peer Review File · Development (Cambridge, England)]

Neanderthal-derived variants increase SOX9 enhancer activity in craniofacial progenitors that shape jaw development

Kirsty Uttley, Hannah J. Jüllig, Carlo De Angelis, Julia M. T. Auer, Ewa Ozga, Hemant Bengani and Hannah K. Long
DOI: 10.1242/dev.204779

Editor: James Briscoe

Review timeline

Submission to Review Commons:	14 November 2024
Submission to Development:	10 March 2025
Editorial Decision:	8 April 2025
First revision received:	9 July 2025
Accepted:	28 July 2025

Reviewer 1:

Evidence, reproducibility and clarity

This is an interesting paper that is logical continuation of authors previous work characterizing a human enhancer mutation implicated in Pierre Robin malformations that alters Sox9 expression. Here using zebrafish as a convenient model organism, the authors test the activity of the human enhancer compared to its Neanderthal ortholog. The results show that both enhancers drive reporter expression in the vicinity of forming cartilage condensations of the jaw. While both enhancers mediate reporter expression in neural crest derived cells, the Neanderthal sequence drives quantitatively higher expression than the orthologous human enhancer. Consistent with this, overexpression of Sox9 using the human enhancer caused an increase in cartilage volume. Altogether, this is a nicely done study that would be appropriate for publication after some revisions as detailed below.

****Major Revisions:****

1. The introduction seems overly long and a bit rambling so diminishes from the excitement of the work. It should be half the length and focus on the novelty of this question and findings.
2. The authors should demonstrate that that human EC1.45 activity overlaps with Sox9 expression. This should be included in Figure 2.
3. There are differences in level of enhancer activity signal between figures (e.g. seems lower in Fig. 3 than Fig. 2). Does enhancer activity vary between embryos or was the imaging protocol different?
4. Some co-staining should be performed to show whether or not the enhancers are active in the same cells but at different levels or if they are actually in different cells.
5. There is an important issue with the single cell RNA seq. Given that the cells were FACS sorted for +GFP and +Cherry, there seem to be many negative cells in their scRNAseq data. Perhaps the FACS gates (figure 4B) were not conservative enough? Did negative cells get included? Authors should verify that their clusters express both GFP and Cherry transcripts.
6. From their scRNAseq data, they talk about enhancer activity in PA1, but this isn't discussed/shown in the enhancer reporter embryos. It would be appropriate to annotate PA1 in figures 2 and 3.
7. Authors should quantify how many Sox9+ cells also have enhancer activity. Looking at the UMAPs in figure 4E and 4F, it actually looks like there is less enhancer activity in the Sox9 dense regions of the clusters.

8. For the over-expression of Sox9 driven by EC1.45, it is important to first establish that EC1.45 activity does indeed overlap with Sox9 gene expression. Does Sox9 itself drive EC1.45?
9. Importantly the authors do not discuss if the Neanderthal SNVs lie in TF binding sites? Which TF motifs? Are they conserved? Are those TF's expressed in the same cells as both enhancers?
10. If you introduce the Neanderthal SNVs into the human sequence, do you gain enhancer activity?
11. The over-expression experiments are tricky as they cause major developmental defects. Would it be possible to drive Sox9 expression at levels that better reflect those driven endogenously by the human versus Neanderthal enhancer?

****Minor Revisions:****

1. Figure 1 - authors should highlight that panel C is a zoom in of panel A.
2. Figure 3 - Why does Human EC1.45 activity looks weaker here than it does in Figure 2.
3. The first sentence of the last paragraph in the Introduction is unclear: "spatiotemporal developmental expression patterns for the human EC1.45 cluster during zebrafish development". Instead should read "reporter expression driven by the human EC1.45 enhancer over developmental time"

Significance

This is a nice paper that advances understanding of jaw development and has disease relevance as well as some evolutionary implications. Thus it is novel and would appeal to developmental biologist, the craniofacial community, and to some extent to evolutionary biologists.

Reviewer 2

Evidence, reproducibility and clarity

The authors provide evidence that nucleotide sequence variants in a remote enhancer, E1.45, which is located 1.45 Mb upstream of the Sox9 promoter, probably contributed to subtle morphological differences in the lower jaws of Neanderthals and modern humans. The study employs the use of a cleverly-designed dual reporter gene for directly comparing the activities of the Neanderthal and modern human enhancers in transgenic zebrafish. The results are clear and convincing: the Neanderthal enhancer is significantly more active than the modern human enhancer.

Here are a few minor recommendations that might help clarify aspects of the study:

1. Is it possible to quantify the different enhancer activities in the zebrafish assays? Is it strictly a question of levels or are there also subtle differences in the timing and/or sites of expression during development?
2. Is the Neanderthal form of the E1.45 enhancer ancestral for the hominids? If so, then reduced expression in modern humans is a derived trait. This could be stated more clearly.
3. Are there potential transcription factor binding motifs associated with the SNVs?

Significance

The authors address one of the most compelling problems in biology: the evolutionary origins of modern humans. This study addresses the role of regulatory DNAs in the divergence of Neanderthals and modern humans. Sox9 is a good focus of study since it has been implicated in the development of craniofacial features in humans. The authors identified three SNVs (single nucleotide variants) in Neanderthal vs. modern human E1.45 enhancer sequences. Direct comparison of these enhancers provide compelling evidence that these SNVs cause upregulation of the Sox9 in Neanderthals. I think this is a very interesting finding and strongly endorse publication.

Reviewer 3

Evidence, reproducibility and clarity

The manuscript by Uttley et al., describes the identification of a candidate sequence for enhancing craniofacial *sox9* expression in Neanderthals and offers functional genomics evidence towards identification of candidate sequence variants in a cis regulatory element (CRE) responsible for jaw morphology variation in hominin evolution.

They generated a transgenic zebrafish model for testing the activity of a previously characterised regulatory element in human, which when mutated causes Pierre Robin developmental disorder and its neanderthal counterpart which has been identified as a candidate enhancer by sequence similarity and by being a DMR in the Neanderthal genome. They show that the Neanderthal CRE is active similarly in distribution to its human counterpart but with elevated activity in anatomically loosely or unspecified cell types in zebrafish cartilaginous neural crest candidates, which they argue are matching the cells where the same enhancer is active in mammalian development.

They then show by single cell transcriptomics the cell distribution for the enhancer activity in relation to neural crest subpopulations and transcription factors involved in craniofacial development.

Finally they carry out overexpression of SOX9 with the human enhancer variant in zebrafish and demonstrate morphology changes which they interpret as evidence towards the capacity of the enhancer to broaden mesenchymal condensations leading to change in jaw morphology.

Taken together, the paper provides evidence for a predicted neanderthal regulatory element candidate to function as enhancer in a zebrafish model and evidence for this enhancer to carry sequence variation which can lead to overactivation in craniofacial cell types relevant to jaw morphology, which the authors interpret as the source of the cis regulatory mechanism for jaw morphology evolution in hominin evolution.

****Main comments:****

I found the conclusion on the functional divergence of sequence variants of Neanderthal v human enhancer convincing as they were provided by an elegant double reporter approach which offers internal control for variant comparison.

However, I found the argument about the role of the sequence variant in craniofacial development less convincing

1. Setting the aims

I found the introduction to the topic and the setting of aims somewhat sketchy. It is not clear from the introduction, why the Neanderthal element was chosen for further study and why the SNVs in this one element were worth pursuing in the lack of broader understanding of the potentially complex regulatory element complexity at the Neanderthal *Sox9* locus. While it is a very reasonable assumption, that a key CRE found and well characterised in human (by the authors in their seminal paper) is a worthy candidate for functional assessment, without better understanding of the overall locus conservation between human and Neanderthal this element may be one of many functionally redundant elements.

2. Justification of the fish model in hominin gene regulation

2.1. For the neanderthal element function to be compared to human in a valuable and informative fashion, one would expect that the host system i.e. the zebrafish is sufficiently conserved by offering a similar developmental context both in terms of gene regulation and in terms of anatomy. From the gene regulation perspective, I would expect that the analysis of the EC1.45 is based on expectation of similar regulatory information content to that in the fish homolog thus one can expect similar TF network activities on them and as a result one can test sequence variation effects relevant to endogenous regulatory interactions both in fish and hominins.

However, there is no data shown for the relevance of fish regulatory background as a test system. No information is provided on the fish *sox9* locus and its activity, or whether the fish homolog enhancer (or any *sox9* enhancer that is expressed in the expected domains of craniofacial lineages and structures) has been identified and how it compares to the hominins. One expects that the hominin enhancers are active in domains of the zebrafish *sox9* for the anatomical structures to give relevant readout. I would expect a comparison and match of the EC1.45 activity to either endogenous *sox9* by WISH or (although less accurate) a cross to one of the several *sox9* reporter transgenic lines available on ZFIN.

2.2. There is an argument about the regulatory networks being conserved (without references), this would need more arguments particularly in the context of Sox9/SOX9 regulation.

3. Further to the justification of the fish model, from the anatomical perspective, the assessment of the parallels of zebrafish and mammalian craniofacial development need strengthening.

3.1. While indeed transparency and external development helps the reporter transgenesis and argues for the fish model, but the generation time is actually comparable to mouse (in contrast to the statement in the introduction), however the understanding of zebrafish craniofacial development and its similarity to human is not well argued, and indeed very superficially compared in the manuscript. I found the anatomical analyses to be rather imprecise and difficult to compare. In the lack of direct comparisons and diagrams comparing mammalian and fish developmental structures and their origins, the statement of 'EC1.45 activity matches expression domains from mammalian development' or 'broadly recapitulate' to be an oversimplification and overstatement. The lineage tracing is an important evidence but again the anatomical homologies need to be more clearly visualized and the lineage history better explained.

3.2. In a similar vein, direct comparison of human and Neanderthal adult morphologies (Figure 1B) would be very helpful.

3.3. I was also confused why the sox10 reporter is used as reference (with no direct overlap of activity to the SOX9 associated EC1.45 reporter) rather than or alongside a sox9a reporter line or even comparison to endogenous sox9a activity by WISH (Figure 2). The anatomical details in Figure 2 would need to be extended with more precisely describing the cell types, where the transgene is active and how the homology to mammalian anatomies are established.

3.4. Overall, the use of the fluorescence reporter is helpful for initial assessments but accurate enhancer activity profiling and comparison should be done by WISH, as mRNA is far more likely to follow the temporal activation dynamics and may explain fluorescence signal intensity differences, the latter important for correct interpretation of sequence variant effects (e.g. is the perceived higher expression by the Ne element is perhaps due to longer expression or earlier activation).

4. Single cell transcriptomics

This experiment was not only used to characterise transgenic reporter active cell types, but to establish transcription factor candidates relevant to neural crest differentiation regulated by EC1.45. What is somewhat confusing, is that the EC1.45 element activity domain is only partially and not predominantly overlapping with the twist1a expressing cells. The authors previously established Twist1 as key regulator of EC1.45 in craniofacial development. How do the authors explain the apparent little relevance of twist1a in regulating the enhancer in fish? Overall the lack of any attempt to link the SNVs to TFBS (including, if available that of the fish homolog sequences) is making the interpretation of the sequence variation harder. BTW, even of the fish elements are not directly identifiable by direct sequence alignment it may be possible to identify the fish homolog through phylogenetic footprinting with stepping stone species such as the non-duplicated paddlefish.

5. Sox9 overexpression

This experiment seems not to add too much to the main claim of the paper. While not essential, for this data to add more value, a comparison to that using the Neanderthal element would be more interesting and not a difficult experiment to carry out.

6. Throughout the paper there is a lack of data on reproducibility of reporter activities. As random integration often leads to position effects, it is expected that more than one lines showing the same patterns is used to identify cell type and tissue specificities. This is lacking in the paper and is a concern, as for example, the human element activity in Fig. 1 appears to be different from that by in the dual reporter shown in Fig. 3.

****Minor points****

A request to the editor as much as the authors: please make sure that legends are on the same

page with figures, it is very hard to follow manuscripts when one needs to scroll between 3 pages at the same time (text, figure, legend). This archaic separation inherited from decades ago when physical prints used to be submitted has no justification in the digital era but continues to make reviewer's life difficult. Similarly, there should be no limit, and it should be encouraged to label anatomical structures directly on panels to point out expression domains, highlight expression variation, or to make a panel more self-explanatory, while making sure that clarity is not lost.

Figure 1A does not support the statement it is referenced to

Figure 1B should include human anatomy in comparison and perhaps a schematic diagram of the hypothesized developmental morphogenesis divergence modelled in this paper

Figure 1D should show why the authors argue the neanderthal is not the ancestral state (BTW, what does the fish homolog look like?)

Figure 4A,B are better suited in Supplemental

Significance

Conceptual: identifying sequence variants in Neanderthal cis-regulatory element as potential source of evolutionary change in morphology.

Technologically mostly following prior art, use of single cell in reporter analysis is technologically improvement on current standards, albeit somewhat rudimentary.

The use of a tractable embryo model to explore a regulatory sequence change leading to morphology change has often been applied for various aspects of evolutionary changes during development pioneering examples include the *shh* ZRA enhancer in fin/limb morphogenesis, or balean fin evolution (PMID: 9860988) or human versus ape hand evolution (PMID: 18772437), but this is the first for applying it to hominin evolution. This will be of interest to human geneticists, evolutionary geneticists and developmental geneticists.

My expertise is in developmental gene regulation with the zebrafish model.

Author response to reviewers' comments

Manuscript number: RC-2024-02782

Corresponding author(s): Hannah, Long

1. General Statements

We thank the reviewers for their constructive critique and suggestions to improve our manuscript, and for highlighting the value of our approach for assessing the regulatory impact of single nucleotide variants on enhancer activity. We were gratified that the reviewers described the study as "addresses a compelling problem", that it is an "interesting" and "novel" study and that it "advances understanding of jaw development" with broad interest to the fields of "development, ... craniofacial ... and ... evolutionary" biology.

We outline below how we plan to address the reviewers' comments through textual changes, additional experiments and analyses in our revised manuscript.

2. Description of the planned revisions

Reviewer #1

This is an interesting paper that is logical continuation of authors previous work

characterizing a human enhancer mutation implicated in Pierre Robin malformations that alters Sox9 expression. Here using zebrafish as a convenient model organism, the authors test the activity of the human enhancer compared to its Neanderthal ortholog. The results show that both enhancers drive reporter expression in the vicinity of forming cartilage condensations of the jaw. While both enhancers mediate reporter expression in neural crest derived cells, the Neanderthal sequence drives quantitatively higher expression than the orthologous human enhancer. Consistent with this, overexpression of Sox9 using the human enhancer caused an increase in cartilage volume. Altogether, this is a nicely done study that would be appropriate for publication after some revisions as detailed below.

We are very pleased to hear the reviewer finds our study interesting, a logical continuation of our previous work and appropriate for publication following the suggested revisions.

Major Revisions:

1. The introduction seems overly long and a bit rambling so diminishes from the excitement of the work. It should be half the length and focus on the novelty of this question and findings.

We thank the reviewer for their suggestion and are happy to edit and shorten the introduction to improve readability and enhance focus on the excitement and novelty of the work.

2. The authors should demonstrate that that human EC1.45 activity overlaps with Sox9 expression. This should be included in Figure 2.

We agree with the reviewer that it is of interest to determine that EC1.45 enhancer activity overlaps with *sox9a/b* expression. Thus far, we assessed this question using scRNA-seq data (Figure 4E-F) and were able to show that the cell type clusters where the enhancers are broadly active (FN CNCC, PA1 CNCC and Proliferative PA1 CNCC) express *sox9a* to a high degree (Rebuttal Figure 1A). We have now also plotted *sox9a* and *sox9b* expression levels in EC1.45- active cells (≥ 1 read detected of either eGFP or mCherry) from the cranial neural crest cell (CNCC) clusters. This further emphasises that the majority, precisely 71%, of EC1.45-active cells express *sox9a* (Rebuttal Figure 1B). The true overlap could in fact be even higher as scRNA-seq data is often sparse due to high levels of read 'drop-out' caused by inefficient mRNA capture, low amounts of starting mRNA in cells, and stochasticity of expression (Kharchenko et al. 2014, PMID: 24836921). See our response to *Rev1 comment 7* below for updates to the manuscript text.

Rebuttal Figure 1. (A) Panels from Figure 4E-F showing widespread activity of EC1.45 (left) and broad expression of *sox9a* in the CNCC clusters (right). (B) Violin plots showing expression of *sox9a* and *sox9b* for cells which have evidence for EC1.45 activity (≥ 1 read of either eGFP

or mCherry) for the 3 CNCC clusters.

We acknowledge that the scRNA-seq does not allow us to assess the spatial overlap of *sox9a* expression with EC1.45 activity. To address this, we plan to leverage immunofluorescence or RNA fluorescence *in situ* hybridization (RNA-FISH) for *sox9* and eGFP to explore the overlap of *sox9* expression and enhancer activity for 2 dpf embryos from *Tg(HuP1P2:GFP)* transgenic line.

3. *There are differences in level of enhancer activity signal between figures (e.g. seems lower in Fig. 3 than Fig. 2). Does enhancer activity vary between embryos or was the imaging protocol different?*

We agree that the difference in signal for the human enhancer between Figures 2 and 3 appears to be confusing. There is indeed a degree of variation in brightness between embryos and lines, as can be seen in the quantification data, but this is not the source of this issue.

Instead, as the reviewer indicates, this is due to differences in the imaging protocol and post-acquisition adjustments.

In Figure 2, we present the expression pattern of human-EC1.45 for the *Tg(HuP1P2:GFP)* line. Therefore, for these images, brightness min/max values were adjusted during image processing in FIJI with an emphasis on visualising the spatial patterns of the human-EC1.45 reporter activity, without the need for absolute quantification of reporter brightness.

In Figure 3, we aimed to quantify absolute reporter brightness for human versus Neanderthal enhancer activity across tissues. Therefore, imaging of all dual-reporter Q-STARZ lines was performed using the same confocal microscope and the same imaging protocol with no adjustments to min/max values post-acquisition. Settings were selected to avoid saturation for the brighter Neanderthal signal while maintaining detection of the weaker human signal. The human signal therefore appears dim by comparison to Figure 2.

To illustrate that the human signal from the *Tg(Ne:GFP;Hu:Ch)*, *Tg(Hu:GFP;Ne:Ch)* and *Tg(Hu:GFP;Hu:Ch)* lines in Figure 3 is equivalent to that seen for the *Tg(HuP1P2:GFP)* line in Figure 2, we have altered the min/max brightness values for the images post-acquisition from Figure 38 (*Rebuttal Figure 2*). We hope this clearly shows the human enhancer activity in these lines matches that observed in Figure 2 and addresses the reviewer's question.

To address this point of confusion in the manuscript, we have added a statement to the text to clarify why the human signal appears dimmer in Figure 3 compared to Figure 2: "Of note, when imaging the Q-STARZ reporter lines, image acquisition settings were optimised to avoid saturation for the brighter Neanderthal signal while maintaining detection of the weaker human signal."

Rebuttal Figure 2. Adjusted minimax brightness values for Figure 3B. Adjustments were performed in FIJI to highlight the equivalent expression patterns for human and Neanderthal EC1.45 elements across all stable dual reporter lines.

4. Some co-staining should be performed to show whether or not the enhancers are active in the same cells but at different levels or if they are actually in different cells.

We agree that whether the differences we detect in enhancer activity levels are due to alterations in absolute expression levels in the same cells, or are due to changes in the cell type or number of cells where the enhancer is active, is an interesting question. From scRNA-seq data from the *Tg(Hu:GFP;Ne:Ch)* line, we observed that the human and Neanderthal enhancer active cells are highly enriched in the same cell type clusters (most prominently in PA1 CNCC and Proliferative PA1 CNCC clusters, Figure 4D), with eGFP and mCherry reads often co-detected in many cells in these clusters (23% of all EC1.45-active cells (≥ 3 total eGFP/mCherry reads) express both eGFP and mCherry). This overlap is likely to be even greater in reality due to the impact of read drop-out in scRNA-seq (see *Rev1 comment 2* above, Kharchenko et al. 2014). With this in mind, we have plotted expression of key neural crest genes and cluster markers for single-positive (eGFP-alone or mCherry-alone) versus double-positive cells. From this analysis, we don't detect any strong evidence of distinct cell type compositions for the human- versus Neanderthal-active cells versus cells where both are active (*Rebuttal figure 3*).

As discussed for *Rev1 comment 2* above, we acknowledge the limitation of scRNA-seq to explore Q-STARZ reporter expression overlap in a spatial manner. Thus far, we have not had high enough resolution imaging data to carefully quantify absolute cell-to-cell expression levels and have instead performed our quantification across larger morphological regions (frontonasal and jaw regions). To explore further whether we can quantify differences in expression levels between or within individual cells and better determine the proportion of co-expressing cells in the embryo, we plan to use either immunofluorescence or RNA-FISH at 2 dpf for eGFP and mCherry.

A limitation we may face here however is the stochasticity observed for enhancer reporter expression, as eGFP and mCherry also do not appear co-expressed for all cells in the *Tg(Hu:GFP;Hu:Ch)* control Q-STARZ reporter line - something we have also

noticed previously for other lines (Uttley et al, 2023). However, we can account for this baseline level of stochasticity of co-reporter expression to set our expectations for the *Tg(Hu:GFP;Ne:Ch)* and *Tg(Ne:GFP;Hu:Ch)* lines.

Rebuttal Figure 3. Violin plots for scRNA-seq CNCC clusters, separated into EC1.45- human positive (eGFP), EC1.45-Neanderthal positive (mCherry), or cells where both are active. Genes plotted include cluster markers and CNCC marker genes.

5. There is an important issue with the single cell RNA seq. Given that the cells were FACS sorted for +GFP and +Cherry, there seem to be many negative cells in their scRNAseq data. Perhaps the FACS gates (figure 4B) were not conservative enough? Did negative cells get included? Authors should verify that their clusters express both GFP and Cherry transcripts.

We believe this is a multi-faceted issue. Firstly, the FACS sorting appears to let through some negative cells without enhancer activity, perhaps due to autofluorescence, or because they are sticking to positive cells. Following trial-sorts for the 10X scRNA-seq from 2 dpf zebrafish embryos, we performed a re-sort to check the purity of the cells sorted for eGFP or mCherry. From this analysis, we observed that only 77.88% of cells were within the original sort gates, i.e. 22.12% of the sorted cells were in fact eGFP- and mCherry-negative (*Rebuttal Figure 4*).

Ultimately, we performed the FACS sorting to enrich for enhancer-active cells, and despite some carry-through of apparently enhancer-negative cells, we can easily identify the enhancer-active cells by their expression of eGFP or mCherry, as is highlighted in Figures 4D-E.

Rebuttal Figure 4. GFP and mCherry-positive cells sorted from dissected cranial regions of *Tg(Hu:GFP;Ne:Ch)* embryos at 2 dpf (left). The sorted population of cells was re-sorted and revealed that a proportion of sorted cells were in fact negative (around 22%, middle). Re-sort of the negative sorted population was indeed 99.99% negative (right).

A second challenge with droplet-based scRNA-seq is read drop-out, as discussed for *Rev1 comment 2 and 4* above (Kharchenko et al. 2014). It is therefore expected that some enhancer-active cells, expressing mCherry or eGFP, may not have any detected sequencing reads for mCherry or eGFP. Taking into account these challenges, the numbers of enhancer-negative cells (with zero reads of eGFP or mCherry) appear reasonable, and in keeping with previous experiments utilising this system (Uttley et al., 2023). Finally, since we dissected zebrafish cranial regions for sorting, the negative cell types (mainly neuronal) identified in the final dataset are logical.

6. From their scRNAseq data, they talk about enhancer activity in PA1, but this isn't discussed/shown in the enhancer reporter embryos. It would be appropriate to annotate PA1 in figures 2 and 3.

We thank the reviewer for this suggestion. We have now annotated pharyngeal arch 1 (PA1) in the legend for Figure 2C and Supplementary Figure 18 and in Figure 4G. For Figure 38 we have indicated the frontonasal and jaw region signals, of which the jaw region is likely derived from PA1 (*Rebuttal Figure 5*).

Figure 2C

PCCs / cartilage template, sox10 ⁺	
p - palatoquadrate] embryonic jaw, PA1
m - Meckel's	
c - ceratohyal] embryonic palate
e - ethmoid plate	
Mesenchymal condensates, sox10 ⁺	
fn - frontonasal	o - oral (PA1)

Supplementary Figure 1B

PCCs / cartilage template, RFP	
p - palatoquadrate] embryonic jaw, PA1
m - Meckel's	
c - ceratohyal] embryonic palate
e - ethmoid plate	
Enhancer reporter, eGFP	
EC1.45-P1P2 activity	
fn - frontonasal region	
j - jaw region	

Figure 3B

Figure 4G

Rebuttal Figure 5. Proposed updates to Figures 2C, 3B, 4G and Supplementary Figure 1B to annotate regions derived from pharyngeal arch 1 (PA1). We further propose to update Supplementary Figure 1B to further highlight enhancer-active regions adjacent to Meckel's cartilage (jaw region) and the frontonasal region.

7. Authors should quantify how many Sox9⁺ cells also have enhancer activity. Looking at the UMAPs in figure 4E and 4F, it actually looks like there is less enhancer activity in the Sox9 dense regions of the clusters.

In the 3 CNCC clusters, 71% of cells expressing eGFP/mCherry also express *sox9a*, and 70% of *sox9a* expressing cells express eGFP/mCherry (based on at least 1 read). As discussed in *Rev1 comments 2, 4 and 5* above, droplet-based single cell RNA-seq suffers from dropouts, and therefore not all cells expressing *sox9a* or eGFP/mCherry will be detected as such.

We can appreciate however that the regions of the CNCC clusters with highest eGFP/mCherry expression are not necessarily the regions with highest *sox9a* expression (Figure 4F versus Figure 4E). *sox9a* is expressed in multiple distinct cell types in the developing face, and we do not expect EC1.45 to be active in all of these, where distinct gene regulatory networks (not activating EC1.45) may control *sox9a* expression. Furthermore, *sox9a* will likely be expressed at different levels across the various detected enhancer-active cell types.

We have updated the text to address this point: "In keeping with endogenous EC1.45 regulating human *SOX9* during development, enrichment of eGFP and mCherry expression in the CNCC clusters was concurrent with ~~greater levels of~~ *sox9a* expression (Figure 4D-E and Supplementary Figure 3D). ~~Indeed, while we wouldn't expect all *sox9a*-expressing cells to be~~

~~EC1.45-positive, we do find that the majority of EC1.45 active cells express *sox9a* (i.e. 71% of all cells expressing eGFP/mCherry also express *sox9a*, Figure 4E-F)."~~

8. For the over-expression of Sox9 driven by EC1.45, it is important to first establish that EC1.45 activity does indeed overlap with Sox9 gene expression. Does Sox9 itself drive EC1.45?

In response to *Rev1 comments 2 and 7* above, from our scRNA-seq data we highlighted that 71% of EC1.45-active cells (marked by eGFP/mCherry expression) also express *sox9a*. This overlap may in fact be higher when we consider the effect of read dropout in scRNA-seq data. We further plan to analyse the overlap of EC1.45 activity with *sox9a/b* expression using immunofluorescence and/or RNA-FISH (see also response to *Rev1 comment 2*).

As for whether SOX9 directly regulates EC1.45 activity, this is an interesting question. From preliminary motif analysis we did not find evidence for SOX9 binding, however putative matches to a SOX9 binding motif (MA0077.2) were detected by FIMO using a relaxed threshold (-- thresh 0.001) setting for the EC1.45 sequence used in the Q-STARZ constructs. This analysis therefore does not strongly support or rule out that SOX9 may regulate EC1.45, and future functional assays would be required to explore whether SOX9 regulates EC1.45 activity.

9. Importantly the authors do not discuss if the Neanderthal SNVs lie in TF binding sites? Which TF motifs? Are they conserved? Are those TFs expressed in the same cells as both enhancers?

To address the impact of Neanderthal SNVs on predicted transcription factor binding sites (TF8Ss), we have leveraged the PERFECTOS-APE tool (Vorontsov et al. 2015, DOI: 10.5220/0005189301020108), which identifies TF8S motifs binding alleles of a single-nucleotide variant with different predicted affinity. We decided to focus on SNV 1 and 2, as SNV3 was only detected in one of the three high quality genomes and may have been polymorphic in the Neanderthal population or represent a sequencing error.

From our preliminary analysis, for both SNV1 and SNV2, a number of transcription factor binding sites were identified that are predicted to be impacted by the sequence change (using the HOCOMOCO v11, HT-SELEX, JASPAR and SwissRegulon TF8S motif collections). While there are limitations associated with this analysis as it relies on consensus TF8S motifs, we believe this is of interest and plan to include the predicted TF8S changes in a new Supplementary Table in the final revised manuscript and add a description of our analysis into the text.

To address whether the transcription factors implicated from the above analysis are expressed in the same cells as the enhancers, we will explore their expression levels in the CNCC clusters from our scRNA-seq data. To address the question of conservation, we will further explore the transcription factor expression levels in recently available human embryonic craniofacial scRNA-seq datasets (Khouri-Farah et al. 2025).

See provided Rebuttal Tables 1-2. Predicted impact to TFBSs of Neanderthal-derived variants in the EC1.45 enhancer using the PERFECTOS-APE tool.

10. If you introduce the Neanderthal SNVs into the human sequence, do you gain enhancer activity?

The human and Neanderthal EC1.45 enhancer sequences only vary by the 3 highlighted SNVs, therefore our experiments have demonstrated that by introducing the 3 Neanderthal SNVs into the human (ancestral) sequence, we increase craniofacial enhancer activity in the zebrafish reporter model specifically at 2 dpf.

11. The over-expression experiments are tricky as they cause major developmental defects. Would it be possible to drive Sox9 expression at levels that better reflect those driven endogenously by the human versus Neanderthal enhancer?

We did observe developmental defects in a subset of the embryos we examined in the overexpression experiments (including for eGFP expression alone, though at lower frequency - see *Supplementary Figure 6A*). In some cases, these developmental defects will be caused by the injection itself, and in other cases will be due to extremely high levels of SOX9 expression that appear to be detrimental during early development. Because we are relying on Tol2-mediated integration for the maintained overexpression of SOX9 or eGFP, the number and location of insertions will have a significant impact on expression, making it challenging to titrate the overexpression purely by adjusting the amount of injected DNA. What we perhaps failed to emphasise well in the text is that only embryos with normal morphology were taken forward for imaging and quantification. We surmise from this that these normal-looking embryos have lower SOX9 overexpression compared to the developmentally abnormal embryos, which should better reflect an endogenous increase in expression caused by the Neanderthal variants.

We will update the manuscript text to emphasise these points: "Only embryos with normal morphology were selected for imaging, ~~and~~ analysis and quantification. Embryos from four replicate experiments with detectable eGFP expression and screened for lack of ~~no~~ overt developmental abnormalities were selected for confocal imaging"

Minor Revisions:

1. Figure 1 - authors should highlight that panel C is a zoom in of panel A.

We propose to update Figure 1 as indicated in *Rebuttal Figure 6*.

Rebuttal Figure 6. Updated Figure 1, panels A and C.

2. Figure 3 - Why does Human EC1.45 activity looks weaker here than it does in Figure 2.

We agree with the reviewer that this is a point of confusion. As outlined in detail for *Rev1 comment 3*, different imaging settings were used for Figure 2 (to show the spatial domains of enhancer activity) versus Figure 3 (to illustrate the differences in absolute activity levels).

We have proposed updates to the manuscript text to emphasize the rationale behind the optimised image acquisition settings used in Figure 3, and have adjusted the min/max brightness values for Figure 38 to demonstrate that the expression patterns for the human enhancer images matches that seen in Figure 2 (see *Rebuttal Figure 2*).

3. The first sentence of the last paragraph in the Introduction is unclear: "spatiotemporal developmental expression patterns for the human EC1.45 cluster during zebrafish development". Instead should read "reporter expression driven by the human EC1.45 enhancer over developmental time"

We have incorporated the suggested edit into the revised manuscript text.

Significance

This is a nice paper that advances understanding of jaw development and has disease relevance as well as some evolutionary implications. Thus it is novel and would appeal to developmental biologist, the craniofacial community, and to some extent to evolutionary biologists.

We thank the reviewer for their comments and are pleased that they highlight the broad interest of our manuscript, its novelty, and the implications of our findings for understanding jaw development and evolution.

Reviewer #2

Evidence, reproducibility and clarity

The authors provide evidence that nucleotide sequence variants in a remote enhancer, E1.45, which is located 1.45 Mb upstream of the Sox9 promoter, probably contributed to subtle morphological differences in the lower jaws of Neanderthals and modern humans. The study employs the use of a cleverly-designed dual reporter gene for directly comparing the activities of the Neanderthal and modern human enhancers in transgenic zebrafish. The results are clear and convincing: the Neanderthal enhancer is significantly more active than the modern human enhancer.

We thank the reviewer for their comments and are pleased that the reviewer finds our work "clear and convincing".

Here are a few minor recommendations that might help clarify aspects of the study:

1. Is it possible to quantify the different enhancer activities in the zebrafish assays? Is it strictly a question of levels or are there also subtle differences in the timing and/or sites of expression during development?

We have begun to explore the question of timing or initiation of enhancer activity by observing when the human and Neanderthal enhancer activity is first detected using the *Tg(Ne:GFP;Hu:Ch)* and *Tg(Hu:GFP;Ne:Ch)* lines. We have determined that eGFP/mCherry expression is first broadly detected in the cranial region at around 23-24 hpf but have not yet observed a striking difference between human and Neanderthal activities. We now propose to perform timelapse imaging from 24 to 48 hpf to identify the earliest enhancer activity specifically in the jaw region for the human versus Neanderthal enhancers. This will help us to understand if the higher activity of the Neanderthal enhancer at 2 dpf is due strictly to the level of expression, or potentially timing as the reviewer suggests.

As for sites of expression during development, our imaging and scRNA-seq indicates that the Neanderthal and human enhancers are active in the same tissues and similar cell types (see also *Rev1 comment 4* and *Rebuttal Figure 3*). To further explore the spatial overlap of human and Neanderthal enhancer activity, we propose to perform immunofluorescence or RNA-FISH for the Q-STARZ reporter lines at 2 dpf.

2. Is the Neanderthal form of the E1.45 enhancer ancestral for the hominids? If so, then reduced expression in modern humans is a derived trait. This could be stated more clearly.

We apologise that this wasn't made clear. We have updated the Figure 1D compressed multiple sequence alignment to include also bonobo, chimpanzee and gorilla, which illustrates that SNV1-3 are only observed in the Neanderthal genome (*Rebuttal Figure 7*). Therefore, these variants appear to be Neanderthal-derived and are predicted to contribute to a gain in enhancer activity in the Neanderthal lineage specifically. For clarity, the bonobo, chimpanzee and gorilla genomes harbour other variants that are not

annotated in the updated figure.

We propose to refer to the updated Figure 1D in the text to support the following statement - "Notably, the three variants do not match the predicted ancestral state and therefore appear to be derived in Neanderthal (Figure 1D)".

D

Neanderthal-derived EC1.45-Peak1 SNVs

	SNV1	SNV2	SNV3
Human (bp)	307	381	941
Human	... ATA GGG TCC ...
Neand	... ACA GCG TTC ...
Bonobo	... ATA GGG TCC ...
Chimp	... ATA GGG TCC ...
Gorilla	... ATA GGG TCC ...

Rebuttal Figure 7. Updated Figure 1, panel D. Compressed multiple sequence alignment depicting three Neanderthal-derived SNVs in the EC1.45 element. SNV3 was detected in only one high quality Neanderthal genome.

3. Are there potential transcription factor binding motifs associated with the SNVs?

We are pleased to be able to include information about the predicted impact of the EC1.45 Neanderthal SNVs on transcription factor binding in our updated manuscript. We indeed find that SNV1 and SNV2 are predicted to impact TFBSs for a number of transcription factors. See our response to *Rev1 comment 9* and *Rebuttal Tables 1-2* for more details.

Significance

The authors address one of the most compelling problems in biology: the evolutionary origins of modern humans. This study addresses the role of regulatory DNAs in the divergence of Neanderthals and modern humans. Sox9 is a good focus of study since it has been implicated in the development of craniofacial features in humans. The authors identified three SNVs (single nucleotide variants) in Neanderthal vs. modern human E1.45 enhancer sequences. Direct comparison of these enhancers provide compelling evidence that these SNVs cause upregulation of the Sox9 in Neanderthals. I think this is a very interesting finding and strongly endorse publication.

We are pleased that the reviewer finds both the topic of the manuscript, and our presented data, compelling. We thank the reviewer for their interest in our findings and endorsing publication of this manuscript.

Reviewer #3 (Evidence, reproducibility and clarity (Required)):

The manuscript by Uttley et al., describes the identification of a candidate sequence for enhancing craniofacial sox9 expression in Neanderthals and offers functional genomics evidence towards identification of candidate sequence variants in a cis regulatory element

They generated a transgenic zebrafish model for testing the activity of a previously characterised regulatory element in human, which when mutated causes Pierre Robin developmental disorder and its neanderthal counterpart which has been identified as a candidate enhancer by sequence similarity and by being a DMR in the Neanderthal genome.

They show that the Neanderthal CRE is active similarly in distribution to its human counterpart but with elevated activity in anatomically loosely or unspecified cell types

in zebrafish cartilaginous neural crest candidates, which they argue are matching the cells where the same enhancer is active in mammalian development.

They then show by single cell transcriptomics the cell distribution for the enhancer activity in relation to neural crest subpopulations and transcription factors involved in craniofacial development.

Finally they carry out overexpression of SOX9 with the human enhancer variant in zebrafish and demonstrate morphology changes which they interpret as evidence towards the capacity of the enhancer to broaden mesenchymal condensations leading to change in jaw morphology.

Taken together, the paper provides evidence for a predicted neanderthal regulatory element candidate to function as enhancer in a zebrafish model and evidence for this enhancer to carry sequence variation which can lead to overactivation in craniofacial cell types relevant to jaw morphology, which the authors interpret as the source of the cis regulatory mechanism for jaw morphology evolution in hominin evolution.

Main comments:

I found the conclusion on the functional divergence of sequence variants of Neanderthal v human enhancer convincing as they were provided by an elegant double reporter approach which offers internal control for variant comparison.

However, i found the argument about the role of the sequence variant in craniofacial development less convincing

We thank the reviewer for their positive summary of our work. We are pleased that the reviewer was convinced by the "elegant" Q-STARZ double enhancer reporter approach.

We wish to clarify, for the avoidance of confusion, that we do not wish to claim that the Neanderthal SNVs in EC1.45 are "*the source of the cis regulatory mechanism for jaw morphology evolution in hominin evolution*". We interpret the increased activity of the Neanderthal EC1.45 as one possible source of genetic divergence which could contribute to evolutionary changes to jaw morphology. We state this in the text - "*Clearly, alteration of EC1.45 activity cannot account for all anatomical differences observed in the Neanderthal jaw, which was likely shaped by multiple genetic changes across multiple loci*" - and in our revised manuscript will endeavor to make this clearer. We hope this clarification that the Neanderthal SNVs in EC1.45 are undoubtedly one piece in a much larger puzzle of genetic changes impacting jaw development will go some way towards assuaging the reviewer's concerns about

We propose to adjust some statements in the manuscript, for example:

In the concluding paragraph of the results section: "We therefore propose that alteration of the EC1.45 enhancer sequence during Neanderthal evolution may have promoted an increase in ~~SOX9 expression~~ **enhancer activity** during a window of craniofacial development that could contribute to altered abundance or morphology of cartilaginous precursors. **Importantly, this model should be considered within the wider context of the locus, and the many additional Neanderthal SNVs across the SOX9 regulatory domain.** Ultimately, while the Neanderthal SNVs in the EC1.45 enhancer cannot alone explain."

And in the concluding paragraph of the discussion: "~~Importantly~~ **Together**, this work ~~implicates alteration in early jaw progenitors in shaping ultimate skeletal form and~~ highlights how regulatory function can be impacted by even very small changes to enhancer sequence, **and proposes that alteration in early jaw progenitors could impact final skeletal form.** Importantly, alteration to enhancer activity at the EC1.45 locus must be understood in the context of the entire regulatory domain, for which the impact of other Neanderthal variants remains to be explored. **Nevertheless, f**uture exploration of regulatory grammar within the EC1.45 enhancer

cluster... "

1. Setting the aims

I found the introduction to the topic and the setting of aims somewhat sketchy. It is not clear from the introduction, why the Neanderthal element was chosen for further study and why the SNVs in this one element were worth pursuing in the lack of broader understanding of the potentially complex regulatory element complexity at the Neanderthal Sox9 locus. While it is a very reasonable assumption, that a key CRE found and well characterised in human (by the authors in their seminal paper) is a worthy candidate for functional assessment, without better understanding of the overall locus conservation between human and Neanderthal this element may be one of many functionally redundant elements.

We agree with the reviewer that numerous alterations to DNA sequence may contribute to changes to SOX9 expression in development between humans and Neanderthals. We therefore identified additional Neanderthal SNVs that overlap with putative CNCC enhancers in the SOX9 landscape, marked by H3K27ac and defined by p300 ChIP-seq peaks. With the caveat that these are only putative enhancers without functional validation, we identify an additional 48 single nucleotide variants between human and Neanderthal at these loci. Of note, these variants were selected to be present in both the Altai and Vindija high coverage genomes. In the future, we hope to explore the impact of these additional sequence variants on enhancer function.

We propose to add additional detail to the text to describe the regulatory complexity at the SOX9 locus, and to clarify further the rationale for focusing on the EC1.45 enhancer.

For example, to describe the regulatory complexity at the SOX9 locus: "**While there are many putative enhancer elements at the SOX9 locus in craniofacial development,** ~~w~~We previously characterised two enhancer clusters that lie upstream of these translocation breakpoints and are ablated by at least one described PRS patient deletion."

For example, to clarify the rationale for focusing on the EC1.45 enhancer: "**Given that ablation of EC1.45 is associated with underdevelopment of the lower jaw in PRS patients,** ~~Here,~~we hypothesised that ~~explored the potential impact of three Neanderthal-derived~~ single nucleotide variants (SNVs) within EC1.45 **may have a more subtle impact on jaw morphology. We thus explored the impact of three Neanderthal-derived SNVs on enhancer function and jaw morphological variation using zebrafish enhancer-reporter lines models.**"

2. Justification of the fish model in hominin gene regulation

2.1. For the neanderthal element function to be compared to human in a valuable and informative fashion, one would expect that the host system i.e. the zebrafish is sufficiently conserved by offering a similar developmental context both in terms of gene regulation and in terms of anatomy. From the gene regulation perspective, i would expect that the analysis of the EC1.45 is based on expectation of similar regulatory information content to that in the fish homolog thus one can expect similar TF network activities on them and as a result one an test sequence variation effects relevant to endogenous regulatory interactions both in fish and hominins. However, there is no data shown for the relevance of fish regulatory background as a test system. No information is provided on the fish sox9 locus and its activity, or whether the fish homolog enhancer (or any sox9 enhancer that is expressed in the expected domains of craniofacial lineages and structures) has been identified and how it compares to the hominins. One expects that the hominin enhancers are active in domains of the zebrafish sox9 for the anatomical structures to give relevant readout. I would expect a comparison and match of the EC1.45 activity to ether endogenous sox9 by WISH or (although less accurate) a cross to one of the several sox9 reporter transgenic lines available on ZFIN.

Despite the incredible diversity of vertebrate craniofacial structures, zebrafish have been extensively used to study craniofacial development and the underlying cell biology and gene regulatory networks controlling craniofacial development are well-conserved

from mammals to fish (Mork and Crump, 2015, PMID: 26589928; Medeiros and Crump, 2012, PMID: 22960284; Fox and Waskiewicz, 2024, PMID: 38385025). *sox9* expression in the zebrafish craniofacial structures has been well-characterised (e.g. Eames et al., 2013, PMID: 23714426), and *sox9a/b* mutants have defects in craniofacial and skeletal development (Yan et al. 2002; 2005). In addition, zebrafish enhancer-reporter assays have been used previously to investigate the activity of putative human *SOX9* enhancers implicated in PRS (Gordon et al, 2014, PMID: 24934569). These observations supported our motivation for using the zebrafish model in this study. We plan to add these additional details and references to our introduction, to highlight the utility of the zebrafish model to study human craniofacial development and disorders.

As suggested by the reviewer, and discussed above (see *Rev1 comment 2*), we intend to explore the overlap between EC1.45 enhancer activity and endogenous *sox9* expression during zebrafish development by immunofluorescence and/or RNA-FISH at 2 dpf. In addition, we have provided further evidence from our scRNA-seq data that EC1.45-active cells robustly express *sox9* (see *Rebuttal Figure 1*).

As for exploring the *sox9a/b* regulatory domains in zebrafish, in our previous work we were not able to identify an orthologous EC1.45 enhancer region in the zebrafish genome at either of the two *sox9* paralogs, *sox9a* or *sox9b* (Long et al, 2020). As suggested below in *Rev3 comment 4*, we have also attempted to identify a zebrafish orthologous EC1.45 element through using an intermediate species. We were able to identify part of a putative orthologous EC1.45 element in the spotted gar genome but have been unable to identify a similar element for the zebrafish.

2.2. There is an argument about the regulatory networks being conserved (without references), this would need more arguments particularly in the context of Sox9/SOX9 regulation.

We are happy to include additional references supporting the conservation of craniofacial gene regulatory networks from human to zebrafish. Please see our response to *Rev3 comment 2.1* above, in which we provide references supporting the conservation of craniofacial gene regulatory networks more broadly. As for *SOX9*, we have previously shown that multiple 'Coordinator' motifs are present in the human EC1.45 enhancer cluster and are functionally important (Long et al, 2020). It has recently been shown that TWIST1 binds to the Coordinator motif, along with ALX1, MSX1 or PRRX1 in a tissue-specific manner (Prescott et al, 2015, PMID: 26365491; Seungsoo et al, 2024, PMID: 38262408). Furthermore, from ChIP-seq datasets we see binding of NR2F1/2 and TFAP2A at the EC1.45 enhancer (Prescott et al, 2015). Each of these factors has a well-established role in zebrafish craniofacial development (*twist1* PMID: 22589745, *alx1* PMID: 23059813, *tfap2a* PMID: 14985255, PMID: 14985255, *msx1* PMID: 16631154, PMID: 22016187, *prrx1* PMID: 15936012, *nr2f1* PMID: 29358039). In addition, we see expression of these genes in the craniofacial clusters of our scRNA-seq dataset (*Rebuttal Figure 8*), and in publicly available datasets (Fabian et al. 2022, PMID: 35013168; Sur et al 2023, PMID: 37995681). Taken together with the observed activity of the human EC1.45 element in zebrafish craniofacial regions by imaging, we feel confident in interpreting that the gene regulatory networks regulating EC1.45 in mammals, are conserved in zebrafish. We plan to include additional references and update the text in the revised manuscript where necessary to support that craniofacial regulatory networks are broadly conserved in the zebrafish.

Rebuttal Figure 8. scRNA-seq expression for CNCC clusters from 2dpf for *sox9a* (left), transcription factors that bind to Coordinator motifs in human (middle) and transcription factors shown to bind the EC1.45 enhancer by ChIP-seq in CNCCs (right).

Furthermore, since the submission of this manuscript, a preprint (Khouri-Farah et al. 2025, doi: <https://doi.org/10.1101/2025.01.18.633396>) has reported gene expression patterns for the developing human face (CS12-20) using single-nucleus RNA-sequencing. We have leveraged this resource to compare gene expression patterns observed for the EC1.45-active cells in our zebrafish model, and those of the developing human face (*Rebuttal Figure 9*). We see expression of genes such as *SHOX*, *BARX1*, *DLX4* and *SMOC1*, in a cluster of cells from the human embryos annotated as mandibular arch1, which strikingly matches expression of the homologous genes in the PA1 CNCC clusters of our zebrafish dataset (*Rebuttal Figure 9A-B*). Furthermore, *PAX7*, *ALX1/4*, *GATA3* and *PITX2*, appear highly expressed in the human maxillary (MxP.surface and MxP2) and lateral nasal process (LNP1/2) clusters, while the zebrafish gene homologs are enriched in the frontonasal CNCC cluster of our zebrafish dataset (*Rebuttal Figure 9A-B*). We believe this further shows the conservation of the gene regulatory networks involved in facial development and intend to include this data in the supplement of our revised manuscript.

Rebuttal Figure 9. Comparison of gene expression patterns for the human and zebrafish developing face. (A) Expression of genes marking zebrafish PA1-like (pink bar) and frontonasal-like (blue bar) craniofacial clusters from scRNA-seq. (B) Expression of homologous human genes from (A) in snRNA-seq dataset of early human facial development (Khouri-Farah et al, 2025).

Figure B, data taken from: Khouri-Farah, Nagham, Emma Wentworth Winchester, Brian M. Schilder, Kelsey Robinson, Sarah W. Curtis, Nathan G. Skene, Elizabeth J. Leslie-Clarkson, and Justin Cotney. 2025. "Gene Expression Patterns of the Developing Human Face at Single Cell Resolution Reveal Cell Type Contributions to Normal Facial Variation and Disease Risk." <https://doi.org/10.1101/2025.01.18.633396>. Plots generated from <https://url.uk.m.mimecastprotect.com/s/JRFnCVJvinNOPIQfZsQzqqq?domain=cotneyshiny.research.chop.edu/> https://cotneyshiny.research.chop.edu/shiny-apps/craniofacial_all_snRNA/

3. Further to the justification of the fish model, from the anatomical perspective, the assessment of the parallels of zebrafish and mammalian craniofacial development need strengthening.

3.1. While indeed transparency and external development helps the reporter transgenesis and argues for the fish model, but the generation time is actually comparable to mouse (in contrast to the statement in the introduction), however the understanding of zebrafish craniofacial development and its similarity to human is not well argued, and indeed very superficially compared in the manuscript. I found the anatomical analyses to be rather imprecise and difficult to compare. In the lack of direct comparisons and diagrams comparing mammalian and fish developmental structures and their origins, the statement of 'EC1.45 activity matches expression domains from mammalian development' or 'broadly recapitulate' to be an oversimplification and overstatement. The lineage tracing is an important evidence but again the anatomical homologies need to be more clearly visualized and the lineage history better explained.

Our previous assessment of EC1.45 activity in mouse was not performed at high

resolution, therefore the anatomical description was somewhat limited, resulting in some comparisons to the activity observed in zebrafish being superficial. The homology between zebrafish and mammalian embryonic jaw anatomy has been extensively reviewed (e.g. Mork and Crump, 2015, PMID: 26589928; Fox and Waskiewicz, 2024, PMID: 38385025). We have focused much of our analyses on Meckel's cartilage as an orthologous embryonic cartilage structure that forms a scaffold for lower jaw formation in zebrafish and mammals (Svandova et al. 2020, PMID: 32984323; Eames et al. 2013, PMID: 23714426; Reeck et al. 2022, PMID: 36278545). We acknowledge that our comparisons could be better argued, and we intend to clarify these points in the text, adding detail where we can and emphasising the limitations of the zebrafish model where necessary. For example, we have removed the comment about rapid generation times from the introduction as indicated by the reviewer. In addition, we intend to add a schematic figure to the supplement to compare zebrafish and mammalian embryonic jaw structures to further improve clarity (see also *Rev3 comment 3.3* and *Rebuttal Figure 11* below).

3.2. In a similar vein, direct comparison of human and Neanderthal adult morphologies (Figure 1B) would be very helpful.

We thank the reviewer for this suggestion and plan to incorporate a schematic of a modern human jaw as a comparison to the Neanderthal jaw schematic in Figure 1B. We agree that this will highlight more clearly distinctions between anatomically modern human and Neanderthal mandibles. A draft proposed update for Figure 1B is illustrated in *Rebuttal Figure 10*.

B

Rebuttal Figure 10. Proposed update to Figure 1B to include comparison of Neanderthal mandible shape to modern human jaw morphology (left).

NOTE: Figure 10 was drawn free-hand inspired by Figure 2 from Rosas A. 2001. "Occurrence of Neanderthal features in mandibles from the Atapuerca-SH site." *American Journal of Physical Anthropology* 114 (1): 74-91. [https://doi.org/10.1002/1096-8644\(200101\)114:1%3C74::AID-AJPA1007%3E3.0.CO;2-U](https://doi.org/10.1002/1096-8644(200101)114:1%3C74::AID-AJPA1007%3E3.0.CO;2-U)

3.3. I was also confused why the sox10 reporter is used as reference (with no direct overlap of activity to the SOX9 associated EC1.45 reporter) rather than or alongside a sox9a reporter line or even comparison to endogenous sox9a activity by WISH (Figure 2). The anatomical details in Figure 2 would need to be extended with more precisely describing the cell types, where the transgene is active and how the homology to mammalian anatomies are established.

We agree that it is interesting and important to explore where the EC1.45 enhancer is active in relation to *sox9a/b* expression. See our response to *Rev1 comment 2* and *Rebuttal Figure 1* where we provide evidence from scRNA-seq data that EC1.45-active cells robustly express *sox9a/b*. In addition, we plan to perform immunofluorescence and/or RNA-FISH for *sox9* and eGFP to compare EC1.45 activity and *sox9* expression in the *Tg(HuP1P2:GFP)* transgenic line.

When exploring where EC1.45 was active during development, we wished to leverage a

lineage marker to landmark craniofacial tissues during development. The *sox10* reporter

Tg(sox10:mRFP) is active in CNCCs early during development, and also in chondrocytes later in development. Leveraging this reporter, we were able to determine that "enhancer activity was detected alongside Meckel's precartilaginous condensations and extended anteriorly along the oral cavity". Based on our revision experiments to perform immunofluorescence or RNA-FISH outlined above, we plan to update the manuscript text to reflect any additional insights we gain into anatomical location of enhancer activity from these experiments. While this will be valuable information, it is difficult to determine cell types from anatomical location alone, and we hope that our scRNA-seq analysis at 2 dpf highlights further the major cell types where EC1.45 is active at this stage (e.g. Figure 4C, E-F).

In response to *Rev3 comment 3.1*, we have provided additional details about the orthology between human and zebrafish jaw formation. We have also included images in *Rebuttal Figure 11A-B* that emphasize the developmental orthology between mandible formation in the two species. While there are clear distinctions in jaw morphology, we use this figure to highlight similarities in the development of Meckel's cartilage - the precursor structure besides which bone of the lower jaw will form.

We propose to extend the schematic of the human adult jaw in Figure 2C to include an embryonic stage based on an image from *Logjes et al, 2018* (see *Rebuttal Figure 11C*).

NOTE: Figure provided for reviewer has been removed. It showed Figure 2 from Logjes, Robrecht J. H., Corstiaan C. Breugem, Gijs Van Haaften, Emma C. Paes, Geoffrey H. Sperber, Marie-Jose H. van den Boogaard, and Peter G. Farlie. 2018. "The Ontogeny of Robin Sequence." *American Journal of Medical Genetics Part A* 176 (6): 1349-68. <https://doi.org/10.1002/ajmg.a.38718>. We have removed unpublished data that had been provided for the referees in confidence.

Rebuttal Figure 11. (A) Morphological comparison of mandible formation in human (Logjes et al, 2018, PMID: 29696787; upper) and (B) zebrafish development (Eames et al, 2013, PMID: 23714426, Facebase; lower). (C) Proposed update to Figure 2C panel inset to highlight orthology of jaw development in zebrafish and human.

NOTE: Figure 11A provided for reviewer has been removed. It showed Figure 2 from Logjes, Robrecht J. H., Corstiaan C. Breugem, Gijs Van Haaften, Emma C. Paes, Geoffrey H. Sperber, Marie-Jose H. van den Boogaard, and Peter G. Farlie. 2018.

“The Ontogeny of Robin Sequence.” *American Journal of Medical Genetics Part A* 176 (6): 1349-68. <https://doi.org/10.1002/ajmg.a.38718>. We have removed unpublished data that had been provided for the referees in confidence.

NOTE: Figure 11B provided was compiled from images on the FaceBase hub from the Fish Face Atlas (Eames et al, 2013). FaceBase 3: analytical tools and FAIR resources for craniofacial and dental research Bridget D. Samuels, Robert Aho, James F. Brinkley, Alejandro Bugacov, Eleanor Feingold, Shannon Fisher, Ana S. Gonzalez-Reiche, Joseph G. Hacia, Benedikt Hallgrímsson, Karissa Hansen, Matthew P. Harris, Thach-Vu Ho, Greg Holmes, Joan E. Hooper, Ethylin Wang Jabs, Kenneth L. Jones, Carl Kesselman, Ophir D. Klein, Elizabeth J. Leslie, Hong Li, Eric C. Liao, Hannah Long, Na Lu, Richard L. Maas, Mary L. Marazita, Jaaved Mohammed, Sara Prescott, Robert Schuler, Licia Selleri, Richard A. Spritz, Tomek Swigut, Harm van Bakel, Axel Visel, Ian Welsh, Cristina Williams, Trevor J. Williams, Joanna Wysocka, Yuan Yuan, Yang Chai. *Development* 2020 Sep 21;147(18):dev191213. doi: <https://url.uk.m.mimecastprotect.com/s/n-ewCYXmSWkjBhGhWixCkTO?domain=doi.org%2F10.1242%2Fdev.191213>.

Eames, B Frank, April DeLaurier, Bonnie Ullmann, Tyler R Huycke, James T Nichols, John Dowd, Marcie McFadden, Mark M Sasaki, and Charles B Kimmel. 2013. “FishFace: Interactive Atlas of Zebrafish Craniofacial Development at Cellular Resolution.” *BMC Developmental Biology* 13 (1): 23. <https://url.uk.m.mimecastprotect.com/s/r42pCZXnSqPXbKi1iBk4HQ?domain=doi.org%2Fhttps://doi.org/10.1186/1471-213X-13-23>

3.4. Overall, the use of the fluorescence reporter is helpful for initial assessments but accurate enhancer activity profiling and comparison should be done by WISH, as mRNA is far more likely to follow the temporal activation dynamics and may explain fluorescence signal intensity differences, the latter important for correct interpretation of sequence variant effects (e.g. is the perceived higher expression by the Ne element is perhaps due to longer expression or earlier activation).

We agree that it will be interesting to explore mRNA expression levels from our reporter embryos, and that this should act as a more direct readout of enhancer activity. We therefore propose to perform RNA-FISH (or hybridisation chain reaction, HCR) to explore mRNA expression profiles for mCherry versus eGFP in our Q-STARZ reporter embryos.

To follow temporal activation dynamics for the EC1.45 enhancer element for the Q-STARZ human-vs-Neanderthal reporter lines during early facial development we propose to perform live timelapse imaging from 24-48 hpf. This will allow us to follow the temporal dynamics of both enhancer reporters in a single live embryo (see also *Rev2 comment 1*), and help determine whether differences in enhancer activity observed at 2 dpf could at least partially be a product of earlier enhancer activation.

4. Single cell transcriptomics

This experiment was not only used to characterise transgenic reporter active cell types, but to establish transcription factor candidates relevant to neural crest differentiation regulated by EC1.45. What is somewhat confusing, is that the EC1.45 element activity domain is only partially and not predominantly overlapping with the twist1a expressing cells. The authors previously established Twist1 as key regulator of EC1.45 in craniofacial development. How do the authors explain the apparent little relevance of twist1a in regulating the enhancer in fish? Overall the lack of any attempt to link the SNVs to TFBS (including, if available that of the fish homolog sequences) is making the interpretation of the sequence variation harder. BTW, even of the fish elements are not directly identifiable by direct sequence alignment it may be possible to identify the fish homolog through phylogenetic footprinting with stepping stone species such as the non-duplicated paddlefish.

In Supplementary Figure 3C, a dot plot shows that *twist1a* is highly expressed in the CNCC

clusters of our scRNA-seq dataset at 2 dpf. We now also show a UMAP plot further illustrating a strong overlap between *twist1a* expression and EC1.45 activity, which would be consistent with the capacity of *twist1a* to regulate EC1.45 in the zebrafish (*Rebuttal Figure 12*).

Rebuttal Figure 12. UMAP of zebrafish CNCC-like clusters displaying broad expression of twist1a (right).

We agree that linking the Neanderthal-derived SNVs with impact to TFBSs is of great interest. We refer to *Rev2 comment 9* for more details for this analysis, and *Rebuttal Tables 1-2*.

We thank the reviewer for their comment regarding the use of the paddlefish for phylogenetic footprinting. Using this strategy, we BLAST-ed the Coelacanth min2 element against the spotted gar genome (available in ENSEMBL BLAST) and were able to identify a short sequence at the *sox9* locus which was longer than any other match in the genome (*Rebuttal Figure 13A-B*) and had two strong Coordinator motifs nearby, a feature of human EC1.45 (*Rebuttal Figure 13C*).

However, we were still unable to identify an orthologous locus in the zebrafish genome. In the future, we hope to further leverage information about other transcription factor binding sites at EC1.45 to help identify a putative orthologous enhancer element in zebrafish.

NOTE: We have removed unpublished data that had been provided for the referees in confidence.

Rebuttal Figure 13. Identification of a short orthologous region of EC1.45 in the spotted gar genome with evidence for Coordinator binding motifs. (A) BLAST hit from Coelacanth EC1.45 sequence to spotted gar genome. (B) ENSEMBL genome browser snapshot of BLAST hit from A (red vertical line). (C) Two strong Coordinator motifs (yellow) near the BLAST hit in the spotted gar genome (blue).

5. Sox9 overexpression

This experiment seems not to add too much to the main claim of the paper. While not essential, for this data to add more value, a comparison to that using the Neanderthal element would be more interesting and not a difficult experiment to carry out.

We considered carrying out the SOX9 overexpression using the Neanderthal enhancer, however, we do not believe our experimental design is amenable to this approach. Our capacity to titrate the assay is limited by the number of Tol2-mediated insertions and we believe we have already saturated the developmental sensitivity to SOX9 overexpression, as we see developmental defects with the human variant at high plasmid concentrations. The aim of this experiment was to determine impact of overexpression of SOX9 in EC1.45 active cells, recapitulating to some degree the effect of the Neanderthal SNVs, and we believe our approach has achieved this. Please also see our response to *Rev1 comment 11*.

6. Throughout the paper there is a lack of data on reproducibility of reporter activities.

As random integration often leads to position effects, it is expected that more than one lines showing the same patterns is used to identify cell type and tissue specificities. This is lacking in the paper and is a concern, as for example, the human element activity in Fig. 1 appears to be different from that by in the dual reporter shown in Fig. 3.

In the manuscript we provided images showing the activity of EC1.45 from 4 independent lines - *Tg(Ne:GFP;Hu:Ch)*, *Tg(Hu:GFP;Ne:Ch)*, and *Tg(Hu:GFP;Hu:Ch)* Q-STARZ double enhancer reporter lines, and the *Tg(HuP1P2:GFP)* single reporter. To demonstrate further reproducibility of EC1.45 reporter activities, we intend to image embryos from additional founders for the *Tg(HuP1P2:GFP)* line, and provide images from the different founders in the supplement.

To address the reviewer's specific comments about differences in reporter signal between Figures 2 and 3, we refer to our response to *Rev1 comment 3*.

Minor points

A request to the editor as much as the authors: please make sure that legends are on the same page with figures, it is very hard to follow manuscripts when one needs to scroll between 3 pages at the same time (text, figure, legend). This archaic separation inherited from decades ago when physical prints used to be submitted has no justification in the digital era but continues to make reviewer's life difficult. Similarly, there should be no limit, and it should be encouraged to label anatomical structures directly on panels to point out expression domains, highlight expression variation, or to make a panel more self explanatory, while making sure that clarity is not lost.

We will bear these comments in mind for future bioRxiv submissions.

Figure 1A does not support the statement it is referenced to

We apologise for this error, the sentence should include a reference to data from our previous study where we explored EC1.45 activity in the developing mouse embryo. We have updated the sentence as follows: "*exhibited activity in the developing mouse craniofacial region and limb bud (Long et al. 2020).*" We have also moved the figure reference in the first sentence of the results section for clarity: "*Given the pathogenic consequences associated with ablation of the EC1.45 enhancer cluster (Figure 1A)*".

Figure 1B should include huma anatomy in comparison and perhaps a schematic diagram of the hypothesized developmental morphogenesis divergence modelled in this paper

We intend to include a human mandible schematic in Figure 1B (see *Rev3 comment 3.2*, and *Rebuttal Figure 10*). We also propose to include a small schematic of a human embryonic jaw morphology into Figure 2C above the adult human jaw image (see *Rev3 comment 3.3* above, and *Rebuttal Figure 11C*). We have a simple schematic outlining that we hypothesise jaw morphogenesis might be impacted by the Neanderthal SNVs in Figure 6.

Figure 1D should show why the authors argue the neanderthal is not the ancestral state (BTW, what does the fish homolog look like)

We have updated Figure 1D in response to this comment to include additional out-group species (bonobo, chimpanzee and gorilla) to demonstrate that the three SNVs appear to be Neanderthal-derived (*Rebuttal Figure 7*). See also *Rev2 comment 2*.

Figure 4A,B are better suited in Supplemental

We are happy to take the reviewer's suggestion and move Figure 4A-B to Supplementary Figure 3.

Reviewer #3 (Significance (Required)):

*Conceptual: identifying sequence variants in Neanderthal cis regulatory element as potential source of evolutionary change in morphology.
Technologically mostly following prior art, use of single cell in reporter analysis is technologically improvement on current standards, albeit somewhat rudimentary
The use of a tractable embryo model to explore a regulatory sequence change leading to morphology change has been often applied for various aspects of evolutionary changes during development (pioneering examples include the shh ZRA enhancer in fin/limb morphogenesis, or balean fin evolution (PMID: 9860988) or human versus ape hand evolution (PMID: 18772437), but this is the first for applying it to hominin evolution. This will be of interest to human geneticists, evolutionary geneticists and developmental geneticists.
My expertise is developmental gene regulation with the zebrafish model.*

We thank the reviewer for their comments and for highlighting the interest of this manuscript for human geneticists, evolutionary geneticists and developmental geneticists, and emphasising the novelty of our work exploring regulatory sequence variation in the context of hominin evolution.

3. Description of the revisions that have already been incorporated in the transferred manuscript

We have already incorporated a number of revisions to the manuscript text in response to the reviewer comments (see our point-by-point responses above). These edits have been marked by track changes in the revised text document.

4. Description of analyses that authors prefer not to carry out

Reviewer 3, comment 5

We have explained in our response to this comment that we do not think overexpression of SOX9 from the Neanderthal-EC1.45 element would show a difference to our existing experiments leveraging the human-EC1.45 enhancer. Given the nature of the experiment using Tol2-mediated integration, there is a difficulty titrating SOX9 overexpression, and our selection of morphologically normal embryos for analysis would likely negate any difference performing this experiment with the Neanderthal enhancer. Ultimately, we use the EC1.45 enhancer to direct tissue-specific overexpression of SOX9 to mimic the impact of the Neanderthal variants driving higher enhancer activity and argue there is unlikely to be a difference if we use the Neanderthal enhancer instead. We note that the reviewer 3 does not find this experiment essential, and hope this explanation is sufficient.

First decision letter

MS ID#: dev.204779

MS Title: Neanderthal-derived variants shape craniofacial enhancer activity at a human disease locus

Authors: Kirsty Uttley; Hannah J. Jüllig; Carlo De Angelis; Julia M. T. Auer; Ewa Ozga; Hemant Bengani; Hannah K. Long

Article Type: Review Commons Transfer

Dear Dr Long,

Thank you for sending your manuscript to Development through Review Commons.

I have now received all the referees' reports on the above manuscript, and have reached a decision. The referees' comments are appended below, or you can access them online: please go to:

As you will see, the referees express considerable interest in your work, but have some significant criticisms and recommend a substantial revision of your manuscript before we can consider publication. I encourage you to implement the revisions suggested by Referee 2. If you are able to revise the manuscript along the lines suggested, which may involve further experiments, I will be happy receive a revised version of the manuscript. Your revised paper will be re-reviewed by one or more of the original referees, and acceptance of your manuscript will depend on your addressing satisfactorily the reviewers' major concerns. Please also note that Development will normally permit only one round of major revision. If it would be helpful, you are welcome to contact us to discuss your revision in greater detail. Please send us a point-by-point response indicating your plans for addressing the referees' comments, and we will look over this and provide further guidance.

Please attend to all of the reviewers' comments and ensure that you clearly highlight all changes made in the revised manuscript. Please avoid using 'Tracked changes' in Word files as these are lost in PDF conversion. I should be grateful if you would also provide a point-by-point response detailing how you have dealt with the points raised by the reviewers in the 'Response to Reviewers' box. If you do not agree with any of their criticisms or suggestions please explain clearly why this is so.

Reviewer 1: COMMENTS ON TEXT

The revisions made by the authors to the previous reviews in Review Commons has improved the paper and addressed the issues that I previously raised. The paper seems like a very good fit for Development.

COMMENTS ON DISPLAY ITEMS

Figures are well done and illustrate the main points of the paper.

Reviewer 2: COMMENTS ON TEXT

I was confused by the submission pdf, which details planned revisions in a revision plan, but it does not describe what has actually has been carried out, which makes the review process frustrating. While this is an improved ms, i still think that the i. detection of reporter mRNA - as was planned by the authors in their revision plan - should have been carried out as a more accurate measure of when and where the transgene enhancer is active in fish embryos and larvae; ii, reproducibility statistics for transgenics to be provided and iii the introduction rewritten more substantially (as was also commented on by reviewer 1) to make it more focussed on highlighting the importance of the problem and the gap of knowledge addressed.

COMMENTS ON DISPLAY ITEMS

Similar to the text, many proposed changes were outlined by the authors in the revision plan, some of them implemented, while others do not appear in the pdf i was able to access, while are shown in rebuttal figures. Perhaps I was not able to access another file, nevertheless this apparent inconsistency between the planned revision and actual revision was puzzling.

First revision

Author response to reviewers' comments

Manuscript number: dev.204779R1 (Review Commons RC-2024-02782)

Corresponding author: Hannah Long

General comments

We thank the reviewers for their constructive comments and recommendations, which we believe have significantly improved the manuscript. We apologise for any confusion arising from the Review Commons review process and the earlier revision plan which did not include a fully revised manuscript. We are now pleased to present a full revision of our manuscript with additional experiments, along with updated text and figures. We have outlined a point-by-point response to all comments below, highlighting textual changes, additional figure panels and analyses that we have incorporated into the revised manuscript.

As stated previously, we were gratified that the reviewers described the study as “addresses a compelling problem”, that it is an “interesting” and “novel” study and that it “advances understanding of jaw development” with broad interest to the fields of “development, ... craniofacial ... and ... evolutionary” biology.

Point-by-point response to reviewers’ comments

Reviewer #1

This is an interesting paper that is logical continuation of authors previous work characterizing a human enhancer mutation implicated in Pierre Robin malformations that alters Sox9 expression. Here using zebrafish as a convenient model organism, the authors test the activity of the human enhancer compared to its Neanderthal ortholog. The results show that both enhancers drive reporter expression in the vicinity of forming cartilage condensations of the jaw. While both enhancers mediate reporter expression in neural crest derived cells, the Neanderthal sequence drives quantitatively higher expression than the orthologous human enhancer. Consistent with this, overexpression of Sox9 using the human enhancer caused an increase in cartilage volume. Altogether, this is a nicely done study that would be appropriate for publication after some revisions as detailed below.

We are very pleased to hear the reviewer finds our study interesting, a logical continuation of our previous work and appropriate for publication following suggested revisions.

Major Revisions:

1. The introduction seems overly long and a bit rambling so diminishes from the excitement of the work. It should be half the length and focus on the novelty of this question and findings.

We thank the reviewer for their suggestion and have edited the introduction to shorten, improve readability and enhance focus on the excitement and novelty of the work. All changes are tracked in the manuscript text.

The introduction is reduced from 1159 to 763 words (including references).

2. The authors should demonstrate that that human EC1.45 activity overlaps with Sox9 expression. This should be included in Figure 2.

From our single cell RNA-sequencing (scRNA-seq) data, EC1.45 activity overlaps well with *sox9a* expression, in keeping with the greater importance of *sox9a* in craniofacial development compared to *sox9b* (Yan et al. 2002; 2005). From our analysis, 71% of EC1.45-active CNCCs also express *sox9a/b* (Revision Figure 1, New Fig. S5F), though this is likely to be an underestimate due to read ‘drop-out’ caused by inefficient mRNA capture (see also response to Reviewer-1 comment 5) (Kharchenko et al. 2014).

Revision Figure 1 (New Fig. S5F). Violin plots showing expression of *sox9a* and *sox9b* for EC1.45 active cells (≥ 1 read of either *eGFP* or *mCherry*) for the 3 CNCC clusters.

We have now further explored the spatial overlap of *sox9a* expression with EC1.45 activity using hybridization chain reaction RNA fluorescent *in situ* hybridisation (HCR RNA-FISH) in the *Tg(HuP1P2:GFP)* transgenic line at 2 dpf, leveraging probe-sets designed to detect *sox9a* and *eGFP* mRNA. This approach demonstrates a clear overlap of EC1.45 enhancer activity with a domain of *sox9a* expression (Revision Figure 2, outline and arrow). Quantification was performed leveraging the InstanSeg tool from QuPath, which revealed that $>95\%$ of EC1.45-active cells express *sox9a* ($n=3$ embryos). This data has been incorporated into updated Fig. 2D-E.

Revision Figure 2 (New Fig. 2D). Hybridisation chain reaction (HCR) for *eGFP* (EC1.45 activity, green) and *sox9a* (yellow), highlighting the overlap of EC1.45 enhancer activity with *sox9a* expression for *Tg(HuP1P2:GFP)* embryos (left). Quantification of overlap of EC1.45 activity (*eGFP*) with *sox9a* expression (right).

The following text has been added to describe this new data: “To confirm that EC1.45 enhancer activity overlaps with endogenous *sox9* gene expression, we performed hybridisation chain reaction RNA fluorescent *in situ* hybridisation (HCR RNA-FISH) for the *Tg(HuP1P2:GFP)* transgenic line, using probes against *eGFP* and *sox9a* mRNA at 2 dpf. We focused on *sox9a* based on a demonstrated greater role in pharyngeal cartilage development compared to *sox9b* (Yan et al. 2005). Enhancer activity, marked by *eGFP* mRNA expression, was observed in both the frontonasal region and a paired region adjacent to Meckel’s cartilage (Fig. 2D), correlating with the expression domains previously observed from *eGFP* protein fluorescence. Quantification of enhancer activity at the jaw-adjacent region showed a high degree of overlap with *sox9a* expression (around 96% of *eGFP*-positive cells also expressed *sox9a*), while enhancer activity was not detected in a nearby domain of *sox9a* expression likely representing condensing cartilage (Fig. 2D-E). This is in concordance with our previous observations *in vitro* that EC1.45 activity is rapidly decommissioned during chondrogenesis (Long et al, 2020). Together, a zebrafish reporter of human EC1.45 regulatory activity matches key expression domains observed from mammalian development, and overlaps with endogenous expression of *sox9a*, revealing a conserved regulatory logic for EC1.45 enhancer activity from fish to man. A zebrafish enhancer reporter line therefore and provides increased enhanced temporal and spatial insights into the developmental activity of this EC1.45 disease-associated regulatory element, especially particularly for a population of cells in proximity to the developing lower jaw.”

3. *There are differences in level of enhancer activity signal between figures (e.g. seems lower in Fig. 3 than Fig. 2). Does enhancer activity vary between embryos or was the imaging protocol different?*

We appreciate that the difference in signal for the human-EC1.45 between Fig. 2 and Fig. 3 is confusing. There is indeed a degree of variation in brightness between embryos and lines, as can be seen in the quantification data, but this is not the source of this issue. Instead, as the reviewer indicates, this is due to differences in the imaging protocol and post-acquisition adjustments.

In Fig. 2, we present the expression pattern of human-EC1.45 for the *Tg(HuP1P2:GFP)* line. For these images, brightness min/max values were adjusted during image processing in FIJI with an emphasis on visualising the spatial patterns of the human-EC1.45 reporter activity, without the need for absolute quantification of reporter brightness.

In Fig. 3B (and Fig. S3A-B), we aimed to quantify absolute reporter brightness for human versus Neanderthal enhancer activity across tissues. Therefore, imaging of all dual-reporter Q-STARZ lines was performed using the same confocal microscope and the same imaging protocol with no adjustments to min/max values post-acquisition. Settings were selected to avoid saturation for the brighter Neanderthal signal while maintaining detection of the weaker human signal. The human signal therefore appears dim by comparison to Fig. 2.

To illustrate that the human signal from the *Tg(Ne:GFP;Hu:Ch)*, *Tg(Hu:GFP;Ne:Ch)* and *Tg(Hu:GFP;Hu:Ch)* lines in Fig. 3 is equivalent to that seen for the *Tg(HuP1P2:GFP)* line in Fig. 2, we have altered the min/max brightness values for the images post-acquisition from Fig. 3B (*Revision Figure 3*). We hope this clearly shows the human enhancer activity in these lines matches that observed in Fig. 2.

To address this point of confusion in the manuscript, we have added a statement to the text to clarify why the human signal appears dimmer in Fig. 3 compared to Fig. 2.

Updated text in Results section: “Of note, when imaging the Q-STARZ reporter lines, image acquisition settings were optimised to avoid saturation for the brighter Neanderthal signal while maintaining detection of the weaker human signal.”

We have also updated the corresponding Methods section to clarify that the imaging settings differ from Fig. 2 and 3.

Updated text in ‘Live imaging’ methods section: “A z-stack step size of 1.5µm was used, and laser power and exposure settings for each channel were the same for all images, optimised to prevent saturation of signal from the brightest samples while still detecting signal from weaker channels.”

Revision Figure 3. Adjusted min/max brightness values for Fig. 3B. Adjustments were performed in FIJI to highlight the equivalent expression patterns for human and Neanderthal EC1.45 elements across all stable dual reporter lines.

4. Some co-staining should be performed to show whether or not the enhancers are active in the same cells but at different levels or if they are actually in different cells.

We agree that whether the differences we detect in enhancer activity levels are due to alterations in absolute expression levels in the same cells or are due to changes in the cell type or number of cells where the enhancer is active, is an interesting question. To address this comment, we have performed hybridization chain reaction (HCR) to explore the spatial overlap of human and Neanderthal enhancer activity at 2 dpf and have incorporated additional scRNA-seq analysis exploring cell types where the human versus Neanderthal EC1.45 enhancer is active.

Hybridisation chain reaction

To explore whether the human and Neanderthal EC1.45 enhancer are active in the same cells and to quantify the proportion of co-expressing cells in the embryo, we performed HCR for the three Q-STARZ reporter lines at 2 dpf, labelling both *eGFP* and *mCherry* transcripts. We observed punctate signal as expected for mRNA molecules in the jaw region, with both *eGFP* and *mCherry* signal detected in broadly the same overlapping region. This co-expression was quantified for single Z-planes at 10 μm intervals, using InstanSeg within QuPath to first segment cells. Positive cells were then classified using fixed fluorescence thresholds for HCR signal. This revealed that for all 3 lines, around 60% of enhancer-active cells had detectable expression of both *eGFP* and *mCherry*. Co-expressing cells tended to be in the core of the enhancer-active domain, with single-positive cells on the periphery. This same pattern held true for the *Tg(Hu:GFP;Hu:Ch)* control reporter line, suggesting that this peripheral single-positive signal is due to stochasticity of enhancer activity at the borders of the active domain, perhaps where key transcription factor levels reach a sub-threshold level for driving robust enhancer activity. These results do not rule out that additional single-positive cells are contributing to the observed differences in enhancer activity, though we don't think this is due to a gain in new cell type specific expression domains based on scRNA-seq results below.

We have added text to the manuscript to describe this new HCR data, and the analysis of enhancer activity overlap:

“HCR RNA-FISH for *eGFP* and *mCherry* in the dual-reporter transgenic lines further demonstrated

that most enhancer-active cells are positive for both human and Neanderthal enhancer activity (Fig. S4). Of note, single-positive cells were detected for all three lines, most commonly at the periphery of the enhancer activity domain, perhaps reflecting stochasticity of enhancer activity in regions where key transcription factor (TF) expression reaches a sub-threshold level for driving robust enhancer activity (Uttley et al. 2023). A skew of detected single-positive *eGFP* or *mCherry* expressing cells for the *Tg(Ne:GFP;Hu:Ch)* or *Tg(Hu:GFP;Ne:Ch)* lines respectively further supports that the Neanderthal enhancer has stronger activity at this time point. ~~We therefore concluded that the~~ Together, three Neanderthal-derived SNVs within EC1.45 cause an increased in enhancer activity in a temporally controlled manner during a specific window of embryonic craniofacial development.”

Revision Figure 4. (Simplified from new Fig. S4). (A) Hybridisation chain reaction (HCR) for *eGFP* and *mCherry* for 3 transgenic lines *Tg(Ne:GFP;Hu:Ch)*, *Tg(Hu:GFP;Ne:Ch)* and *Tg(Hu:GFP;Hu:Ch)* at 2 dpf. (B) Quantification of number of double- or single-positive cells (*eGFP* and/or *mCherry*) at the jaw-adjacent region for dual enhancer reporter transgenic lines ($n=3$ per line).

scRNA-seq analysis

scRNA-seq for the *Tg(Hu:GFP;Ne:Ch)* transgenic reporter line revealed that human and Neanderthal enhancer active cells fall within the same cell clusters, most prominently in the PA1 CNCC and Proliferative PA1 CNCC clusters (Fig. 4B), with 43.4 and 45.9% of cells being double-positive for *eGFP* and *mCherry* (≥ 3 *eGFP* and/or *mCherry* reads). To determine whether the detected single-positive cells represent distinct cell states, we plotted expression of key neural crest genes for single-positive human or Neanderthal enhancer-active versus double-positive cells. Importantly, this analysis doesn't support distinct cell type compositions for the human- versus Neanderthal-active cells (Revision Figure 5). Instead, the detected single-positive cells could be due to enhancer activity stochasticity. Supported by our new HCR results (Revision Figure 4), this may occur more at the periphery of the EC1.45 expression domain where key transcription factor levels may be suboptimal for robust activity.

We have added the following text to the manuscript to describe this plot: “...which appear to broadly group together in the CNCC clusters and express similar marker genes (Fig. 4B-C-D-E and Fig. S5G).”

Revision Figure 5 (New Fig. S5G). Violin plots for scRNA-seq CNCC clusters, separated into EC1.45-human positive (eGFP), EC1.45-Neanderthal positive (mCherry), or double positive cells. Genes plotted include cluster markers and CNCC marker genes.

5. There is an important issue with the single cell RNA seq. Given that the cells were FACS sorted for +GFP and +Cherry, there seem to be many negative cells in their scRNAseq data. Perhaps the FACS gates (figure 4B) were not conservative enough? Did negative cells get included? Authors should verify that their clusters express both GFP and Cherry transcripts.

We believe this is a multi-faceted issue. Firstly, the FACS sorting appears to let through some negative cells without enhancer activity, perhaps due to autofluorescence, or because they are sticking to positive cells. Following trial-sorts for the 10X scRNA-seq from 2 dpf zebrafish embryos, we performed a re-sort to check the purity of the cells sorted for eGFP and/or mCherry. From this analysis, we observed that only 77.88% of cells were within the original sort gates, i.e. 22.12% of the sorted cells were in fact eGFP- and mCherry-negative (Revision Figure 6). Ultimately, we performed the FACS sorting to enrich for enhancer-active cells, and despite some carry-through of apparently enhancer-negative cells, we can easily identify the enhancer-active cells by their expression of eGFP or mCherry, as is highlighted in Fig. 4B-C.

Revision Figure 6. GFP and mCherry-positive cells sorted from dissected cranial regions of Tg(Hu:GFP;Ne:Ch) embryos at 2 dpf (left panel). The sorted population of cells was re-analysed, revealing that a proportion of sorted cells were in fact negative (around 22%, middle panel). Re-analysis of the negative sorted population was indeed 99.99% negative (right panel).

A second challenge with droplet-based scRNA-seq is read drop-out, as mentioned for *Reviewer-1 comment 2* above (Kharchenko et al. 2014). It is therefore expected that some enhancer-active cells, expressing *mCherry* or *eGFP*, may not have any detected sequencing reads for *mCherry* or *eGFP*.

6. From their scRNAseq data, they talk about enhancer activity in PA1, but this isn't discussed/shown in the enhancer reporter embryos. It would be appropriate to annotate PA1 in figures 2 and 3.

We thank the reviewer for this suggestion. We have now annotated pharyngeal arch 1 (PA1) in the legend for Fig. 2Ci and Fig. S1B and in Fig. 4E. For Fig. S1C we have indicated the frontonasal (fn) and jaw (j) region signals on the enhancer reporter embryos, of which the jaw region is likely derived from PA1 (see *Revision Figure 7*).

Fig. 2Ci

PCCs / cartilage template, *sox10+*
 p - palatoquadrate } embryonic
 m - Meckel's } jaw, PA1
 c - ceratohyal
 e - ethmoid plate } embryonic palate
 Mesenchymal condensations, *sox10+*
 fn - frontonasal o - oral (PA1)

Fig. 4E

Fig. S1B

PCCs / cartilage template, RFP
 p - palatoquadrate } embryonic
 m - Meckel's } jaw, PA1
 c - ceratohyal
 e - ethmoid plate } embryonic palate
 Enhancer reporter, eGFP
 EC1.45-P1P2 activity
 j - jaw region
 fn - frontonasal region

Fig. S1C

Revision Figure 7. Updated panels from Fig. 2Ci and Fig. 4E, and Fig. S1B-C to annotate regions derived from pharyngeal arch 1 (PA1).

7. Authors should quantify how many Sox9+ cells also have enhancer activity. Looking at the UMAPs in figure 4E and 4F, it actually looks like there is less enhancer activity in the Sox9 dense regions of the clusters.

In the 3 CNCC clusters, 71% of cells expressing *eGFP/mCherry* also express *sox9a*, and 70% of *sox9a* expressing cells express *eGFP/mCherry* (based on at least 1 read). As discussed in *Reviewer-1 comments 2 and 5* above, droplet-based single cell RNA-seq suffers from dropouts, and therefore not all cells expressing *sox9a* or *eGFP/mCherry* will be detected as such.

We appreciate however that the regions of the CNCC clusters with highest *eGFP/mCherry* expression are not necessarily the regions with highest *sox9a* expression - see Fig. 4D (previously 4F)

versus Fig. 4C (previously 4E). This may reflect different expression levels of *sox9a* across the various detected enhancer-active cell types, or stochasticity for detection of *sox9a* reads across single cells in the cluster. Furthermore, we expect EC1.45 to be decommissioned in chondrocytes, which we believe arise from the EC1.45-active population, and will further upregulate *sox9* expression during chondrogenesis (Fig. S7-8). Therefore, in these cells eGFP/mCherry protein may remain stable while the mRNA is more rapidly turned over. Our HCR data from *Tg(HuP1P2:GFP)* reporter embryos at 2 dpf support that enhancer-active cells reside within a wider domain of *sox9a* expression, where enhancer-active cells exhibit noticeably lower *sox9a* expression compared to the adjacent condensing pre-cartilaginous cells (see *Revision Figure 2*).

We have updated the text to address this point: “In keeping with endogenous EC1.45 regulating human SOX9 during development, enrichment of eGFP and mCherry expression in the CNCC clusters was concurrent with ~~greater levels of~~ *sox9a* expression (Fig. 4B-CD-E and Fig. S5E3D). ~~Indeed, while we wouldn't expect all *sox9a*-expressing cells to be EC1.45-positive, we do find that the majority of EC1.45 active cells express *sox9a* (i.e. 71% of all cells expressing eGFP/mCherry also express *sox9a*, Fig. 4C-D). This is in keeping with our earlier observations from HCR RNA-FISH, where the EC1.45- active domain lies within a wider domain of *sox9a* expression (Fig. 2D-E).”~~

8. For the over-expression of Sox9 driven by EC1.45, it is important to first establish that EC1.45 activity does indeed overlap with Sox9 gene expression. Does Sox9 itself drive EC1.45?

From our scRNA-seq data, we highlighted that 71% of EC1.45-active cells (marked by eGFP/mCherry expression) also express *sox9a*. This overlap may in fact be higher when we consider the effect of read dropout in scRNA-seq data. We have now also performed HCR RNA-FISH to compare EC1.45 activity with *sox9a* expression, demonstrating that >95% of enhancer-active cells express *sox9a* (Fig. 2E; see also response to *Reviewer-1 comment 2*).

As for whether SOX9 directly regulates EC1.45 activity, this is an interesting question. From our differential motif analysis (new Fig. S10 and Tables S6-7) we identified that SNV1 increases the strength of a putative SOX9 binding site for Neanderthal-EC1.45. Future functional assays will be required to explore whether SOX9 regulates EC1.45 activity, and whether this is greater for the Neanderthal enhancer compared to human. See updated text for *Reviewer-1 comment 9*.

9. Importantly the authors do not discuss if the Neanderthal SNVs lie in TF binding sites? Which TF motifs? Are they conserved? Are those TFs expressed in the same cells as both enhancers?

To address the impact of the Neanderthal-derived SNVs on predicted transcription factor binding sites (TFBSs), we have leveraged the PERFECTOS-APE tool (Vorontsov et al. 2015), which identifies TFBS motifs that have an altered predicted affinity due to a single-nucleotide variant. We focused on SNV 1 and 2, as SNV3 was only detected in one of the three high quality Neanderthal genomes and may have been polymorphic in the Neanderthal population or represent a sequencing error.

From our analysis of both SNV1 and SNV2, several TFBSs were identified and predicted to be impacted by the SNV1/2 sequence changes using the HOCOMOCO v11, HT-SELEX, JASPAR and SwissRegulon TFBS motif collections. We have included the predicted TFBS changes in a new Tables S6-7. A subset of these TFs were expressed in the CNCC clusters from our dual reporter zebrafish 2 dpf scRNA-seq data (see new Fig. S10). This analysis suggests that Tead transcription factor family (TEAD1 and TEAD3), SOX9, JUN, XBP1 and CREB3L2 are candidate TFs for differential binding and regulating differential EC1.45 between human and Neanderthal. To examine whether these TFs may also be expressed in human craniofacial development, we further explored transcription factor expression levels from recently available human embryonic craniofacial scRNA-seq datasets (Khouri- Farah et al. 2025). These candidate TFs were also found to be expressed within relevant cell clusters, including human mandibular, maxillary and nasal prominence (see *Revision Figure 8* and Fig. S10). We have included this analysis in the manuscript, highlighting interesting TFs for future follow-up.

Revision Figure 8 (New Fig. S10). Plotting transcription factor binding sites overlapping SNV1 and 2 in the EC1.45 enhancer predicted to be impacted by the Neanderthal-derived variants. See also new Tables S6-7.

Data for these figures was taken from: Khouri-Farah, Nagham, Emma Wentworth Winchester, Brian M. Schilder, Kelsey Robinson, Sarah W. Curtis, Nathan G. Skene, Elizabeth J. Leslie-Clarkson, and Justin Cotney. 2025. “Gene Expression Patterns of the Developing Human Face at Single Cell Resolution Reveal Cell Type Contributions to Normal Facial Variation and Disease Risk.” <https://doi.org/10.1101/2025.01.18.633396>. Plots generated from <https://url.uk.m.mimecastprotect.com/s/0bxsCPJEi105yczfMixP-MK?domain=cotneyshiny.research.chop.edu/> https://cotneyshiny.research.chop.edu/shiny-apps/craniofacial_all_snRNA/

We have added a description of our analysis in the results: “Finally, we hypothesised that increased EC1.45 activity driven by the three Neanderthal-derived SNVs may be due to altered TF binding and therefore performed differential motif calling for SNV1-2 which were detected in all three Neanderthal genomes. This analysis revealed several candidate TFs, a subset of which were expressed in the zebrafish EC1.45-active cell populations at 2 dpf and also in human embryonic facial cell types (Tables S6-7 and Fig. S10) (Khouri-Farah et al. 2025). These TFs, which include TEAD1/3, JUN, XBP1, CREB3L2 and SOX9 itself, represent excellent candidates for future study, in addition to exploration of the new CpG dinucleotide generated by SNV2.”

We have further discussed the candidate TFs in the discussion: “Leveraging available tools

~~include databases of empirically- defined motifs for transcription factor binding site preference, we which can help to identify several candidate factors~~TFs where binding is predicted to be impacted by the Neanderthal SNVs (Fig. S10) ~~nucleotide sequence changes~~ (Hume et al. 2015; Vorontsov et al. 2015; ~~Steinhaus, Robinson, and Seelow 2022~~). Prioritising for expression in the neural crest cell clusters, TEAD1/3 and SOX9 have a predicted gain of binding to the Neanderthal sequence for SNV1, while XBP1 and CREB3L2 are predicted to gain binding to SNV2. Of interest, Yap/Tead signaling has been shown to play an important role in neural crest migration and development (Wang et al. 2016; Bhattacharya, Azambuja, and Simoes-Costa 2020), with mutations in YAP1 associated with orofacial clefting (Williamson et al. 2014).”

10. If you introduce the Neanderthal SNVs into the human sequence, do you gain enhancer activity?

The human and Neanderthal EC1.45 enhancer sequences only vary by the 3 highlighted SNVs, therefore our experiments have demonstrated that by introducing the 3 Neanderthal SNVs into the human (ancestral) sequence, we increase craniofacial enhancer activity in the zebrafish reporter model specifically at 2 dpf.

11. The over-expression experiments are tricky as they cause major developmental defects. Would it be possible to drive Sox9 expression at levels that better reflect those driven endogenously by the human versus Neanderthal enhancer?

We did observe developmental defects in a subset of the embryos we examined in the overexpression experiments (including for eGFP expression alone, though at lower frequency - see Fig. S9A). In some cases, these developmental defects will be caused by the injection itself, and in other cases will be due to extremely high levels of SOX9 expression that appears to be detrimental during early development. Because we are relying on Tol2-mediated integration for the maintained overexpression of SOX9 or eGFP, the number and location of insertions will have a significant impact on expression, making it challenging to titrate the overexpression purely by adjusting the amount of injected DNA. What we perhaps failed to emphasise in the text is that only embryos with normal morphology were taken forward for imaging and quantification. We surmise from this that these normal-looking embryos have lower SOX9 overexpression compared to the developmentally abnormal embryos, which should better model an endogenous increase in expression caused by the Neanderthal variants.

We have updated the manuscript text to emphasise these points.

Updated Results text: “Only embryos with normal morphology were selected for imaging, ~~and~~ analysis and quantification ... Embryos from four replicate experiments with detectable eGFP expression and ~~screened for lack of~~ overt developmental abnormalities were selected for confocal imaging”.

We have also updated Fig. S9A to emphasize this point further.

Revision Figure 9 (updated Fig. S9A). An additional arrow and box emphasise that only normal appearing embryos were taken forward for imaging and quantification to determine the impact of SOX9 overexpression on the development of precartilaginous condensations in the jaw region.

Minor Revisions:

1. Figure 1 - authors should highlight that panel C is a zoom in of panel A.

We have updated Fig. 1 as indicated in Revision Figure 10.

Revision Figure 10. Updated Fig. 1, panels A and C.

2. Figure 3 - Why does Human EC1.45 activity looks weaker here than it does in Figure 2.

We agree with the reviewer that this is a point of confusion. As outlined in detail for *Reviewer-1 comment 3*, different imaging settings were used for Fig. 2 (to show the spatial domains of human EC1.45 enhancer activity) versus Fig. 3 (to illustrate the differences in absolute activity levels).

We have updated the manuscript text to emphasize the rationale behind the optimised image acquisition settings used in Fig. 3 and have adjusted the min/max brightness values for Fig. 3B to demonstrate that the expression patterns for the human enhancer images matches that seen in Fig. 2 (see *Revision Figure 3*).

3. The first sentence of the last paragraph in the Introduction is unclear: "spatiotemporal developmental expression patterns for the human EC1.45 cluster during zebrafish development". Instead should read "reporter expression driven by the human EC1.45 enhancer over developmental time"

We have incorporated the suggested edit into the revised manuscript text.

Significance

This is a nice paper that advances understanding of jaw development and has disease relevance as well as some evolutionary implications. Thus it is novel and would appeal to developmental biologist, the craniofacial community, and to some extent to evolutionary biologists.

We thank the reviewer for their comments and are pleased that they highlight the broad interest of our manuscript, its novelty, and the implications of our findings for understanding jaw development and evolution.

Reviewer #2

Evidence, reproducibility and clarity

The authors provide evidence that nucleotide sequence variants in a remote enhancer, E1.45, which is located 1.45 Mb upstream of the *Sox9* promoter, probably contributed to subtle morphological differences in the lower jaws of Neanderthals and modern humans. The study employs the use of a cleverly-designed dual reporter gene for directly comparing the activities of the Neanderthal and modern human enhancers in transgenic zebrafish. The results are clear and convincing: the Neanderthal enhancer is significantly more active than the modern human enhancer.

We thank the reviewer for their comments and are pleased that the reviewer finds our work “clear and convincing”.

Here are a few minor recommendations that might help clarify aspects of the study:

1. Is it possible to quantify the different enhancer activities in the zebrafish assays? Is it strictly a question of levels or are there also subtle differences in the timing and/or sites of expression during development?

We have addressed the two aspects of this question with additional experiments.

Enhancer activity timing

To explore if differences in enhancer activity at 2 dpf could be due to distinct developmental onset of human and Neanderthal enhancer activity, we have performed time-lapse imaging for the *Tg(Ne:GFP;Hu:Ch)* and *Tg(Hu:GFP;Ne:Ch)* transgenic lines from 1-2 dpf (see *Movies 6-7*). At 1 dpf eGFP and mCherry expression is seen broadly across the cranial region, and during these time-lapse movies we can track emergence of enhancer activity in the jaw-adjacent region. For *Tg(Ne:GFP;Hu:Ch)*, eGFP (Neanderthal-EC1.45) in the jaw region is first seen at 38 hpf, while mCherry (human-EC1.45) signal is detected from 40 hpf. For *Tg(Hu:GFP;Ne:Ch)*, eGFP (human-EC1.45) can be seen from 39 hpf, with mCherry (Neanderthal-EC1.45) detected from 40 hpf. The slightly earlier detection of eGFP in both lines indicates differences in fluorophore biophysical properties drive this difference in enhancer timing, rather than robust alterations to onset of enhancer activity between the human and Neanderthal sequences. The higher activity of the Neanderthal EC1.45 enhancer at 2 dpf therefore appears not to be due to an earlier initiation of the Neanderthal enhancer activity.

We have added the following text to the manuscript to describe this result: “To explore further the differences in enhancer activity observed at 2 dpf, we performed timelapse imaging from 1-2 dpf which revealed no apparent differences in the onset or spatial localisation of enhancer activity in the jaw region (*Movies 6 and 7*).”

Sites of spatial enhancer activity during development

Our scRNA-seq indicates that the Neanderthal and human enhancers are active in the same tissues and similar cell types (*Revision Figure 5*). To further explore the spatial overlap of human and Neanderthal enhancer activity, we have now performed HCR for the dual enhancer reporter lines at 2 dpf and observe that the human and Neanderthal enhancers are active in the same domains of the embryo (*Revision Figure 4*), with the two fluorescent reporters co-expressed in the majority of cells. We therefore conclude that the differences in enhancer activity are likely due to changes in activity in individual cells (see also response to *Reviewer-1, comment 4*).

Updated text in the results section: “We next explored further whether differences in enhancer activity may be due to distinct cell type expression patterns. However, ~~In accordance with our imaging data,~~ we did not observe any differences in the cell type identity of human versus Neanderthal active cells ~~from our scRNA-seq data,~~ which appear to broadly group together in the CNCC clusters ~~and express similar marker genes~~ (Fig. 4B-CD-E and Fig. S5G). Similar results were observed from the smaller sample of cells obtained from the *Tg(Ne:GFP;Hu:Ch)* line, where 407 cells were grouped into three clusters of PA1 CNCCs, FN CNCCs, and neuronal cells (Fig. S6A-B3E-

F), where the eGFP and mCherry expression was again most enriched in the PA1 CNCC cluster (Fig. S6C3G). These observations were in accordance with our earlier HCR RNA-FISH analysis of mCherry and eGFP transcripts which appeared to broadly overlap spatially at 2 dpf, especially in the core of the enhancer activity domain (Fig. S4)."

Updated text in the discussion section: "From time-lapse imaging, scRNA-seq and spatial assessment of enhancer activity domains, the human and Neanderthal enhancers were active in broadly overlapping cell-types and spatial domains, suggesting for the most-part that activity differences were driven by increased Neanderthal EC1.45 activity in individual cells."

2. Is the Neanderthal form of the E1.45 enhancer ancestral for the hominids? If so, then reduced expression in modern humans is a derived trait. This could be stated more clearly.

We apologise that this wasn't made clear. We have updated the Fig. 1D compressed multiple sequence alignment to include also bonobo, chimpanzee and gorilla, which illustrates that SNVs 1-3 are only observed in the Neanderthal genome (see *Revision Figure 11*). Therefore, the three variants appear to be Neanderthal-derived and are predicted from our work to contribute to a gain in enhancer activity in the Neanderthal lineage specifically. For clarity, the bonobo, chimpanzee and gorilla genomes harbour other variants that are not annotated in the updated figure.

We have added a reference to the updated Fig. 1D in the text to support the following statement: "Notably, the three variants do not match the predicted ancestral state and therefore appear to be derived in Neanderthal (Fig. 1D)".

Neanderthal-derived EC1.45-Peak1 SNVs

	SNV1	SNV2	SNV3
Human (bp)	307	381	941
Human	... ATA ...	GGG ...	TCC ...
Neand	... ACA ...	GCG ...	TTC ...
Bonobo	... ATA ...	GGG ...	TCC ...
Chimp	... ATA ...	GGG ...	TCC ...
Gorilla	... ATA ...	GGG ...	TCC ...

Revision Figure 11. Updated Fig. 1D. Compressed multiple sequence alignment depicting three Neanderthal-derived SNVs in the EC1.45 element. SNV3 was detected in only one high quality Neanderthal genome.

3. Are there potential transcription factor binding motifs associated with the SNVs?

We are pleased to be able to include information about the predicted impact of the EC1.45 Neanderthal SNVs on transcription factor binding in our updated manuscript. We indeed find that SNV1 and SNV2 are predicted to impact transcription factor binding sites for a number of transcription factors. See our response to *Reviewer-1 comment 9* and new Tables S6-7 and Fig. S10 for more details.

Significance

The authors address one of the most compelling problems in biology: the evolutionary origins of modern humans. This study addresses the role of regulatory DNAs in the divergence of Neanderthals and modern humans. Sox9 is a good focus of study since it has been implicated in the development of craniofacial features in humans. The authors identified three SNVs (single nucleotide variants) in Neanderthal vs. modern human E1.45 enhancer sequences. Direct comparison of these enhancers provide compelling evidence that these SNVs cause upregulation of the Sox9 in Neanderthals. I think this is a very interesting finding and strongly endorse publication.

We are pleased that the reviewer finds both the topic of the manuscript, and our presented data, compelling. We thank the reviewer for their interest in our findings and endorsing publication of this manuscript.

Reviewer #3 (Evidence, reproducibility and clarity (Required)):

The manuscript by Uttley et al., describes the identification of a candidate sequence for enhancing craniofacial sox9 expression in Neanderthals and offers functional genomics evidence towards identification of candidate sequence variants in a cis regulatory element (CRE) responsible for jaw morphology variation in hominin evolution.

They generated a transgenic zebrafish model for testing the activity of a previously characterised regulatory element in human, which when mutated causes Pierre Robin developmental disorder and its neanderthal counterpart which has been identified as a candidate enhancer by sequence similarity and by being a DMR in the Neanderthal genome.

They show that the Neanderthal CRE is active similarly in distribution to its human counterpart but with elevated activity in anatomically loosely or unspecified cell types in zebrafish cartilaginous neural crest candidates, which they argue are matching the cells where the same enhancer is active in mammalian development.

They then show by single cell transcriptomics the cell distribution for the enhancer activity in relation to neural crest subpopulations and transcription factors involved in craniofacial development.

Finally they carry out overexpression of SOX9 with the human enhancer variant in zebrafish and demonstrate morphology changes which they interpret as evidence towards the capacity of the enhancer to broaden mesenchymal condensations leading to change in jaw morphology.

Taken together, the paper provides evidence for a predicted neanderthal regulatory element candidate to function as enhancer in a zebrafish model and evidence for this enhancer to carry sequence variation which can lead to overactivation in craniofacial cell types relevant to jaw morphology, which the authors interpret as the source of the cis regulatory mechanism for jaw morphology evolution in hominin evolution.

Main comments:

I found the conclusion on the functional divergence of sequence variants of Neanderthal v human enhancer convincing as they were provided by an elegant double reporter approach which offers internal control for variant comparison.

However, i found the argument about the role of the sequence variant in craniofacial development less convincing

We thank the reviewer for their positive summary of our work. We are pleased that the reviewer was convinced by the “elegant” Q-STARZ dual enhancer reporter approach.

We wish to clarify, for the avoidance of confusion, that we do not wish to claim that the Neanderthal SNVs in EC1.45 are “the source of the cis regulatory mechanism for jaw morphology evolution in hominin evolution”. We interpret the increased activity of the Neanderthal EC1.45 as one possible source of genetic divergence which could contribute to evolutionary changes to jaw morphology. We state this in the text - “Clearly, alteration of EC1.45 activity cannot account for all anatomical differences observed in the Neanderthal jaw, which was likely shaped by multiple genetic changes across multiple loci” - and in our revised manuscript have endeavoured to make this clearer in the results and in a new ‘Limitations’ section (see text changes below). We hope this clarification that the Neanderthal SNVs in EC1.45 are undoubtedly one piece in a much larger puzzle of genetic changes impacting jaw development will go some way towards assuaging the reviewer’s concerns about our arguments concerning the “role of the sequence variant in craniofacial development”.

Updates to the results section: “We therefore propose that alteration of the EC1.45 enhancer sequence during Neanderthal evolution may have promoted an increase in **SOX9 expression enhancer activity** during a window of craniofacial development that could contribute to altered abundance or morphology of cartilaginous precursors ... **Importantly, there are many**

additional Neanderthal SNVs across the *SOX9* regulatory domain, and hence the impact of these regulatory changes should be considered within the wider context of the locus.”

Update to concluding paragraph of the discussion and new ‘Limitations’ section:
 “~~Importantly Together~~, this work ~~implicates alteration in early jaw progenitors in shaping ultimate skeletal form and~~ highlights how regulatory function can be impacted by even very small changes to enhancer sequence and proposes that alteration in early jaw progenitors could impact final skeletal form. ... Thus, the Neanderthal-derived SNVs within EC1.45 may contribute to the larger picture of regulatory alterations leading to mandibular morphological divergence in the Neanderthal lineage. Importantly, alteration to enhancer activity for EC1.45 must be understood in the context of the entire regulatory domain, for which the impact of other Neanderthal variants remains to be explored. Nevertheless, ~~f~~Future exploration of regulatory grammar within the EC1.45 enhancer cluster ...”

1. Setting the aims

I found the introduction to the topic and the setting of aims somewhat sketchy. It is not clear from the introduction, why the Neanderthal element was chosen for further study and why the SNVs in this one element were worth pursuing in the lack of broader understanding of the potentially complex regulatory element complexity at the Neanderthal Sox9 locus. While it is a very reasonable assumption, that a key CRE found and well characterised in human (by the authors in their seminal paper) is a worthy candidate for functional assessment, without better understanding of the overall locus conservation between human and Neanderthal this element may be one of many functionally redundant elements.

We agree with the reviewer that numerous alterations to DNA sequence may contribute to changes to *SOX9* expression in development between humans and Neanderthals. We therefore identified additional Neanderthal SNVs that overlap with putative CNCC enhancers in the *SOX9* landscape, marked by H3K27ac and defined by p300 ChIP-seq peaks. With the caveat that these are only putative enhancers without functional validation, we identify an additional 48 SNVs between human and Neanderthal at these loci. Of note, these variants were selected to be present in both the Altai and Vindija high coverage genomes. In the future, we hope to explore the impact of these additional sequence variants on enhancer function.

We have added additional detail to the manuscript to describe the regulatory complexity at the *SOX9* locus, and to clarify further the rationale for focusing on the EC1.45 enhancer.

To describe the regulatory complexity at the *SOX9* locus: “~~While there are many putative regulatory elements at the *SOX9* locus, w~~We previously characterised two enhancer clusters ~~that lie~~ upstream of the PRSse translocation breakpoint ~~cluster which are each s-and-are~~ ablated by at least one ~~described PRS~~ patient deletion”

To clarify the rationale for focusing on the EC1.45 enhancer:

“~~Informed by PRS patient mutations that cause severe jaw morphological malformations, this work highlighted a key role for this-regulatory-element~~EC1.45 in shaping jaw morphology and function (Long et al. 2020). ~~While~~Based on these observations, we hypothesised that single nucleotide variants (SNVs) within EC1.45 ... may ~~also~~ alter enhancer activity, impacting to a more subtle degree *SOX9* developmental expression and ~~ultimately-shaping~~ jaw morphology.”

“We previously identified three Neanderthal-derived SNVs within EC1.45 that are associated with a Neanderthal-specific hypomethylated region, suggestive of a gain of function for the Neanderthal enhancer (Gokhman et al. 2014; 2020; Long et al. 2020).”

2. Justification of the fish model in hominin gene regulation

2.1. *For the neanderthal element function to be compared to human in a valuable and informative fashion, one would expect that the host system i.e. the zebrafish is sufficiently conserved by offering a similar developmental context both in terms of gene regulation and in*

terms of anatomy. From the gene regulation perspective, i would expect that the analysis of the EC1.45 is based on expectation of similar regulatory information content to that in the fish homolog thus one can expect similar TF network activities on them and as a result one can test sequence variation effects relevant to endogenous regulatory interactions both in fish and hominins. However, there is no data shown for the relevance of fish regulatory background as a test system. No information is provided on the fish sox9 locus and its activity, or whether the fish homolog enhancer (or any sox9 enhancer that is expressed in the expected domains of craniofacial lineages and structures) has been identified and how it compares to the hominins. One expects that the hominin enhancers are active in domains of the zebrafish sox9 for the anatomical structures to give relevant readout. I would expect a comparison and match of the EC1.45 activity to either endogenous sox9 by WISH or (although less accurate) a cross to one of the several sox9 reporter transgenic lines available on ZFIN.

Despite distinctions in adult jaw morphology compared to human, zebrafish have been extensively used as a model to investigate craniofacial developmental processes as the underlying cell biology and gene regulatory networks controlling craniofacial development are well-conserved from mammals to fish (Fox and Waskiewicz, 2024; Medeiros and Crump, 2012; Mork and Crump, 2015). *sox9* expression in the zebrafish craniofacial structures has been well-characterised (e.g. Eames et al., 2013), and *sox9a/b* mutants have defects in craniofacial and skeletal development (Yan et al. 2002; 2005). In addition, zebrafish enhancer-reporter assays have been used previously to investigate the activity of putative human *SOX9* enhancers implicated in PRS (Gordon et al, 2014). These observations supported our motivation for using the zebrafish model in this study. We have added additional references and details to the introduction, to highlight the relevance and orthology of the zebrafish model for studying human craniofacial development and disorders (see also *Reviewer-3 comment 3.1* below).

Updated introduction text: “To explore the impact of these variants, we leveraged zebrafish as a model system for exploring alterations in hominin gene regulatory mechanisms due to broadly conserved craniofacial gene regulatory networks, well-characterised and orthologous craniofacial developmental processes to human (including formation of Meckel’s cartilage), and external and transparent embryonic development, and rapid generation times (Fox and Waskiewicz, 2024; Medeiros and Crump, 2012; Mork and Crump 2015; Raterman et al. 2020). Relevant to lower jaw formation, Meckel’s cartilage represents a conserved embryonic structure that provides a scaffold for adjacent mandibular bone formation in both zebrafish and humans (Eames et al. 2013; Logjes et al. 2018; Reeck et al. 2022; Svandova et al. 2020).”

As suggested by Reviewer-3 here, and discussed above (see *Reviewer-1 comment 2*), we have now explored further the overlap between EC1.45 enhancer activity and endogenous *sox9a* expression during zebrafish development by HCR at 2 dpf, revealing a robust overlap (see *Revision Figure 2*). In addition, we have provided further evidence from our scRNA-seq and new HCR data that EC1.45-active cells robustly express *sox9a* (see *Revision Figures 1-2*).

As for exploring the *sox9a/b* regulatory domains in zebrafish, in our previous work we could not identify an orthologous EC1.45 enhancer region in the zebrafish genome at either of the two *sox9* paralogs, *sox9a* or *sox9b* (Long et al, 2020). As suggested below in *Reviewer-3 comment 4*, we have now also attempted to identify a zebrafish orthologous EC1.45 element by leveraging an intermediate species. We were able to identify part of a putative orthologous EC1.45 element in the spotted gar genome but have been unable to identify a similar element for the zebrafish. It has been shown that many enhancers can have functional conservation but divergent sequence and therefore can be challenging to identify by sequence conservation alone. In future work we would be interested to extend our search for a zebrafish ortholog of EC1.45 using synteny-based approaches (Phan et al, 2025), or conservation of transcription factor motif combinations (e.g. Cornejo-Páramo et al, 2024).

2.2. There is an argument about the regulatory networks being conserved (without references), this would need more arguments particularly in the context of Sox9/SOX9 regulation.

We are happy to include additional references supporting the conservation of craniofacial gene regulatory networks from human to zebrafish. Please see our response to *Reviewer-3 comment 2.1* above, in which we provide references supporting the conservation of craniofacial gene regulatory

networks more broadly. As for *SOX9*, we have previously shown that multiple ‘Coordinator’ motifs are present in the human EC1.45 enhancer cluster which are functionally important for regulatory activity (Long et al, 2020). It has recently been shown that TWIST1 binds to the Coordinator motif, along with ALX1, MSX1 or PRRX1 in a tissue-specific manner (Prescott et al, 2015, PMID: 26365491; Kim et al, 2024, PMID: 38262408). Furthermore, from ChIP-seq datasets we see binding of NR2F1/2 and TFAP2A at the EC1.45 enhancer (Prescott et al, 2015). These factors have well-established roles in zebrafish craniofacial development, including *twist1* (Das and Crump, 2012), *alx1* (Dee et al. 2013), *tfap2a* (Barallo-Gimeno et al. 2004), *msx1* (Phillips et al. 2006; Swartz et al. 2011) and *nr2f1* (Barske et al. 2018). We observe expression of these genes in the craniofacial clusters of our scRNA-seq dataset (*Revision Figure 12*), and in publicly available datasets (Fabian et al. 2022; Sur et al 2023). Taken together with the observed activity of the human EC1.45 element in zebrafish craniofacial regions by imaging, we feel confident in interpreting that the gene regulatory networks regulating EC1.45 in mammals, are conserved in zebrafish.

We have updated the manuscript to provide additional references to support that craniofacial regulatory networks are broadly conserved in the zebrafish (Fox and Waskiewicz, 2024; Medeiros and Crump, 2012). See also updated text for *Reviewer- 3 comment 2.1* above.

*Revision Figure 12. scRNA-seq from this study showing expression of a select panel of genes in the CNCC clusters. These include *sox9a* (left), transcription factors that bind to Coordinator motifs in human (middle) and transcription factors shown to bind the EC1.45 enhancer by ChIP-seq in human in vitro-derived CNCCs (right).*

Furthermore, since the submission of this manuscript, a preprint (Khouri-Farah et al. 2025) has reported gene expression patterns for the developing human face (CS12-20) using single-nucleus RNA-sequencing. We have leveraged this valuable resource to compare gene expression patterns observed for the EC1.45-active cells in our zebrafish model, and those of the developing human face (*Revision Figure 13*). We see expression of genes such as *SHOX*, *BARX1*, *DLX4* and *SMOC1*, in a cluster of cells from the human embryos annotated as mandibular arch1, which matches expression of the homologous genes in the PA1 CNCC clusters of our zebrafish dataset (*Revision Figure 13A-B*). Furthermore, *PAX7*, *ALX1/4*, *GATA3* and *PITX2*, appear highly expressed in the human maxillary (MxP.surface and MxP2) and lateral nasal process (LNP1/2) clusters, while the zebrafish gene homologs are enriched in the frontonasal CNCC cluster of our zebrafish dataset (*Revision Figure 13A- B*). This analysis further demonstrates the conservation of gene regulatory networks involved in facial development between human and zebrafish development.

Revision Figure 13. Comparison of gene expression patterns for the human and zebrafish developing face. (A) Expression of genes marking zebrafish PA1-like (pink bar) and frontonasal-like (blue bar) craniofacial clusters from scRNA-seq. (B) Expression of homologous human genes from (A) in snRNA-seq dataset of early human facial development (Khouri-Farah et al. 2025).

Figure 13B, data taken from: Khouri-Farah, Nagham, Emma Wentworth Winchester, Brian M. Schilder, Kelsey Robinson, Sarah W. Curtis, Nathan G. Skene, Elizabeth J. Leslie-Clarkson, and Justin Cotney. 2025. "Gene Expression Patterns of the Developing Human Face at Single Cell Resolution Reveal Cell Type Contributions to Normal Facial Variation and Disease Risk." <https://doi.org/10.1101/2025.01.18.633396>. Plots generated from <https://url.uk.m.mimecastprotect.com/s/0bxsCPJEi105yczfMixP-MK?domain=cotneyshiny.research.chop.edu/> https://cotneyshiny.research.chop.edu/shiny-apps/craniofacial_all_snRNA/

3. Further to the justification of the fish model, from the anatomical perspective, the assessment of the parallels of zebrafish and mammalian craniofacial development need strengthening.

3.1. While indeed transparency and external development helps the reporter transgenesis and argues for the fish model, but the generation time is actually comparable to mouse (in contrast to the statement in the introduction), however the understanding of zebrafish craniofacial development and its similarity to human is not well argued, and indeed very superficially compared in the manuscript. I found the anatomical analyses to be rather imprecise and difficult to compare. In the lack of direct comparisons and diagrams comparing mammalian and fish developmental structures and their origins, the statement of 'EC1.45 activity matches expression domains from mammalian development' or 'broadly recapitulate' to be an oversimplification and overstatement. The lineage tracing is an important evidence but again the anatomical homologies need to be more clearly visualized and the lineage history better explained.

Our previous assessment of EC1.45 regulatory activity in mouse was not performed at high resolution, therefore the anatomical description was somewhat limited, resulting in our comparison to the activity observed in zebrafish lacking a precise description. All we can say is that EC1.45 is active across craniofacial regions, in addition to limb/fin in both mouse and zebrafish.

The orthology between zebrafish and mammalian embryonic jaw anatomy has been extensively reviewed (e.g. Mork and Crump, 2015; Fox and Waskiewicz, 2024). We have focused much of our analyses on Meckel's cartilage as an embryonic cartilage structure that forms a scaffold for lower jaw formation in both zebrafish and mammals (Svandova et al. 2020; Eames et al. 2013; Reeck et al. 2022). We have included additional details of jaw development and orthology to the introduction text (see response to *Reviewer-3 comment 2.1*), and have added a schematic figure to Fig. 2Cii which depicts human embryonic jaw structures to further improve clarity (see also *Reviewer-3 comment 3.3* and *Revision Figure 15C* below). We have further removed the comment about rapid generation times from the introduction as indicated by the reviewer.

Description of Fig. 2Cii, highlighting orthology of Meckel's cartilage: "(see Fig. 2C and Fig. S1B for schematics of developing zebrafish cartilage structures, orthology to human Meckel's cartilage development and relative location of EC1.45-P1P2 reporter activity)."

3.2. *In a similar vein, direct comparison of human and Neanderthal adult morphologies (Figure 1B) would be very helpful.*

We thank the reviewer for this suggestion and plan to incorporate a schematic of a modern human jaw as a comparison to the Neanderthal jaw schematic in Fig. 1B (*Revision Figure 14*). We agree that this highlights more clearly distinctions between anatomically modern human and Neanderthal mandibles.

Revision Figure 14. Update to Fig. 1B to include comparison of Neanderthal mandible shape to modern human jaw morphology.

3.3. *I was also confused why the sox10 reporter is used as reference (with no direct overlap of activity to the SOX9 associated EC1.45 reporter) rather than or alongside a sox9a reporter line or even comparison to endogenous sox9a activity by WISH (Figure 2). The anatomical details in Figure 2 would need to be extended with more precisely describing the cell types, where the transgene is active and how the homology to mammalian anatomies are established.*

We agree that it is interesting and important to explore where the EC1.45 enhancer is active in relation to *sox9a* expression. We have now performed HCR for the *Tg(HuP1P2:GFP)* transgenic line for both *eGFP* and *sox9a* and find that >95% of EC1.45-active cells also express *sox9a*. See also our response to *Reviewer-1 comment 2* and new Fig. 2D-E.

When exploring where EC1.45 was active during development, we wished to leverage a lineage marker to landmark craniofacial tissues during development and facilitate live-imaging approaches. The *sox10* reporter *Tg(sox10:mRFP)* is active in CNCCs early during development, and also in chondrocytes later in development and therefore provided a useful landmark for annotating EC1.45 activity. Leveraging this reporter, we were able to determine that "enhancer activity was detected alongside Meckel's precartilaginous condensations and extended anteriorly along the oral cavity". It is difficult to determine cell types from anatomical location alone, and we hope that our scRNA-seq analysis at 2 dpf highlights further the major cell types where EC1.45 is active at this stage (e.g. updated Fig. 4D and Fig. S5D and S5G).

In response to *Reviewer-3 comments 2.1 and 3.1*, we have updated our description in the manuscript regarding the orthology between human and zebrafish jaw formation. While there are clear distinctions in jaw morphology, we have included *Revision Figure 15A-B* below to emphasize the developmental orthology between human and zebrafish of Meckel's cartilage development (the precursor structure besides which mandibular bone will form).

We have extended the schematic of the human adult jaw in Fig. 2Cii to include an embryonic stage based on an image from *Logjes et al, 2018* (see *Revision Figure 15C*).

NOTE: Figure 15A provided for reviewer has been removed. It showed Figure 2 from Logjes, Robrecht J. H., Corstiaan C. Breugem, Gijs Van Haaften, Emma C. Paes, Geoffrey H. Sperber, Marie-Jose H. van den Boogaard, and Peter G. Farlie. 2018. "The Ontogeny of Robin Sequence." *American Journal of Medical Genetics Part A* 176 (6): 1349-68. <https://doi.org/10.1002/ajmg.a.38718>. We have removed unpublished data that had been provided for the referees in confidence.

Revision Figure 15. (A) Morphological comparison of mandible formation in human (Logjes et al, 2018) and (B) zebrafish development (Eames et al, 2013; Facebase). (C) Updated Fig. 2Cii panel inset to highlight orthology of jaw development in zebrafish and human.

NOTE: Figure 15B provided was compiled from images on the FaceBase hub from the Fish Face Atlas (Eames et al, 2013). **FaceBase 3: analytical tools and FAIR resources** for craniofacial and dental research Bridget D. Samuels, Robert Aho, James F. Brinkley, Alejandro Bugacov, Eleanor Feingold, Shannon Fisher, Ana S. Gonzalez-Reiche, Joseph G. Hacia, Benedikt Hallgrimsson, Karissa Hansen, Matthew P. Harris, Thach-Vu Ho, Greg Holmes, Joan E. Hooper, Ethylin Wang Jabs, Kenneth L. Jones, Carl Kesselman, Ophir D. Klein, Elizabeth J. Leslie, Hong Li, Eric C. Liao, Hannah Long, Na Lu, Richard L. Maas, Mary L. Marazita, Jaaved Mohammed, Sara Prescott, Robert Schuler, Licia Selleri, Richard A. Spritz, Tomek Swigut, Harm van Bakel, Axel Visel, Ian Welsh, Cristina Williams, Trevor J. Williams, Joanna Wysocka, Yuan Yuan, Yang Chai. Development 2020 Sep 21;147(18):dev191213. doi: <https://url.uk.m.mimecastprotect.com/s/n-ewCYXmSWkjBhGhWixCkTO?domain=doi.org%10.1242/dev.191213>.

Eames, B Frank, April DeLaurier, Bonnie Ullmann, Tyler R Huycke, James T Nichols, John Dowd, Marcie McFadden, Mark M Sasaki, and Charles B Kimmel. 2013. "FishFace: Interactive Atlas of Zebrafish Craniofacial Development at Cellular Resolution." *BMC Developmental Biology* 13 (1): 23. <https://url.uk.m.mimecastprotect.com/s/r42pCZnSqPXBcKi1iBk4HQ?domain=doi.org><https://doi.org/10.1186/1471-213X-13-23>

3.4. Overall, the use of the fluorescence reporter is helpful for initial assessments but accurate enhancer activity profiling and comparison should be done by WISH, as mRNA is far more likely to follow the temporal activation dynamics and may explain fluorescence signal intensity differences, the latter important for correct interpretation of sequence variant effects (e.g. is the perceived higher expression by the Ne element is perhaps due to longer expression or earlier activation).

We have performed HCR at 2 dpf to explore mRNA expression profiles for *mCherry* versus *eGFP* in our dual enhancer reporter embryos (see also *Reviewer-1 comment 4*). We find that the human and Neanderthal EC1.45 enhancers are active in overlapping cells in the jaw-adjacent region at 2 dpf, with small numbers of single-positive cells at the periphery of the enhancer-active domain (*Revision figure 4 and New Fig. S4*).

To follow temporal activation dynamics for the EC1.45 enhancer element for the dual enhancer reporter lines during early facial development we have performed time-lapse imaging from 1-2 dpf (see also *Reviewer-2 comment 1 and new Movies 6-7*). Together, these experiments indicate that the Neanderthal and human enhancers are activated at around the same time and in similar cell types, thereby suggesting that differences in expression are likely mostly due to absolute differences in enhancer activity, with some contribution from additional Neanderthal-active cells at the activity domain periphery, and not due to differences in enhancer timing or activity in distinct cell types.

4. Single cell transcriptomics

*This experiment was not only used to characterise transgenic reporter active cell types, but to establish transcription factor candidates relevant to neural crest differentiation regulated by EC1.45. What is somewhat confusing, is that the EC1.45 element activity domain is only partially and not predominantly overlapping with the *twist1a* expressing cells. The authors previously established *Twist1* as key regulator of EC1.45 in craniofacial development. How do the authors explain the apparent little relevance of *twist1a* in regulating the enhancer in fish? Overall the lack of any attempt to link the SNVs to TFBS (including, if available that of the fish homolog sequences) is making the interpretation of the sequence variation harder. BTW, even of the fish elements are not directly identifiable by direct sequence alignment it may be possible to identify the fish homolog through phylogenetic footprinting with stepping stone species such as the non-duplicated paddlefish.*

In Fig. S5D, we showed in a dot plot that *twist1a* is highly expressed in the CNCC clusters of our scRNA-seq dataset at 2 dpf. We now also show a UMAP plot further illustrating a strong overlap between *twist1a* expression and EC1.45 activity, which would be consistent with the capacity of *twist1a* to regulate EC1.45 in the zebrafish (see *Revision Figure 16*). We have included this plot in updated Fig. 4D.

Revision Figure 16 (see also updated Fig. 4D). UMAP of zebrafish CNCC-like clusters displaying broad expression of twist1a (right).

We agree that linking the Neanderthal-derived SNVs with an impact to TFBSs is of great interest. We refer to *Reviewer-1 comment 9* for more details for this analysis, and new Fig. S10 and Tables S6-7.

We thank the reviewer for their comment regarding the use of the paddlefish for phylogenetic footprinting. Using this strategy, we BLAST-ed the Coelacanth min2 element against the spotted gar genome (available in ENSEMBL BLAST) and were able to identify a short sequence at the sox9 locus which was longer than any other match in the genome (*Revision Figure 17A-B*) and had two strong Coordinator motifs nearby the matched sequence, a feature of human EC1.45 (*Revision Figure 17C*). However, using this sequence we were still unable to identify an orthologous locus in the zebrafish genome. In the future, we hope to further leverage information about other transcription factor binding sites at EC1.45 to help identify a putative orthologous enhancer element in zebrafish.

NOTE: We have removed unpublished data that had been provided for the referees in confidence.

Revision Figure 17. Identification of a short orthologous region of EC1.45 in the spotted gar genome with evidence for Coordinator binding motifs. (A) A BLAST hit from Coelacanth EC1.45 sequence to the spotted gar genome. (B) ENSEMBL genome browser snapshot of BLAST hit from A (red vertical line). (C) Two strong Coordinator motifs (yellow) near the BLAST hit in the spotted gar genome (blue).

5. Sox9 overexpression

This experiment seems not to add too much to the main claim of the paper. While not essential, for this data to add more value, a comparison to that using the Neanderthal element would be more interesting and not a difficult experiment to carry out.

We considered performing SOX9 overexpression using the Neanderthal enhancer, however, we do not believe our experimental design is amenable to this approach. Our capacity to titrate the assay is limited by the number of Tol2-mediated insertions and we believe we have already saturated the developmental sensitivity to SOX9 overexpression, as we see developmental defects with the human variant at higher plasmid concentrations. The aim of this experiment was to determine the impact of slight overexpression of SOX9 in EC1.45 active cells (a population of cells which appears equivalent for the human and Neanderthal enhancer, see *Revision Figures 4-5*), recapitulating to some degree the effect of the Neanderthal SNVs, and we believe our approach has achieved this. See also our response to *Reviewer-1 comment 11*.

6. Throughout the paper there is a lack of data on reproducibility of reporter activities. As random integration often leads to position effects, it is expected that more than one lines showing the same patterns is used to identify cell type and tissue specificities. This is lacking in the paper and is a concern, as for example, the human element activity in Fig. 1 appears to be different from that by in the dual reporter shown in Fig. 3.

In the original manuscript we provide evidence of EC1.45 activity domains from four independent lines

- including three Q-STARZ dual enhancer reporter lines *Tg(Ne:GFP;Hu:Ch)*, *Tg(Hu:GFP;Ne:Ch)*, and *Tg(Hu:GFP;Hu:Ch)*, and the *Tg(HuP1P2:GFP)* single reporter. To demonstrate further the reproducibility of EC1.45 reporter activity across multiple lines, we have now imaged embryos from four independent founders for the *Tg(HuP1P2:GFP)* transgenic line (see *Revision Figure 18*).

We have included these images in a new Fig. S2, and have referred to these additional founder lines in the updated manuscript: “**These activity patterns were consistent across four founder lines (Fig. S2)**”.

To address the reviewer’s comments about differences in reporter signal between Fig. 2 and 3, we refer to our response to *Reviewer-1 comment 3*. In brief, the imaging settings were adjusted in

when quantifying the dual enhancer reporter fluorescence signal in Fig. 3 and Fig. S3A-B to avoid saturation of the brighter Neanderthal signal, hence the apparent difference of human EC1.45 activity between Fig. 3 and Fig. 2. Details of the imaging settings are provided in the Materials and Methods section.

Revision Figure 18 (new Fig. S2). Four independent *Tg(HuEC1.45-P1P2:eGFP)* transgenic reporter lines were imaged at 1, 2 and 3 days post fertilisation (dpf) demonstrating reproducibility of activity domains for EC1.45 across early zebrafish craniofacial development.

Minor points

A request to the editor as much as the authors: please make sure that legends are on the same page with figures, it is very hard to follow manuscripts when one needs to scroll between 3 pages at the same time (text, figure, legend). This archaic separation inherited from decades ago when physical prints used to be submitted has no justification in the digital era but continues to make reviewer's life difficult. Similarly, there should be no limit, and it should be encouraged to label anatomical structures directly on panels to point out expression domains, highlight expression variation, or to make a panel more self explanatory, while making sure that clarity is not lost.

We will bear these comments in mind for future bioRxiv submissions.

Figure 1A does not support the statement it is referenced to

We apologise for this error; the sentence should include a reference to data from our previous study where we explored EC1.45 activity in the developing mouse embryo. We have updated the sentence as follows: “*exhibited activity in the developing mouse craniofacial region and limb bud (Long et al. 2020).*” We have also moved the figure reference in the first sentence of the results section for clarity:

“*Given the pathogenic consequences associated with ablation of the EC1.45 enhancer cluster (Fig. 1A)*”.

Figure 1B should include huma anatomy in comparison and perhaps a schematic diagram of the hypothesized developmental morphogenesis divergence modelled in this paper

We have included a human mandible schematic in an updated version of Fig. 1B (see *Reviewer-3 comment 3.2*, and *Revision Figure 14*). We have also included a schematic of human embryonic jaw morphology in Fig. 2C above the adult human jaw image (see *Reviewer-3 comment 3.3* above, and *Revision Figure 15C*). We have a simple schematic outlining that we hypothesise jaw morphogenesis might be impacted by the Neanderthal SNVs in Fig. 6.

Figure 1D should show why the authors argue the neanderthal is not the ancestral state (BTW, what does the fish homolog look like)

We have updated Fig. 1D in response to this comment to include additional out-group species (bonobo, chimpanzee and gorilla) to demonstrate that the three SNVs appear to be Neanderthal-derived (*Revision Figure 11*). See also *Reviewer-2 comment 2*.

Figure 4A,B are better suited in Supplemental

We are happy to take the reviewer's suggestion and have moved Fig. 4A to new Fig S5A. We have discarded Fig. 4B as it is repeated in Fig. S5B (upper left).

Reviewer #3 (Significance (Required)):

Conceptual: identifying sequence variants in Neanderthal cis regulatory element as potential source of evolutionary change in morphology.

Technologically mostly following prior art, use of single cell in reporter analysis is technologically improvement on current standards, albeit somewhat rudimentary

*The use of a tractable embryo model to explore a regulatory sequence change leading to morphology change has been often applied for various aspects of evolutionary changes during development (pioneering examples include the *shh* ZRA enhancer in fin/limb morphogenesis, or balean fin evolution (PMID: 9860988) or human versus ape hand evolution (PMID: 18772437), but this is the first for applying it to hominin evolution. This will be of interest to human geneticists, evolutionary geneticists and developmental geneticists.*

My expertise is developmental gene regulation with the zebrafish model.

We thank the reviewer for their comments and for highlighting the interest of this manuscript for human geneticists, evolutionary geneticists and developmental geneticists, and emphasising the novelty of our work exploring regulatory sequence variation in the context of hominin evolution.

Barrallo-Gimeno, Alejandro, Jochen Holzschuh, Wolfgang Driever, and Ela W. Knapik. 2004. "Neural Crest Survival and Differentiation in Zebrafish Depends on Mont Blanc/Tfap2a Gene Function." *Development (Cambridge, England)* 131 (7): 1463-77. <https://doi.org/10.1242/dev.01033>.

Barske, Lindsey, Pauline Rataud, Kasra Behizad, Lisa Del Rio, Samuel G. Cox, and J. Gage Crump. 2018. "Essential Role of Nr2f Nuclear Receptors in Patterning the Vertebrate Upper Jaw." *Developmental Cell* 44 (3): 337-347.e5. <https://doi.org/10.1016/j.devcel.2017.12.022>.

Bhattacharya, Debadrita, Ana Paula Azambuja, and Marcos Simoes-Costa. 2020. "Metabolic Reprogramming Promotes Neural Crest Migration via Yap/Tead Signaling." *Developmental Cell* 53 (2): 199-211.e6. <https://doi.org/10.1016/j.devcel.2020.03.005>.

Cornejo-Páramo, Paola, Xuan Zhang, Lithin Louis, Yi-Hua Yang, Zelun Li, David Humphreys, and Emily S. Wong. 2024. "A Bag-Of-Motif Model Captures Cell States at Distal Regulatory Sequences." <https://doi.org/10.1101/2024.01.03.574012>.

Das, Ankita, and J. Gage Crump. 2012. "Bmps and Id2a Act Upstream of Twist1 to Restrict Ectomesenchyme Potential of the Cranial Neural Crest." *PLoS Genetics* 8 (5): e1002710. <https://doi.org/10.1371/journal.pgen.1002710>.

Dee, Chris T., Christoph R. Szymoniuk, Peter E. D. Mills, and Tokiharu Takahashi. 2013. "Defective Neural Crest Migration Revealed by a Zebrafish Model of Alx1-Related Frontonasal Dysplasia." *Human Molecular Genetics* 22 (2): 239-51. <https://doi.org/10.1093/hmg/ddt423>.

Vorontsov, Ilya E., Ivan V. Kulakovskiy, Grigory Khimulya, Daria D. Nikolaeva, and Vsevolod

- J. Makeev. 2015. "PERFECTOS-APE - Predicting Regulatory Functional Effect of SNPs by Approximate P-Value Estimation:" In *Proceedings of the International Conference on Bioinformatics Models, Methods and Algorithms*, 102-8. Lisbon, Portugal: SCITEPRESS -Science and and Technology Publications.
<https://doi.org/10.5220/0005189301020108>.
- Eames, B Frank, April DeLaurier, Bonnie Ullmann, Tyler R Huycke, James T Nichols, John Dowd, Marcie McFadden, Mark M Sasaki, and Charles B Kimmel. 2013. "FishFace: Interactive Atlas of Zebrafish Craniofacial Development at Cellular Resolution." *BMC Developmental Biology* 13 (1): 23. <https://doi.org/10.1186/1471-213X-13-23>.
- Fabian, Peter, Kuo-Chang Tseng, Mathi Thiruppathy, Claire Arata, Hung-Jhen Chen, Joanna Smeeton, Nellie Nelson, and J. Gage Crump. 2022. "Lifelong Single-Cell Profiling of Cranial Neural Crest Diversification in Zebrafish." *Nature Communications* 13 (1): 13. <https://doi.org/10.1038/s41467-021-27594-w>.
- Fox, Sabrina C., and Andrew J. Waskiewicz. 2024. "Transforming Growth Factor Beta Signaling and Craniofacial Development: Modeling Human Diseases in Zebrafish." *Frontiers in Cell and Developmental Biology* 12 (February):1338070. <https://doi.org/10.3389/fcell.2024.1338070>.
- Gokhman, David, Eitan Lavi, Kay Prüfer, Mario F. Fraga, José A. Riancho, Janet Kelso, Svante Pääbo, Eran Meshorer, and Liran Carmel. 2014. "Reconstructing the DNA Methylation Maps of the Neandertal and the Denisovan." *Science* 344 (6183): 523-27. <https://doi.org/10.1126/science.1250368>.
- Gokhman, David, Malka Nissim-Rafinia, Lily Agranat-Tamir, Genevieve Housman, Raquel García-Pérez, Esther Lizano, Olivia Cheronet, et al. 2020. "Differential DNA Methylation of Vocal and Facial Anatomy Genes in Modern Humans." *Nature Communications* 11 (1): 1189. <https://doi.org/10.1038/s41467-020-15020-6>.
- Gordon, Christopher T., Catia Attanasio, Shipra Bhatia, Sabina Benko, Morad Ansari, Tiong Y. Tan, Arnold Munnich, et al. 2014. "Identification of Novel Craniofacial Regulatory Domains Located Far Upstream of *SOX9* and Disrupted in Pierre Robin Sequence." *Human Mutation* 35 (8): 1011-20. <https://doi.org/10.1002/humu.22606>.
- Kharchenko, Peter V, Lev Silberstein, and David T Scadden. 2014. "Bayesian Approach to Single-Cell Differential Expression Analysis." *Nature Methods* 11 (7): 740-42. <https://doi.org/10.1038/nmeth.2967>.
- Khouri-Farah, Nagham, Emma Wentworth Winchester, Brian M. Schilder, Kelsey Robinson, Sarah W. Curtis, Nathan G. Skene, Elizabeth J. Leslie-Clarkson, and Justin Cotney. 2025. "Gene Expression Patterns of the Developing Human Face at Single Cell Resolution Reveal Cell Type Contributions to Normal Facial Variation and Disease Risk." <https://doi.org/10.1101/2025.01.18.633396>.
- Kim, Seungsoo, Ekaterina Morgunova, Sahin Naqvi, Seppe Goovaerts, Maram Bader, Mervenez Koska, Alexander Popov, et al. 2024. "DNA-Guided Transcription Factor Cooperativity Shapes Face and Limb Mesenchyme." *Cell* 187 (3): 692-711.e26. <https://doi.org/10.1016/j.cell.2023.12.032>.
- Logjes, Robrecht J. H., Corstiaan C. Breugem, Gijs Van Haaften, Emma C. Paes, Geoffrey H. Sperber, Marie-José H. van den Boogaard, and Peter G. Farlie. 2018. "The Ontogeny of Robin Sequence." *American Journal of Medical Genetics Part A* 176 (6): 1349-68. <https://doi.org/10.1002/ajmg.a.38718>.
- Long, Hannah K., Marco Osterwalder, Ian C. Welsh, Karissa Hansen, James O.J. Davies, Yiran E. Liu, Mervenez Koska, et al. 2020. "Loss of Extreme Long-Range Enhancers in Human Neural Crest Drives a Craniofacial Disorder." *Cell Stem Cell* 27 (5): 765-783.e14. <https://doi.org/10.1016/j.stem.2020.09.001>.
- Medeiros, Daniel Meulemans, and J. Gage Crump. 2012. "New Perspectives on Pharyngeal Dorsoventral Patterning in Development and Evolution of the Vertebrate Jaw." *Developmental Biology* 371 (2): 121-35. <https://doi.org/10.1016/j.ydbio.2012.08.026>.
- Mork, Lindsey, and Gage Crump. 2015. "Zebrafish Craniofacial Development." In *Current Topics in Developmental Biology*, 115:235-69. Elsevier. <https://doi.org/10.1016/bs.ctdb.2015.07.001>.
- Phan, Mai H. Q., Tobias Zehnder, Fiona Puntieri, Andreas Magg, Blanka Majchrzycka, Milan Antonović, Hannah Wieler, et al. 2025. "Conservation of Regulatory Elements with Highly Diverged Sequences across Large Evolutionary Distances." *Nature Genetics* 57 (6): 1524-34. <https://doi.org/10.1038/s41588-025-02202-5>.
- Phillips, Bryan T., Hye-Joo Kwon, Colt Melton, Paul Houghtaling, Andreas Fritz, and Bruce B.

- Riley. 2006. “Zebrafish msxB, msxC and msxE Function Together to Refine the Neural-Nonneural Border and Regulate Cranial Placodes and Neural Crest Development.” *Developmental Biology* 294 (2): 376-90. <https://doi.org/10.1016/j.ydbio.2006.03.001>.
- Prescott, Sara L., Rajini Srinivasan, Maria Carolina Marchetto, Irina Grishina, Iñigo Narvaiza, Licia Selleri, Fred H. Gage, Tomek Swigut, and Joanna Wysocka. 2015. “Enhancer Divergence and Cis-Regulatory Evolution in the Human and Chimp Neural Crest.” *Cell* 163 (1): 68-83. <https://doi.org/10.1016/j.cell.2015.08.036>.
- Raterman, S. T., J. R. Metz, Frank A. D. T. G. Wagener, and Johannes W. Von Den Hoff. 2020. “Zebrafish Models of Craniofacial Malformations: Interactions of Environmental Factors.” *Frontiers in Cell and Developmental Biology* 8 (November):600926. <https://doi.org/10.3389/fcell.2020.600926>.
- Reeck, Jonathon C., and Julia Thom Oxford. 2022. “The Shape of the Jaw—Zebrafish Col11a1a Regulates Meckel’s Cartilage Morphogenesis and Mineralization.” *Journal of Developmental Biology* 10 (4): 40. <https://doi.org/10.3390/jdb10040040>.
- Sur, Abhinav, Yiqun Wang, Paulina Capar, Gennady Margolin, Morgan Kathleen Prochaska, and Jeffrey A. Farrell. 2023. “Single-Cell Analysis of Shared Signatures and Transcriptional Diversity during Zebrafish Development.” *Developmental Cell* 58 (24): 3028-3047.e12. <https://doi.org/10.1016/j.devcel.2023.11.001>.
- Svandova, Eva, Neal Anthwal, Abigail S. Tucker, and Eva Matalova. 2020. “Diverse Fate of an Enigmatic Structure: 200 Years of Meckel’s Cartilage.” *Frontiers in Cell and Developmental Biology* 8 (August):821. <https://doi.org/10.3389/fcell.2020.00821>.
- Swartz, Mary E., Kelly Sheehan-Rooney, Michael J. Dixon, and Johann K. Eberhart. 2011. “Examination of a Palatogenic Gene Program in Zebrafish.” *Developmental Dynamics* 240 (9): 2204-20. <https://doi.org/10.1002/dvdy.22713>.
- Wang, Jun, Yang Xiao, Chih-Wei Hsu, Idaliz M. Martinez-Traverso, Min Zhang, Yan Bai, Mamoru Ishii, et al. 2016. “Yap and Taz Play a Crucial Role in Neural Crest-Derived Craniofacial” <https://doi.org/10.1242/dev.126920>.
- Williamson, Kathleen A., Joe Rainger, James A. B. Floyd, Morad Ansari, Alison Meynert, Kishan V. Aldridge, Jacqueline K. Rainger, et al. 2014. “Heterozygous Loss-of-Function Mutations in YAP1 Cause Both Isolated and Syndromic Optic Fissure Closure Defects.” *The American Journal of Human Genetics* 94 (2): 295-302. <https://doi.org/10.1016/j.ajhg.2014.01.001>.
- Yan, Yi-Lin, Craig T. Miller, Robert M. Nissen, Amy Singer, Dong Liu, Anette Kirn, Bruce Draper, et al. 2002. “A Zebrafish Sox9 Gene Required for Cartilage Morphogenesis.” *Development (Cambridge, England)* 129 (21): 5065-79. <https://doi.org/10.1242/dev.129.21.5065>.
- Yan, Yi-Lin, John Willoughby, Dong Liu, Justin Gage Crump, Catherine Wilson, Craig T. Miller, Amy Singer, Charles Kimmel, Monte Westerfield, and John H. Postlethwait. 2005. “A Pair of Sox: Distinct and Overlapping Functions of Zebrafish Sox9 Co-Orthologs in Craniofacial and Pectoral Fin Development.” *Development (Cambridge, England)* 132 (5): 1069-83. <https://doi.org/10.1242/dev.01674>.

To aid navigation of updated figure numbering, we have provided tables here:

Old Figure	New Fig.
Figure 1A	Fig. 1A
Figure 1B	Fig. 1B
Figure 1C	Fig. 1C
Figure 1D	Fig. 1D
Figure 2A	Fig. 2A
Figure 2B	Fig. 2B
Figure 2C	Fig. 2C
	Fig. 2D
	Fig. 2E
Figure 3A	Fig. 3A
Figure 3B	Fig. 3B
Figure 3C	Fig. 3C
Figure 4A	Fig. S5A
Figure 4B	Fig. S5B
Figure 4C	Fig. 4A
Figure 4D	Fig. 4B
Figure 4E	Fig. 4C
Figure 4F	Fig. 4D
Figure 4G	Fig. 4E
Figure 5A	Fig. 5A
Figure 5B	Fig. 5B
Figure 5C	Fig. 5C
Figure 5D	Fig. 5D
Figure 6	Fig. 6

Old Supplementary Figure	New Fig. S
Supplementary Figure 1A	Fig. S1A
Supplementary Figure 1B	Fig. S1B
Supplementary Figure 1C	Fig. S1C
Supplementary Figure 1D	Fig. S1D
	Fig. S2
Supplementary Figure 2A	Fig. S3A
Supplementary Figure 2B	Fig. S3B
Supplementary Figure 2C	Fig. S3C
Supplementary Figure 2D	Fig. S3D
Supplementary Figure 2E	Fig. S3E
	Fig. S4A -F
Figure 4A	Fig. S5A
Supplementary Figure 3A + Figure 4B	Fig. S5B
Supplementary Figure 3B	Fig. S5C
Supplementary Figure 3C	Fig. S5D
Supplementary Figure 3D	Fig. S5E
	Fig. S5F
	Fig. S5G
Supplementary Figure 3E	Fig. S6A
Supplementary Figure 3F	Fig. S6B
Supplementary Figure 3G	Fig. S6C
Supplementary Figure 4A	Fig. S7A
Supplementary Figure 4B	Fig. S7B
Supplementary Figure 5A	Fig. S8A
Supplementary Figure 5B	Fig. S8B
Supplementary Figure 5C	Fig. S8C
Supplementary Figure 6A	Fig. S9A
Supplementary Figure 6B	Fig. S9B
Supplementary Figure 6C	Fig. S9C
	Fig. S10A-F

Second decision letter

MS ID#: dev.204779R1

MS Title: Neanderthal-derived variants at a craniofacial disease locus shape enhancer activity across recent human evolution

Authors: Kirsty Uttley; Hannah J. Jüllig; Carlo De Angelis; Julia M. T. Auer; Ewa Ozga; Hemant

Bengani; Hannah K. Long

Article Type: Review Commons Transfer

Dear Dr Long,

I am happy to tell you that your manuscript has been accepted for publication in Development, pending our standard publication integrity checks.

Comments from the Reviewers:

Reviewer 1: The authors have done an excellent job of addressing the reviewers' comments. I have no additional concerns and recommend acceptance.

Reviewer 2: COMMENTS ON TEXT

no further comments

COMMENTS ON DISPLAY ITEMS

no further comments